# DANCETOGETHER: GENERATING INTERACTIVE MULTI-PERSON VIDEO WITHOUT IDENTITY DRIFTING

**Junhao Chen**[1,2]  **Mingjin Chen**[3]  **Jianjin Xu**[4]  **Xiang Li**[5]  **Junting Dong**[6,†]
**Mingze Sun**[1]  **Puhua Jiang**[1]  **Hongxiang Li**[7]  **Yuhang Yang**[8]
**Hao Zhao**[2,10,†]  **Xiaoxiao Long**[9]  **Ruqi Huang**[1,†]

[1]Shenzhen International Graduate School, Tsinghua University  [2]AIR, Tsinghua University
[3]The Hong Kong Polytechnic University  [4]Carnegie Mellon University
[5]Peking University  [6]Shanghai AI Laboratory  [7]The Hong Kong University of Science and Technology
[8]University of Science & Technology of China  [9]Nanjing University  [10]BAAI

**Project Page:** `https://dancetog.github.io/`

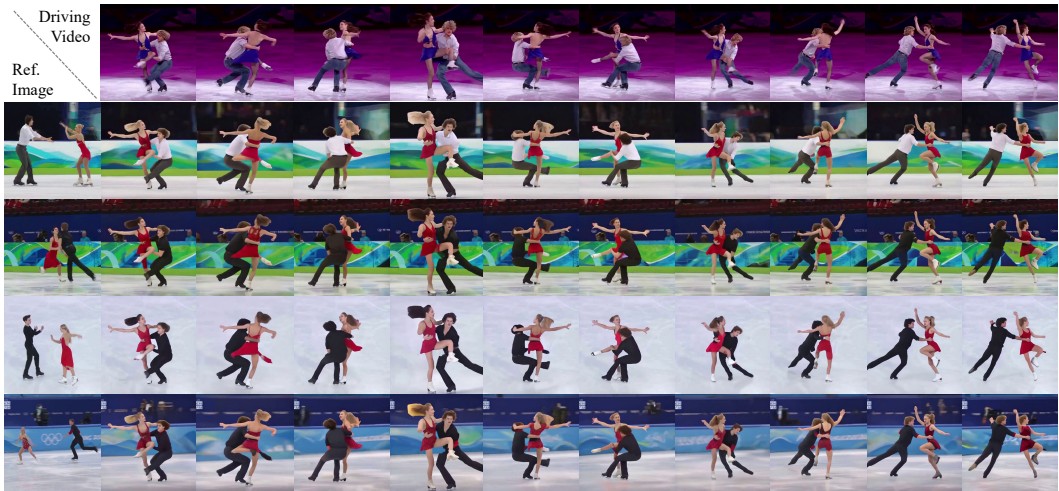

Figure 1: *DanceTogether* generates complex two-person interaction videos with interactive details and consistent identity preservation from a single reference image (see the left-most of each row), using independent multi-person pose and mask sequences as control signals.

## ABSTRACT

Controllable video generation (CVG) has advanced rapidly, yet current systems falter when more than one actor must move, interact, and exchange positions under noisy control signals. We address this gap with *DanceTogether*, the first end-to-end diffusion framework that turns a single reference image plus independent pose–mask streams into long, photorealistic videos while *strictly preserving every identity*. A novel *MaskPoseAdapter* binds "who" and "how" at every denoising step by fusing robust tracking masks with semantically rich—but noisy—pose heat-maps, eliminating the identity drift and appearance bleeding that plague frame-wise pipelines. To train and evaluate at scale, we introduce (i) `PairFS-4K`, 26 h of dual-skater footage with 7,000+ distinct IDs, (ii) `HumanRob-300`, a one-hour humanoid–robot interaction set for rapid cross-domain transfer, and (iii) `TogetherVideoBench`, a three-track benchmark centred on the `DanceTogEval-100` test suite covering dance, boxing, wrestling, yoga, and figure skating. On `TogetherVideoBench`, *DanceTogether* outperforms the prior arts by significant margin. Moreover, we show that a one-hour fine-tune yields convincing human–robot videos, underscoring broad generalization to embodied-AI and HRI tasks. Extensive ablations confirm that persistent identity–action binding is critical to these gains. Together, our model, datasets, and benchmark lift CVG from single-subject choreography to **compositionally controllable, multi-actor interaction**, opening new avenues for digital production, simulation, and embodied intelligence.

# 1 INTRODUCTION

*Controllable video generation* (CVG) (Ma et al., 2024) seeks to translate explicit control signals—*e.g.* per-frame human poses, body masks, or trajectory commands—into photorealistic human-motion videos. Compared to AI generation tasks that use single conditioning (reference images or text) (Podell et al., 2024; OpenAI, 2024; Khachatryan et al., 2023), some controllable generation tasks typically combine multi-modal conditions as input (Zhang et al., 2023b; Peng et al., 2024; Hu, 2024; Chen et al., 2025a). Such tasks using multi-modal control signals have broad and important applications in film production (Bugliarello et al., 2025; Song et al., 2024), digital human (Chen et al., 2025c; Tian et al., 2025; Miao et al., 2026; Wang et al., 2025c; Sun et al., 2024b), and embodied AI (Albaba et al., 2025). In particular, we investigate the task of CVG with multi-person interactions, which is highly challenging as it simultaneously requires (i) **preserve the identities of multiple actors** over hundreds of frames, (ii) **maintain the spatio-temporal coherence of complex interactions** such as hand-holding, lifts, position exchanges, and synchronous choreography, and (iii) **faithfully obey noisy control signals** in the presence of occlusion, motion blur, and rapid viewpoint changes.

Most existing systems adopt a frame-wise synthesis followed by temporal smoothing paradigm: each image is generated independently from pose or text conditions and then stitched into a video via interpolation, optical-flow warping, or temporal convolutions (Di Chang et al., 2023; Ma et al., 2024; Zhu et al., 2024b). Nearly all of these models are trained solely on single-person dance datasets (Zhu et al., 2024a; Xu et al., 2024c; Hu, 2024; Zhang et al., 2024b; Wang et al., 2024c). A handful of works incorporate multi-person footage (Wang et al., 2024e; Zhang et al., 2024a; Xue et al., 2024), but they exhibit pronounced *identity drift* and appearance bleeding when the actors exchange positions. In general, state-of-the-art methods struggle with identity inconsistency, cross-subject contamination, and missing interaction details—issues that rapidly worsen once more than one performer is involved.

We present *DanceTogether*, the first end-to-end diffusion framework expressly tailored for controllable multi-person interaction video generation. Our guiding hypothesis is that robust multi-actor synthesis requires an *explicit, persistent binding between identity and motion* throughout the diffusion process. To this end, we deliberately disentangle identity from action and then re-couple them: instead of relying solely on fragile pose estimates, we fuse stable tracking masks with semantically rich pose cues. This fusion is realised by a novel conditional adapter, MaskPoseAdapter, which combines the *reliable, easy-to-obtain body masks* with the *informative yet noisy poses* into a bimodal control signal. By integrating each subject's mask and pose into a unified representation, the adapter enforces precise identity-to-action alignment at every generative step.

Our framework operationalizes the identity–action binding principle through three tightly coupled modules. (i) MultiFace Encoder distills a compact set of identity tokens from a single image and injects them into every cross-attention layer, ensuring subject appearance is held constant throughout the sequence. (ii) MaskPoseAdapter fuses robust per-person tracking masks with semantically rich—but noisy—pose maps to deliver a bimodal conditional signal that aligns "who" and "how" at every diffusion step, thereby safeguarding both identity integrity and motion fidelity. (iii) Video Diffusion Backbone leverages these aligned signals to synthesize high-resolution clips whose multi-actor motions remain coherent, physically plausible, and free of identity drift.

Extensive evaluation on the new TogetherVideoBench—built around our 100-clip DanceTogEval-100 set—shows that DanceTogether decisively advances controllable multi-person video generation. Across the three core tracks (Identity-Consistency, Interaction-Coherence, Video Quality) it raises the bar over the strongest prior (StableAnimator (Tu et al., 2024) +swing dance data (Maluleke et al., 2024) finetune) by +12.6 HOTA, +7.1 IDF1, +5.9 MOTA, trims $\text{MPJPE}_{2D}$ by 69 % ($1555 \rightarrow 492$ px), and boosts OKS/PoseSSIM to 0.83/0.93. Visual fidelity also improved accordingly: human mask region FVD/FID decreased from 29.0/66.7 to 17.1/48.0, without sacrificing CLIP alignment effect. Fine-tuning on our proposed one-hour HumanRob-300 dataset can generate convincing human-robot interaction videos, which highlights the framework's broad generalization capability and prospects in embodied AI research.

To summarize, our main contributions include:

1. **DanceTogether framework.** We present the first end-to-end diffusion framework for controllable multi-person interaction video generation. Our novel *MaskPoseAdapter* fuses stable tracking masks with pose cues to enforce identity-action binding throughout the generation process.

2. **Data curation pipeline and datasets.** We develop a monocular-RGB pipeline for extracting tracking-aware human poses and masks. Using this, we curate `PairFS-4K` (26h dual-person figure skating) and `HumanRob-300` (1h robot interaction) datasets.

3. **TogetherVideoBench benchmark.** We introduce a comprehensive evaluation benchmark with three tracks (*Identity-Consistency*, *Interaction-Coherence*, *Video Quality*) and `DanceTogEval-100` containing 100 dual-actor clips across diverse activities.

4. **Superior performance and generalization.** Our method achieves significant improvements: +12.6 HOTA, +7.1 IDF1, +5.9 MOTA over the strongest baseline, 69% reduction in pose error, and enhanced visual fidelity (FVD: 29.0→17.1). Cross-domain fine-tuning demonstrates strong generalization to human-robot scenarios.

## 2 RELATED WORK

### 2.1 DIFFUSION MODELS FOR VIDEO GENERATION

Diffusion models have achieved significant progress in video generation (Poole et al., 2022; Li et al., 2023; Kong et al., 2024; Li et al., 2024; Wang et al., 2024a;b; Chen et al., 2024; Lin et al., 2024; Han et al., 2024; Liu et al., 2025; Yang et al., 2025b). Early works used 3D-UNet for spatiotemporal fusion (Singer et al., 2022; Ho et al., 2022). Building on this, (Blattmann et al., 2023b) introduced temporal dimensions into latent diffusion models; (Ho et al., 2022) employed cascaded spatial-temporal super-resolution; (Xu et al., 2024a) enabled multimodal control through spatiotemporal consistency modeling; (Blattmann et al., 2023a) proposed temporal interpolation strategies for 2D-to-video adaptation. Despite advances in commercial models (Kong et al., 2024; Kuaishou, 2025; OpenAI, 2024) achieving good temporal consistency and resolution, they remain inadequate for fine-grained human motion control tasks.

### 2.2 CONTROLLABLE HUMAN VIDEO GENERATION

Diffusion models (Rombach et al., 2022; Blattmann et al., 2023a) have significantly advanced controllable human video generation. Most methods build on pre-trained Stable Diffusion with pose guidance, using keypoints or skeletons as conditions (Zhang et al., 2023b; Hu, 2024). Disco (Wang et al., 2024d) separates background and pose control, a strategy extended by (Hu, 2024; Xu et al., 2024c). Other approaches use geometric priors from 3D models (Wang et al., 2024c; Zhang et al., 2024b) or SMPL models (Zhu et al., 2024a; Li et al., 2025a), but remain limited to single-person scenarios. Despite these advances, most methods struggle with multi-person interactions and identity consistency. Recent works (Wang et al., 2024e; Luo et al., 2025; Ma et al., 2024; Xue et al., 2024; Karras et al., 2023; Feng et al., 2023; Tu et al., 2024) explore identity preservation using encodings or masks (Yoon et al., 2021), but are limited to short videos. Some methods (Wang et al., 2024c; Zhang et al., 2024b) use local masks or attention mechanisms, yet lack explicit identity-action binding, causing drift in long sequences.

### 2.3 RECENT MULTI-PERSON ANIMATION WORKS

Multi-person controllable animation (Huang et al., 2024; Xu et al., 2024b; Chang et al., 2025) has recently advanced through works like Multi-HumanVid (Wang et al., 2025d) and Follow-your-multipose (Zhang et al., 2024a) demonstrating spatial guidance for motion synthesis. Concurrent mask-based approaches include MagicMotion (Li et al., 2025b), Multi-HumanVid (Wang et al., 2025d), EverybodyDance (Ling et al.), and ReMask-Animate (Xiang et al., 2025), which leverage masks, geometric priors, or identity correspondence for controllable generation. DiT-based models like Wan2.2 (Cheng et al., 2025; Jiang et al., 2025) and commercial systems like Kling (Kuaishou, 2025) and Veo3.1 (Google, 2025) enable text-driven animation with strong temporal consistency. However, these methods primarily handle scenarios with fixed relative positions and lack fine-grained per-person control, making them inadequate for **identity drift during position exchanges**—when people cross paths or heavily occlude each other. Our approach addresses this gap through explicit identity-action binding via gated mask-pose fusion and cross-person attention (Sec. 3.2), preserving consistent identities throughout dynamic interactions and position swaps.

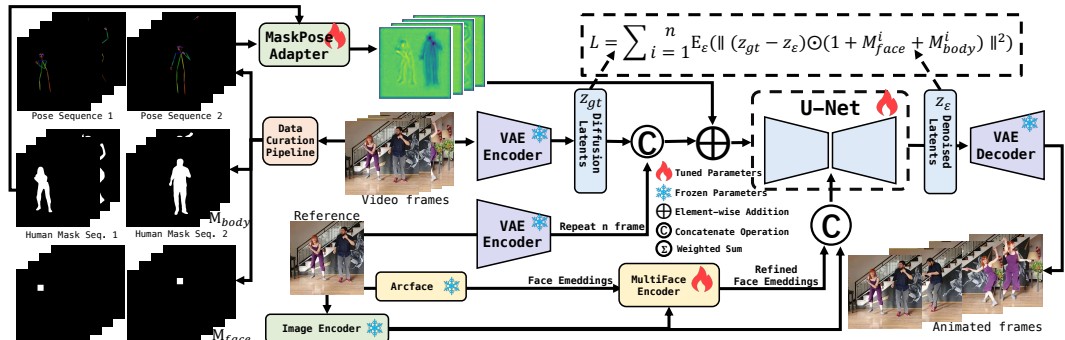

Figure 2: *DanceTogether* pipeline overview: A single reference image and per-person pose/mask sequences enter the system; the MaskPoseAdapter fuses these control signals, the MultiFace Encoder injects identity tokens, and the video-diffusion backbone synthesizes an interaction video that preserves consistent identities for all actors. The implementation details of MaskPoseAdapter and MultiFaceEncoder can be found in Sec. A.

## 3 METHOD

### 3.1 VIDEO DIFFUSION BACKBONE

**Starting point – *StableAnimator*.** Our backbone follows the **StableAnimator** architecture (Tu et al., 2024): a 16-frame latent UNet $f_\theta$ derived from Stable Video Diffusion (SVD). For every training clip we take as input $\left(\mathbf{I}_{\text{ref}}, \mathbf{P}_{1:T}, \mathbf{M}_{1:T}\right)$ where $\mathbf{I}_{\text{ref}} \in \mathbb{R}^{3 \times H \times W}$ is a reference image, and $\mathbf{P}_t / \mathbf{M}_t$ are the pose map and tracking mask at frame $t$.

**Three conditioning streams.** The UNet is conditioned by three streams, each of which begins with a *frozen* pretrained encoder and is then refined by trainable adapters (see Fig. 2):

- **Latent image stream.** A frozen SVD VAE encoder maps both the reference image $\mathbf{I}_{\text{ref}}$ and each input video frame to their respective latent representations. The reference latent $\mathbf{z}_{\text{ref}} \in \mathbb{R}^{C \times 64 \times 64}$ is tiled along the temporal axis and concatenated with the per-frame latents $\mathbf{z}_{gt}$. This concatenated tensor is then fused with the *trainable* MaskPoseAdapter's condition latents via element-wise addition, producing the final latent input to the UNet.
- **CLIP image embeddings.** A frozen ViT-H/14 encoder $\phi_{\text{CLIP}}$ produces $\mathbf{e}^{\text{clip}} \in \mathbb{R}^{1024}$. These embeddings serve as keys/values in every *trainable* cross-attention block.
- **Refined face embeddings.** A frozen ArcFace model $\phi_{\text{ID}}$ outputs $\mathbf{e}^{\text{id}} \in \mathbb{R}^{512}$, which is then refined by the *trainable* MultiFaceEncoder $g_\psi$:

$$\mathbf{E}^{\text{face}} = g_\psi\left(\mathbf{e}^{\text{id}}, \mathbf{e}^{\text{clip}}\right) \in \mathbb{R}^{K \times d}, \tag{1}$$

implemented as four Perceiver-IO layers ($K = 4$, $d = 768$). The resulting identity tokens modulate the same trainable cross-attention layers.

**Distribution-aware ID Adapter.** To prevent a feature-distribution shift when injecting identity tokens, StableAnimator inserts an ID Adapter before each temporal block. Given input features $\mathbf{h}$, we first apply spatial self-attention and two cross-attention steps, then align and fuse the face branch to the image branch in a single fused update:

$$\hat{\mathbf{h}} = \text{SA}(\mathbf{h}), \quad \mathbf{h}_{\text{img}} = \text{CA}(\hat{\mathbf{h}}, \mathbf{e}_{\text{clip}}), \quad \mathbf{h}_{\text{face}} = \text{CA}(\hat{\mathbf{h}}, \mathbf{E}_{\text{face}}),$$
$$\tilde{\mathbf{h}}_{\text{face}} = \frac{\mathbf{h}_{\text{face}} - \mu_{\text{face}}}{\sigma_{\text{face}}} \sigma_{\text{img}} + \mu_{\text{img}}, \quad \mathbf{h}_{\text{out}} = \mathbf{h}_{\text{img}} + \tilde{\mathbf{h}}_{\text{face}}. \tag{2}$$

Here SA/CA denote self-/cross-attention, $(\mu, \sigma)$ are the per-token mean and standard deviation, and $\mathbf{E}_{\text{face}}$ the set of $K$ identity tokens. By matching the first and second moments of the face and image features, this adapter preserves identity information consistently across all frames.

**Human-tracking masked reconstruction loss.** Building upon StableAnimator's face-focused loss, we incorporate per-person *binary* masks for face and body regions. Original $512 \times 512$ masks are downsampled via nearest-neighbor interpolation to the latent resolution $64 \times 64$. Given $N$ individuals with binary masks $M_{\text{face}}^i, M_{\text{body}}^i \in \{0,1\}^{1 \times 64 \times 64}$, we optimize

$$\mathcal{L}_{\text{rec}} = \sum_{i=1}^{N} \mathbb{E}_{\epsilon \sim \mathcal{N}(0,1)} \left\| (\mathbf{z}_{\text{gt}} - \mathbf{z}_\epsilon) \odot \left( 1 + M_{\text{body}}^i + 2\, M_{\text{face}}^i \right) \right\|_2^2. \tag{3}$$

Here body masks have weight 1 and face masks weight 2, encouraging the model to focus capacity on identity-critical regions while preserving overall reconstruction fidelity.

## 3.2 MASKPOSEADAPTER: BINDING IDENTITY TO ACTION

**Motivation.** Pose keypoints alone cannot reliably distinguish individuals when they overlap or exchange positions. Conversely, binary masks lack semantic motion information. We propose to *fuse* these complementary signals in a learned feature space, explicitly binding each person's identity (via masks) to their motion (via poses).

**Architecture overview.** For each person $i$, we process pose maps $\mathbf{P}_i \in \mathbb{R}^{3 \times 512 \times 512}$ and masks $\mathbf{M}_i \in \{0,1\}^{1 \times 512 \times 512}$ through separate encoders, producing pose features $\mathbf{f}_i^{\text{pose}} \in \mathbb{R}^{320 \times 64 \times 64}$ and compact mask features $\mathbf{f}_i^{\text{mask}} \in \mathbb{R}^{3 \times 64 \times 64}$. The key innovation lies in three stages:

**(1) Gated fusion.** We learn per-pixel gates that adaptively weight pose vs. mask contributions:

$$\mathbf{f}_i = \lambda \cdot w_i^{\text{pose}} \odot \mathbf{f}_i^{\text{pose}} + (1 - \lambda) \cdot w_i^{\text{mask}} \odot \mathbf{f}_i^{\text{mask}} + \text{residual}, \tag{4}$$

where $w_i^{\text{pose}}, w_i^{\text{mask}} \in [0,1]$ are gate activations and $\lambda \approx 0.8$ biases toward pose. This allows the network to emphasize identity (mask) when poses are ambiguous, and vice versa. A lightweight PoseEnhancer further refines local details.

**(2) Cross-person attention.** To handle occlusions and position exchanges, we apply LayerNorm to each $\mathbf{f}_i$ and concatenate across persons. A shallow network produces attention logits, normalized via temperature-scaled softmax:

$$\alpha_{\text{att}} = \text{softmax}(\phi([\text{LN}(\mathbf{f}_1), \ldots, \text{LN}(\mathbf{f}_N)])/\tau), \tag{5}$$

where $\tau$ is learnable. This reweights person importance dynamically—e.g., emphasizing the foreground actor during occlusion.

**(3) Integration.** The final conditioning $\mathbf{F} \in \mathbb{R}^{C \times 64 \times 64}$ combines attention-weighted features with a small residual mean:

$$\mathbf{F} = 0.95 \cdot \text{Conv}_{1 \times 1} \left( \sum_{i=1}^{N} \alpha_{\text{att},i} \odot \mathbf{f}_i \right) + 0.05 \cdot \frac{1}{N} \sum_{i=1}^{N} \mathbf{f}_i, \tag{6}$$

which is reshaped to $(B, T, C, 64, 64)$ and injected into the UNet via element-wise addition.

**Why it works.** By fusing masks and poses at the feature level with learned gating and cross-person reasoning, MaskPoseAdapter creates a unified representation where identity and motion are inseparable. This eliminates the identity confusion observed when processing poses alone (see ablations in Sec. B). Full implementation details—including gate architectures, channel dimensions, and layer specifications—are provided in Sec. A.1.

## 3.3 MULTIFACE ENCODER: COMPACT IDENTITY INJECTION

To handle multiple identities efficiently, we extend single-person identity encoding to a multi-identity regime. Each ArcFace embedding $\mathbf{e}_i^{\text{id}} \in \mathbb{R}^{512}$ is projected to $K = 4$ tokens via a two-layer MLP, then refined through a lightweight FacePerceiver (4-layer cross-attention with CLIP embeddings as memory). The resulting tokens $\mathbf{t}_i \in \mathbb{R}^{K \times 768}$ for all $N$ persons are concatenated:

$$\mathbf{T} = [\mathbf{t}_1; \mathbf{t}_2; \ldots; \mathbf{t}_N] \in \mathbb{R}^{(NK) \times 768}. \tag{7}$$

This extended token sequence modulates every cross-attention layer in the UNet without architectural changes. Compared to naively duplicating face encoders, this shared-weight design reduces parameters while enabling the model to learn inter-identity relationships. Detailed layer configurations and attention mechanisms are in Sec. A.2.

### 3.4 LIGHTWEIGHT EXTENSION TO MULTIPLE PERSONS

MaskPoseAdapter's modular design enables effortless scaling to arbitrary person counts. To extend from two to $N$ persons, we instantiate $N$ independent branches with shared weights. The cross-person attention (Equation 5) naturally generalizes without modification.

**Training strategy.** We leverage single and dual-person data through adaptive branch activation: for single-person clips, randomly activate one branch (zero others); for two-person clips, randomly activate two branches; for $N$-person clips, activate all branches. New branches initialize by copying weights from trained branches. This curriculum requires no architectural changes, additional losses, or multi-person training data.

### 3.5 DATA CURATION PIPELINE

To address the lack of two-person interaction datasets with diverse identities, static backgrounds, and fixed cameras, we propose a comprehensive data curation pipeline that recovers poses and mask annotations from monocular RGB videos. As shown in Fig. 3, our pipeline segments videos into scenes, detects and tracks individuals using YOLOv8x (Jocher et al., 2023) and OSNet-based ReID (Zhou et al., 2019; 2021), and selects primary subjects based on coverage and consistency. We then generate high-quality per-person masks and 133-point pose annotations using SAMURAI (Yang et al., 2024), DWPose (Yang et al., 2023), and MatAnyone (Yang et al., 2025a), followed by automatic and manual filtering to ensure data quality. We aggregate a wide range of single- and two-person motion datasets—including TikTokDataset (Jafarian & Park, 2021), Champ (Zhu et al., 2024a), DisPose (Li et al., 2025a), HumanVid (Wang et al., 2024e), Swing Dance (Maluleke et al., 2024), Harmony4D (Khirodkar et al., 2024), CHI3D (Fieraru et al., 2023), Beyond Talking (Sun et al., 2025), and our newly collected `PairFS-4K`—to maximize identity diversity and interaction types. `PairFS-4K`, comprising 4.8K figure skating segments and over 7,000 unique identities, is the first large-scale two-person figure skating video dataset. All datasets are summarized in Tab. 1, providing a rich foundation for controllable human interaction video generation in real-world scenarios. More details of the Data Curation Pipeline can be found in Sec. C. For specifics on the collection and processing of `PairFS-4K`, please refer to Sec. C.5. The segmentation module can easily be substituted with newer segmentation models Ravi et al. (2024); Tian et al. (2023); Chen et al. (2022); Liu et al. (2023a).

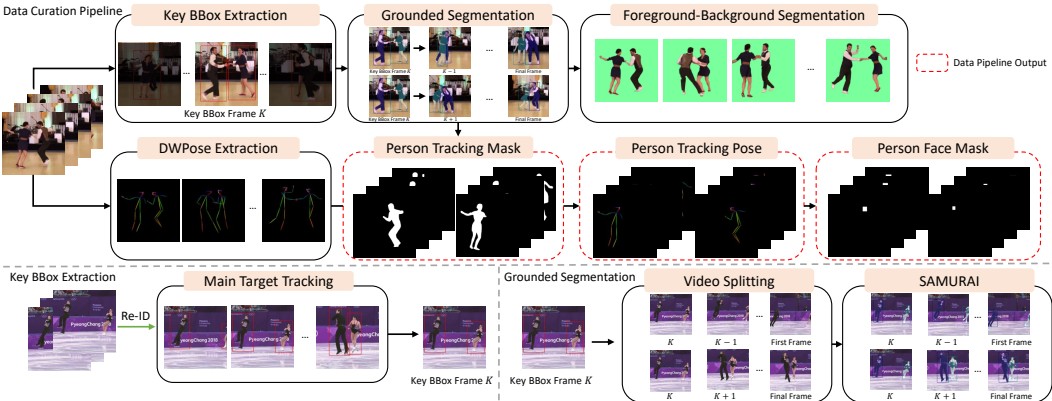

Figure 3: Data Curation Pipeline Overview. Our pipeline processes raw videos through human tracking, mask generation with SAMURAI (Vlasic et al., 2008), pose estimation with DW-Pose (Yang et al., 2023), and alpha matting to produce per-person annotations.

### 3.6 TOGETHERVIDEOBENCH BENCHMARK

We introduce **TogetherVideoBench**, a comprehensive benchmark for controllable multi-person video generation, which systematically evaluates three orthogonal tracks: *Identity-Consistency*, *Interaction-Coherence*, and *Video Quality*. Please refer to details Sec. D in the Appendix.

Table 1: Summary of datasets used in DanceTogether training. *Static competition background; †Static laboratory background; ‡Multi-view setup.

| Dataset | Type | Action | IDs | Total | Avg. | Scene | Camera |
|---|---|---|---|---|---|---|---|
| TikTokDataset (Jafarian & Park, 2021) | Single | Dance | 332 | 1.03 hrs | 11 s | Static | Fixed |
| Champ (Zhu et al., 2024a) | Single | Dance | 832 | 9.73 hrs | 42 s | Static | Fixed |
| DisPose (Li et al., 2025a) | Single | Dance | 8,636 | 38.12 hrs | 11 s | Static | Fixed |
| HumanVid (Wang et al., 2024e) | Single | Dance | 16,310 | 89.89 hrs | 17 s | Dynamic | Moving |
| Hi4D (Yin et al., 2023) | Double | Interact | 40 | 0.10 hrs | 3.6 s | Static† | Fixed‡ |
| Harmony4D (Khirodkar et al., 2024) | Double | Interact | 24 | 0.58 hrs | 12 s | Static† | Fixed‡ |
| CHI3D (Fieraru et al., 2023) | Double | Interact | 6 | 1.75 hrs | 4 s | Static† | Fixed‡ |
| Swing Dance (Maluleke et al., 2024) | Double | Dance | 1,356 | 23.36 hrs | 122 s | Static* | Moving |
| HoCo (Sun et al., 2025) | Double | Talking Head | 26 | 45 hrs | 7 s | Static† | Fixed |
| **PairFS-4K** | Double | Figure Skating | 7,273 | 26.87 hrs | 20 s | Static* | Moving |
| **HumanRob-300** | Single | Robot Interact | 336 | 0.83 hrs | 9 s | Dynamic | Moving |
| **DanceTogEval-100** | Double | Interact & Dance | 200 | 0.54 hrs | 20 s | Static | Fixed |

# 4 RESULTS

## 4.1 EXPERIMENTAL SETUP

We collect several publicly available video datasets, as detailed in Sec. C.1. We utilize DWPose (Yang et al., 2023) and ArcFace (Deng et al., 2019) to extract skeletal poses and facial embeddings/masks. To evaluate the robustness of our model, we conduct experiments on DanceTogEval-100, a curated set of 100 previously unseen two-person interaction videos from the internet. Following recent advances in animation generation, we initialize our U-Net, PoseNet, and Face Encoder with the pre-trained weights from StableAnimator, then further train them on large-scale single-person datasets (Tu et al., 2024; Li et al., 2025a; Zhu et al., 2024a; Wang et al., 2024e). We subsequently transfer the pre-trained weights to our proposed MaskPoseAdapter and MultiFace Encoder, and perform full fine-tuning using multi-person datasets—including our proposed PairFS dataset (Maluleke et al., 2024; Sun et al., 2025; Khirodkar et al., 2024; Fieraru et al., 2023). Our model is trained for 20 epochs on 8 NVIDIA A100 80G GPUs, with a batch size of 1 per GPU and a learning rate set to $1e - 5$.

## 4.2 BASELINES

We compare our approach with state-of-the-art pose-conditioned human video generation models, including AnimateAnyone, Champ, MimicMotion, HumanVid, UniAnimate, UniAnimate-DiT, DisPose, and StableAnimator. In particular, we fine-tune StableAnimator for 40 epochs on the dual-person dancing subset from the Swing Dance dataset (Maluleke et al., 2024), and include this fine-tuned variant as a new baseline in our evaluation. Fig. 4 compares our proposed *DanceTog* with four strong baselines, AnimateAnyone, HumanVid, UniAnimate, and StableAnimator – all of which achieve relatively high scores in the quantitative evaluation. All baselines exhibit severe identity drift, loss of interaction details, or even missing subjects when dealing with position exchanges and complex interactive poses. Additional comparisons, including more baselines and dual-person interaction examples, are provided in the Sec. I. **Video comparison results with baselines are provided in the Supp. Mat.** User study results are provided in Sec. E, demonstrating that our proposed benchmark aligns with human preferences.

Table 2: Multiple Object Tracking results on `TogetherVideoBench`. * Negative values occur because the sum of false positives (FP) and false negatives (FN) exceeds the number of ground truth objects. This happens when the frames only contain a single person.

| Method | Venue | HOTA family | | | CLEAR/MOTA family | | Identity |
|---|---|---|---|---|---|---|---|
| | | HOTA↑ | DetA↑ | AssA↑ | MOTA↑ | MOTP↑ | IDF1↑ |
| Animate Anyone (Hu, 2024) | CVPR 2024 | 41.26 | 39.99 | 43.21 | 26.67 | 75.73 | 51.54 |
| Champ (Zhu et al., 2024a) | ECCV 2024 | 19.32 | 14.78 | 26.32 | -19.54* | 67.92 | 17.84 |
| MimicMotion (Zhang et al., 2024b) | Arxiv 2024 | 21.14 | 16.06 | 30.50 | -55.77* | 62.13 | 15.24 |
| HumanVid (Wang et al., 2024e) | NeurIPS 2024 | 56.12 | 58.89 | 53.69 | 58.86 | 84.20 | 68.84 |
| UniAnimate (Wang et al., 2025a) | SCIS 2025 | 48.43 | 46.71 | 50.69 | 42.33 | 80.74 | 59.58 |
| UniAnimate-DiT (Wang et al., 2025b) | Arxiv 2025 | 35.02 | 31.15 | 40.65 | 10.66 | 77.99 | 39.51 |
| DisPose (Li et al., 2025a) | ICLR 2025 | 20.68 | 15.91 | 29.47 | -52.49* | 62.00 | 15.42 |
| StableAnimator (Tu et al., 2024) | CVPR 2025 | 67.75 | 67.91 | 67.70 | 69.62 | 87.67 | 79.37 |
| StableAnimator w. $Data_{swing}$ | CVPR 2025 | 71.35 | 70.91 | 71.89 | 73.89 | 88.22 | 82.53 |
| **DanceTog w. $Data_{swing}$** | – | 80.26 | 74.44 | 86.57 | 73.68 | 95.45 | 86.28 |
| **DanceTog w. $Data_{full}$** | – | 81.79 | 77.19 | 86.69 | 77.04 | 95.69 | 87.73 |
| **DanceTog w. $Data_{full} + Data_{PairFS}$** | – | 83.94 | 79.48 | 88.68 | 79.80 | 95.49 | 89.59 |

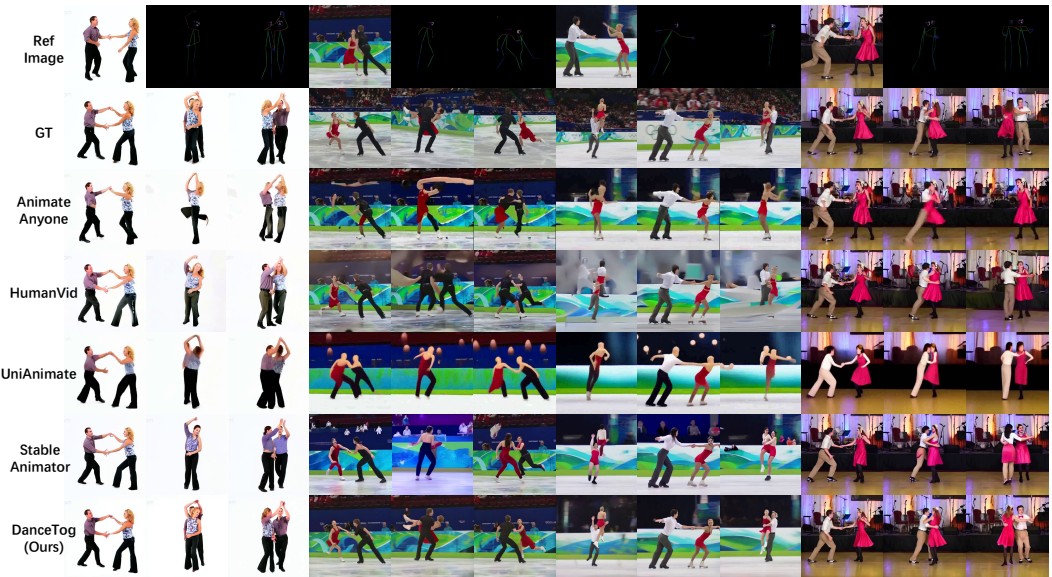

Figure 4: The RGB image in the "Ref. Image" row is the input reference frame, and the two pose maps in that row correspond to the inference results shown immediately below.

Table 3: Comparison of models across interaction coherence metrics.

| Method | MPJPE$_{2D}\downarrow$ | OKS$\uparrow$ | PoseSSIM$\uparrow$ | SmoothRMS$\downarrow$ ($\times10^6$) | TimeDyn$_{RMSE}\downarrow$ ($\times10^4$) | FVMD$\downarrow$ ($\times10^5$) |
|---|---|---|---|---|---|---|
| Animate Anyone | 3255.07 | 0.27 | 0.67 | 1.26 | 2.43 | 1.87 |
| Champ | 4117.88 | 0.06 | 0.78 | 0.78 | 1.60 | 0.90 |
| MimicMotion | 5542.99 | 0.09 | 0.74 | 1.02 | 1.94 | 1.15 |
| HumanVid | 3480.74 | 0.48 | 0.78 | 1.09 | 2.10 | 1.11 |
| UniAnimate | 2286.26 | 0.37 | 0.72 | 1.24 | 2.36 | 2.13 |
| UniAnimate-DiT | 2184.81 | 0.22 | 0.71 | 1.53 | 2.92 | 3.72 |
| DisPose | 2791.60 | 0.08 | 0.73 | 1.07 | 2.04 | 1.36 |
| StableAnimator | 1571.50 | 0.63 | 0.82 | 0.96 | 1.84 | 1.00 |
| StableAnimator w. $Data_{swing}$ | 1555.16 | 0.70 | 0.84 | 0.89 | 1.72 | 0.77 |
| **DanceTog w.** $Data_{swing}$ | 858.99 | 0.75 | 0.88 | 0.84 | 1.62 | 0.51 |
| **DanceTog w.** $Data_{full}$ | 557.60 | 0.81 | 0.92 | 0.85 | 1.64 | 0.66 |
| **DanceTog w.** $Data_{full} + Data_{PairFS}$ | 492.24 | 0.83 | 0.93 | 0.83 | 1.59 | 0.54 |

## 4.3 QUANTITATIVE RESULTS

**Track 1: Identity–Consistency.** Tab. 2 reports multiple-object-tracking (MOT) scores on *DanceTogEval-100*. Across all eight published baselines, StableAnimator fine-tuned on SwingDance (*StableAnimator* +$Data_{swing}$) is the previous best performer, reaching 71.35 HOTA and 82.53 IDF1. *DanceTogether* markedly exceeds this strong baseline on *every single metric*: with full training data it lifts HOTA from 71.35 to 81.79 (+10.44 %) and IDF1 from 82.53 to 87.73 (+6.3 %), while pushing AssA to 86.69. Adding the proposed PairFS-4K dataset provides a further gain, culminating in 83.94 HOTA, 89.59 IDF1, and a 79.80 MOTA. These results establish a new state of the art for long-range identity preservation under frequent occlusions and position exchanges.

**Track 2: Interaction–Coherence.** Tab. 3 evaluates how faithfully each method follows the target motion and how smoothly the interaction unfolds. Our model slashes MPJPE$_{2D}$ by 68 % relative to the top baseline (from 1555 px to 492 px) and attains the highest OKS (0.83) and PoseSSIM (0.93). At the same time, *DanceTogether* records the lowest motion-discontinuity scores—SmoothRMS and TimeDyn$_{RMSE}$ —indicating physically plausible, temporally consistent choreography. Champ achieves high scores on SmoothRMS and TimeDyn$_{RMSE}$ due to its use of estimated SMPL as guidance, which incorporates smoothing methods in the process of generating SMPL sequences. These two metrics only compare the motion continuity of each individual person without applying weights to the pair. Champ's inference results typically contain only a single person. The FVMD is halved compared with *StableAnimator* (0.54 vs. 1.00), further corroborating superior interaction quality.

**Track 3: Video Quality.** Tab. 4 present full-frame and mask-aware appearance metrics. Benefiting from dense identity–action binding and the high-diversity PairFS-4K corpus, *DanceTogether* delivers the best perceptual fidelity in both settings. In full-frame evaluation it attains the lowest FVD (76.3) and FID (75.1), alongside the highest CLIP score (0.95) and ST-SSIM (0.70). Within the human-masked regions—the areas most sensitive to identity drift—mask-aware FVD plunges from 29.0 to **17.1**, and C-FID shrinks from 12.5 to **7.9**, highlighting crisp texture reproduction and identity accuracy. Notably, these improvements are achieved without sacrificing low-level reconstruction fidelity: L1 and LPIPS fall in tandem, while PSNR and SSIM increase. Complete quantitative evaluation results can be found in Sec. F.

Table 4: Comparison of models on Full Frame and Human Masked Region.

| Method | L1↓ | | PSNR↑ | | SSIM↑ | | LPIPS↓ | | FVD↓ | | FID↓ | |
| | Full | Masked | Full | Masked | Full | Masked | Full | Masked | Full | Masked | Full | Masked |
|---|---|---|---|---|---|---|---|---|---|---|---|---|
| AnimateAnyone | 37.32 | 59.92 | 13.23 | 10.45 | 0.49 | 0.92 | 0.56 | 0.06 | 108.2 | 44.8 | 118.1 | 101.4 |
| Champ | 43.70 | 83.35 | 11.93 | 8.36 | 0.49 | 0.92 | 0.56 | 0.07 | 125.7 | 69.2 | 114.6 | 178.7 |
| MimicMotion | 52.08 | 77.02 | 11.04 | 8.75 | 0.47 | 0.92 | 0.58 | 0.07 | 121.0 | 65.5 | 116.6 | 180.9 |
| HumanVid | 38.93 | 47.97 | 13.67 | 12.13 | 0.52 | 0.93 | 0.50 | 0.05 | 97.2 | 34.9 | 90.2 | 72.4 |
| UniAnimate | 37.95 | 56.34 | 13.62 | 11.05 | 0.55 | 0.92 | 0.53 | 0.06 | 132.0 | 45.0 | 151.2 | 109.8 |
| UniAnimate-DiT | 43.11 | 64.48 | 12.34 | 9.89 | 0.50 | 0.91 | 0.53 | 0.06 | 111.9 | 51.4 | 100.3 | 119.6 |
| DisPose | 42.52 | 76.75 | 12.28 | 8.93 | 0.54 | 0.92 | 0.54 | 0.07 | 127.4 | 64.7 | 127.9 | 196.0 |
| StableAnimator | 33.44 | 48.51 | 14.60 | 12.00 | 0.57 | 0.93 | 0.44 | 0.05 | 85.7 | 38.4 | 84.1 | 71.8 |
| StableAnimator w. $Data_{swing}$ | 30.31 | 41.41 | 15.27 | 13.06 | 0.60 | 0.93 | 0.42 | 0.04 | 78.8 | 29.0 | 79.3 | 66.7 |
| **DanceTog w.** $Data_{swing}$ | 32.62 | 34.49 | 15.12 | 14.76 | 0.59 | 0.94 | 0.44 | 0.03 | 79.3 | 21.5 | 82.1 | 57.5 |
| **DanceTog w.** $Data_{full}$ | 29.94 | 32.80 | 15.81 | 15.15 | 0.61 | 0.94 | 0.42 | 0.03 | 76.9 | 20.6 | 77.6 | 56.1 |
| **DanceTog w.** $Data_{full} + Data_{PairFS}$ | 29.52 | 30.14 | 15.85 | 15.82 | 0.61 | 0.94 | 0.42 | 0.03 | 76.3 | 17.1 | 75.1 | 48.0 |

## 4.4 ABLATION STUDY

We perform ablation studies demonstrating that each proposed component, MaskPoseAdapter for pose-mask fusion, MultiFaceEncoder for multi-identity encoding, and PairFS-4K dataset—contributes significantly to performance improvements (see Sec. B for details).

## 4.5 INFERENCE EFFICIENCY

DanceTog achieves competitive inference speed (0.88 fps) while maintaining superior generation quality, significantly faster than DiT-based methods like UniAnimate-DiT (0.03 fps), and comparable to baselines such as StableAnimator (0.89 fps) and AnimateAnyone (0.97 fps). Like all diffusion-based human animation methods, DanceTog is not real-time but provides a practical balance between quality and efficiency for controllable multi-person video generation. All baseline methods are tested on a single A800 GPU. See Sec. G for details on training/inference costs and multi-person training support of baselines.

Table 5: Inference speed comparison (fps).

| Method | FPS |
|---|---|
| **DanceTog (Ours)** | **0.88** |
| Champ | 1.42 |
| Animate Anyone | 0.97 |
| StableAnimator | 0.89 |
| HumanVid | 0.81 |
| Dispose | 0.42 |
| UniAnimate | 0.25 |
| MimicMotion | 0.06 |
| UniAnimate-DiT | 0.03 |

## 5 CONCLUSION

We present DanceTogether, the first end-to-end diffusion framework for generating long, photorealistic multi-actor videos from a single reference image and independent pose–mask streams, while strictly preserving each identity. Our method integrates a novel MaskPoseAdapter for persistent identity–action alignment and a MultiFace Encoder for compact appearance encoding. Trained on our newly curated multi-actor datasets and evaluated on a comprehensive benchmark, DanceTogether outperforms all existing pose-conditioned video generation models by a significant margin. It generalizes well across domains, as demonstrated by convincing human–robot interactions after minimal adaptation. This work marks a step forward toward compositionally controllable, identity-aware video synthesis, laying a foundation for future advances in digital content creation, simulation, and embodied AI.

## ACKNOWLEDGMENTS

This work is supported in part by the National Natural Science Foundation of China under contract No. 62171256, and in part by the Shanghai Artificial Intelligence Laboratory.

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

## A  DETAILS OF THE PROPOSED COMPONENTS

### A.1  MASKPOSEADAPTER

Relying solely on pose keypoints (pose maps) makes it difficult to distinguish different individuals in multi-person scenarios; directly treating binary tracking masks as additional channels would compromise the translational invariance of the pose encoder. We therefore propose **MaskPoseAdapter**: first performing lightweight transformations on masks in the "pose feature space," then injecting them into pose latents using a gated-weighting strategy, and finally applying cross-person soft-attention to reorder per-person importance. Fig. 5 illustrates MaskPoseAdapter, which fuses independent pose streams and masks into a single pose–mask latent $\mathbf{F} \in \mathbb{R}^{B\times C\times 64\times 64}$.

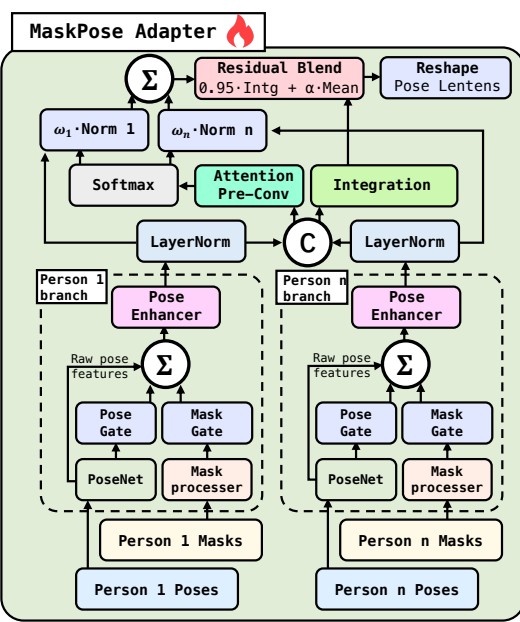

Figure 5: Details of the MaskPoseAdapter.

**Per-person Pose Encoding.** For each person $i$, an independent PoseNet processes the RGB pose map $\mathbf{P}_i \in \mathbb{R}^{3\times 512\times 512}$. PoseNet consists of eight convolutional layers, expanding the channels from 3 to 128, followed by a $1\times 1$ convolution, with weights shared across all persons. The output pose features are then scaled by a learnable factor $s$. The final output can be expressed as:

$$\mathbf{f}_i^{\text{pose}} = s\cdot\text{Conv}_{1\times 1}\big(\text{PoseNet}(\mathbf{P}_i)\big) \ \in \ \mathbb{R}^{C\times 64\times 64},\ C = 320, \tag{8}$$

**Light Mask Processor.** Binary human/facial masks $\mathbf{M}_i \in \{0,1\}^{1\times 512\times 512}$ are processed through two $3\times 3$ convolutional layers to produce a 3-channel feature map:

$$\mathbf{f}_i^{\text{mask}} = \psi(\mathbf{M}_i) \in \mathbb{R}^{3\times 64\times 64}, \tag{9}$$

which preserves contour information while avoiding mask features from dominating the pose features.

**Gate-based Fusion.** We apply two per-pixel gates to control how much of the pose and mask features to trust. These gates are implemented as convolutional layers followed by Sigmoid activations. The gate outputs are:

$$w_i^{\text{pose}} = \sigma\big(\gamma(\tilde{\mathbf{f}}_i^{\text{pose}})\big), \quad w_i^{\text{mask}} = \sigma\big(\eta(\mathbf{f}_i^{\text{mask}})\big), \tag{10}$$

where $\gamma$ and $\eta$ are each a Conv→SiLU→Conv→Sigmoid sequence. The gated features are then combined with a learnable weight $\lambda \approx 0.8$ as:

$$\tilde{\mathbf{f}}_i = \underbrace{\lambda\, w_i^{\text{pose}} \odot \tilde{\mathbf{f}}_i^{\text{pose}}}_{\text{ID-dominant}} + \underbrace{(1-\lambda)\, w_i^{\text{mask}} \odot \mathbf{f}_i^{\text{mask}}}_{\text{fine mask}}, \tag{11}$$

where $\tilde{\mathbf{f}}_i^{\text{pose}}$ is the pose feature reduced to 3 channels for gating. A residual link is added to refine the fusion, where the coefficient $\alpha_{\text{res}}$ controls the strength of the residual term:

$$\mathbf{f}_i = \tilde{\mathbf{f}}_i + \alpha_{\text{res}}\big((1-\lambda)\, w_i^{\text{mask}} \odot \mathbf{f}_i^{\text{mask}}\big), \quad \alpha_{\text{res}} = 0.5. \tag{12}$$

**Pose Enhancer.** The fusion output is passed through a lightweight *PoseEnhancer* module consisting of a $3\times 3$ convolution, followed by SiLU activation and BatchNorm, and a $1\times 1$ convolution:

$$\mathbf{h}_i = \text{PoseEnhancer}(\mathbf{f}_i). \tag{13}$$

To further refine the pose features, a scaling factor $s_p = 1.5$ is applied to the raw features before final integration:

$$\mathbf{f}_i = s_p \cdot \mathbf{f}_i + (1-\alpha_{\text{res}}) \cdot \mathbf{h}_i. \tag{14}$$

**LayerNorm and Attention.** Each of the enhanced pose features $\mathbf{f}_i$ is normalized per-channel using LayerNorm, resulting in $\bar{\mathbf{f}}_i$. The normalized features are concatenated along the channel dimension and processed through a lightweight attention mechanism consisting of three $1 \times 1$ convolution layers, each followed by BatchNorm and ReLU. This generates attention logits $\ell_i$ for each person:

$$\ell = \phi\big[\text{LayerNorm}(\tilde{\mathbf{f}}_1), \ldots, \text{LayerNorm}(\tilde{\mathbf{f}}_N)\big] \in \mathbb{R}^{N \times 64 \times 64}. \tag{15}$$

These logits are normalized across the person dimension using a temperature-scaled softmax function:

$$\alpha_{\text{att}} = \text{SoftmaxWithTemp}_\tau(\ell), \quad \text{SoftmaxWithTemp}_\tau(x) = \text{softmax}(x/\tau), \tag{16}$$

where $\tau$ is a learnable temperature parameter.

**Cross-Person Integration.** The normalized features are integrated via attention-weighted sum $S = \sum_{i=1}^N \alpha_{\text{att},i} \odot \bar{\mathbf{f}}_i$, followed by a $1 \times 1$ convolution and residual averaging:

$$\mathbf{F} = 0.95 \cdot \text{Conv}_{1\times1}(S) + 0.05 \cdot \frac{1}{N}\sum_{i=1}^N \bar{\mathbf{f}}_i, \tag{17}$$

where $\mathbf{F} \in \mathbb{R}^{C \times 64 \times 64}$ is reshaped to $(B, T, C, 64, 64)$ and injected into the UNet.

## A.2 MULTIFACE ENCODER

Fig. 6 illustrates MultiFace Encoder. For every mini-batch we receive $\mathbf{E}^{\text{id}} \in \mathbb{R}^{N \times B \times D_{\text{id}}}$ with $D_{\text{id}} = 512$ and $D_{\text{clip}} = 1024$, where the first axis enumerates the $N$ identities and the second the $B$ samples in the batch. Each sample also carries a length-1 CLIP embedding $\mathbf{e}^{\text{clip}} \in \mathbb{R}^{B \times 1 \times D_{\text{clip}}}$, which is used as key/value memory in all cross-attention steps.

**Stage I — Per-identity token projection.** For identity $i \in \{1, \ldots, N\}$ and sample $b$ we transform the ArcFace vector $\mathbf{e}_{i,b}^{\text{id}}$ with a two-layer MLP (`Linear(512,1024)` $\to$ `GELU` $\to$ `Linear(1024,KD)`) and reshape it into $K = 4$ learnable tokens of width $D = 768$:

$$\tilde{\mathbf{x}}_{i,b} = \text{MLP}_{2\times\text{GELU}}(\mathbf{e}_{i,b}^{\text{id}}) \in \mathbb{R}^{KD}, \tag{18}$$

$$\mathbf{t}_{i,b}^{(0)} = \text{LN}\big(\text{reshape}_{K\times D}(\tilde{\mathbf{x}}_{i,b})\big) \in \mathbb{R}^{K \times D}. \tag{19}$$

**Stage II — FacePerceiver refinement.** The $K$ latent tokens $\mathbf{t}_{i,b}^{(0)}$ query a lightweight *FacePerceiver* with depth $L_p = 4$:

$$\mathbf{t}_{i,b}^{(\ell+1)} = \mathbf{t}_{i,b}^{(\ell)} + \text{FFN}\Big(\mathbf{t}_{i,b}^{(\ell)} + \text{CrossAttn}\big(\mathbf{t}_{i,b}^{(\ell)}, \mathbf{e}_b^{\text{clip}}\big)\Big), \quad \ell = 0, \ldots, 3. \tag{20}$$

Queries originate from the latent tokens, whereas keys/values are the concatenation of the projected CLIP embedding and the tokens (cf. `PerceiverAttention` in the code). A residual shortcut controlled by the flags `shortcut`, `scale` ($\lambda$) reproduces the exact behaviour of `MultiFace Encoder`:

$$\mathbf{t}_{i,b} = \begin{cases} \mathbf{t}_{i,b}^{(4)}, & \text{shortcut} = 0, \\ \mathbf{t}_{i,b}^{(0)} + \lambda\,\mathbf{t}_{i,b}^{(4)}, & \text{shortcut} = 1. \end{cases} \tag{21}$$

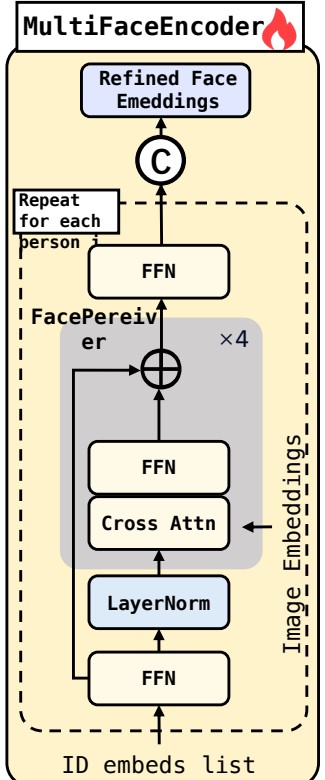

Figure 6: Details of the Multi-Face Encoder.

**Stage III — Multi-person concatenation.** After processing all identities with *shared* weights, the refined tokens are stacked along the sequence axis:

$$\mathbf{T}_b = \big[\mathbf{t}_{1,b}; \mathbf{t}_{2,b}; \ldots; \mathbf{t}_{N,b}\big] \in \mathbb{R}^{(NK) \times D}, \qquad \mathbf{T} \in \mathbb{R}^{B \times NK \times D} \text{ for the batch.} \tag{22}$$

The UNet's cross-attention layers can therefore read $\mathbf{T}$ directly, gaining $NK$ extra tokens without any architectural change.

# B ABLATION STUDY

## B.1 DATASET ABLATION STUDY

Ablation study on the datasets have been compared in the main text in Tabs. 2, 3, 8, and 9. StableAnimator (Tu et al., 2024) fine-tuned for 40 epochs on the swing dance dataset (Maluleke et al., 2024) (StableAnimator w. $Data_{swing}$) shows significant improvement over the original pre-trained weights provided by the authors, but still performs noticeably worse than DanceTog trained for 20 epochs on the same Swing dance dataset (**DanceTog w.** $Data_{swing}$). Using all training data except PairFS-4K (**DanceTog w.** $Data_{full}$) clearly performs better than the model trained only on the swing dance dataset (**DanceTog w.** $Data_{swing}$), but still underperforms compared to DanceTog trained on all data including PairFS-4K (denoted as **DanceTog w.** $Data_{full} + Data_{PairFS}$).

## B.2 ABLATION STUDY ON SUB-MODULES AND INPUTS

Tab. 6 presents ablation results for the new model and multi-input approaches proposed in DanceTog. The following configurations are evaluated:

- **w/o mask input**: The model does not use a separate mask input during the input process.
- **w/o pose input**: The model does not use a separate pose input during the input process.
- **w/o MaskPoseAdapter**: The original PoseNet is used instead, where poses of all individuals are directly fed as condition inputs to the model.
- **w/ SimpleFusion**: Instead of using MaskPoseAdapter's fusion strategy, the Pose Enhancer latents from individual person branches are directly summed element-wise, bypassing the multi-person attention mechanism.
- **w/o MultiFaceEncoder**: The original FaceEncoder is used instead, where only the embedding of the largest detected face in the reference image serves as the condition input.

| Model Variant | Track 1: Identity–Consistency | | | Track 2: Interaction–Coherence | | | | Track 3: Video Quality | | | |
|---|---|---|---|---|---|---|---|---|---|---|---|
| | HOTA↑ | MOTA↑ | IDF1↑ | MPJPE$_{2D}$↓ | OKS↑ | PoseSSIM↑ | FVMD×$10^5$↓ | PSNR↑ | FVD↓ | FID↓ | C-FID↓ |
| w/o mask input | 33.63 | 15.48 | 42.49 | 1625.04 | 0.28 | 0.85 | 2.97 | 11.02 | 40.4 | 73.1 | 14.7 |
| w/o pose input | 81.48 | 74.23 | 86.38 | 1292.33 | 0.46 | 0.85 | 4.91 | 14.98 | 19.7 | 58.1 | 9.4 |
| w/o MaskPoseAdapter | 48.95 | 40.93 | 62.02 | 1692.55 | 0.48 | 0.79 | 3.80 | 11.19 | 41.3 | 72.0 | 14.2 |
| w/ SimpleFusion | 58.23 | 46.15 | 65.71 | 1487.26 | 0.39 | 0.84 | 3.82 | 12.85 | 31.2 | 64.8 | 12.4 |
| w/o MultiFaceEncoder | 83.31 | 78.81 | 88.55 | 893.32 | 0.74 | 0.89 | 1.26 | 15.67 | 17.9 | 49.2 | 8.4 |
| **DanceTog** | 83.94 | 79.80 | 89.59 | 492.24 | 0.83 | 0.93 | 0.54 | 15.82 | 17.1 | 48.0 | 7.9 |

Table 6: Module ablation study.

Fig. 7 and Fig. 8 present qualitative comparisons of our ablation studies. DanceTogether is compatible with StableAnimator's "Inference with HJB-based Face Optimization" (Tu et al., 2024). Since our task and test samples focus on full-body two-person interaction video generation rather than large-area face-mask talking heads or single-person half-body dance sequences, the benefit of HJB-based Face Optimization is less pronounced. In our tests, inference without HJB-based Face Optimization runs at approximately 0.8 s/iteration, whereas enabling HJB-based Face Optimization reduces throughput to about 15 s/iteration. Furthermore, our ablation study indicates that applying HJB-based Face Optimization does not significantly impact the quality of two-person interaction video generation. Consequently, all experiments reported in the main text for StableAnimator and DanceTog were performed without HJB-based Face Optimization.

## B.3 COMPARISON BETWEEN POSENET AND MASKPOSEADAPTER

Fig. 9 shows the feature maps obtained by the original PoseNet and our proposed MaskPoseAdapter from consecutive frames with the same input. It can be clearly observed that the output of MaskPoseAdapter strongly binds pose and mask information, enabling clear identification of which ID each pose corresponds to, and still providing sufficient mask information even when input poses are missing in some occluded frames. In contrast, the original PoseNet's output makes it difficult to distinguish each individual pose, and pose features may be lost in occluded frames.

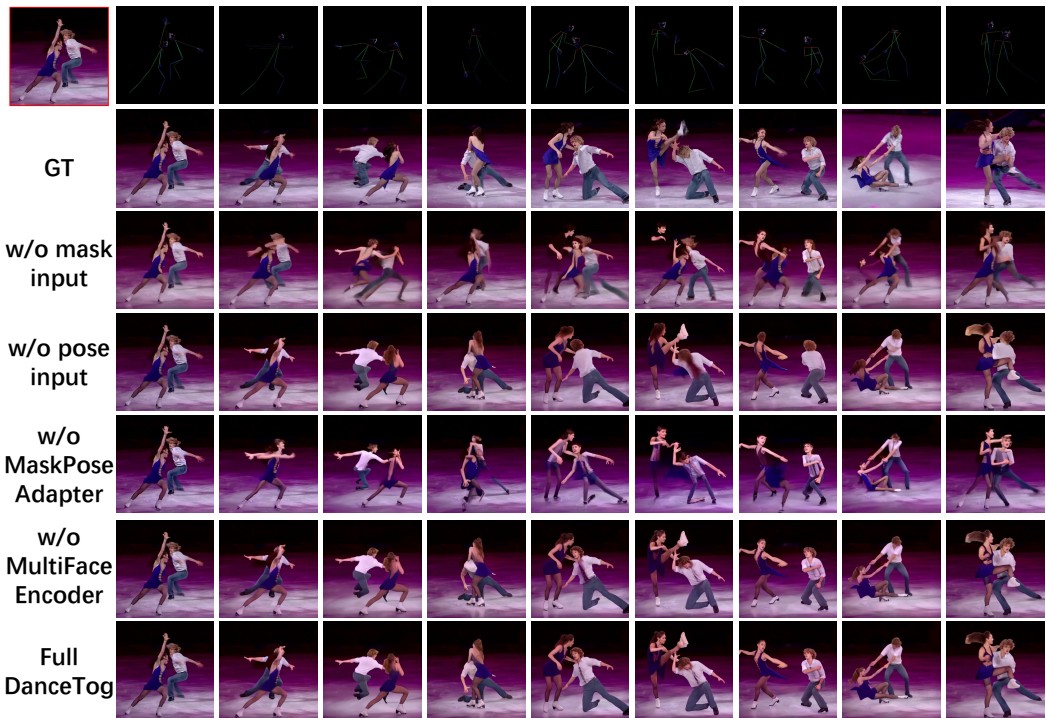

Figure 7: Ablation study animation results (1/2).

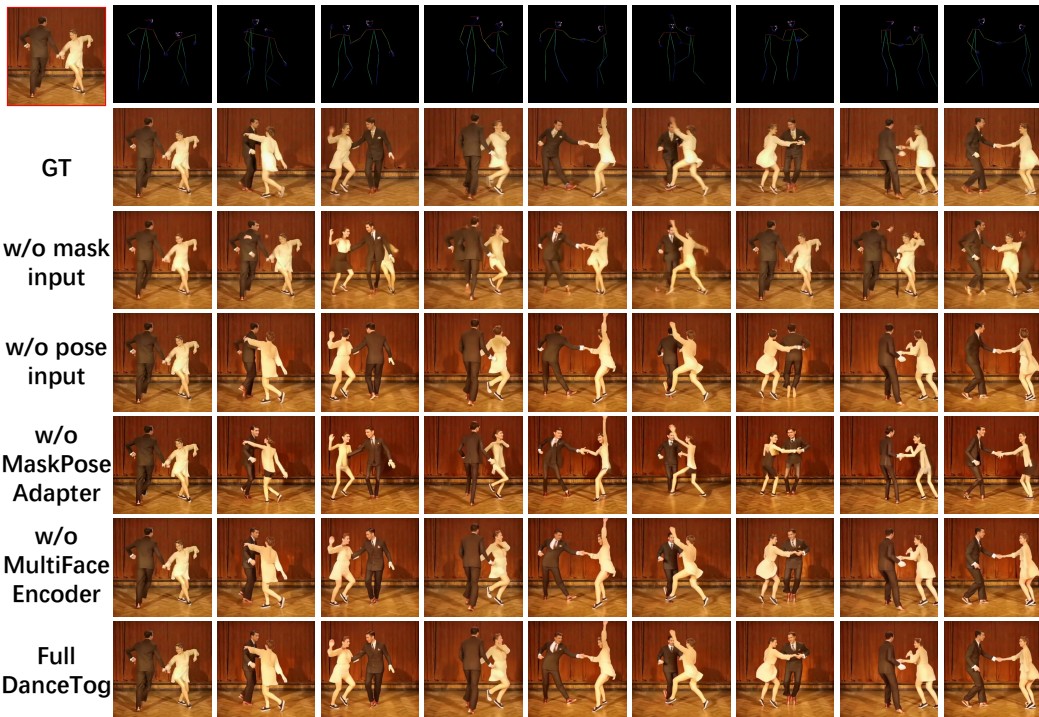

Figure 8: Ablation study animation results (2/2).

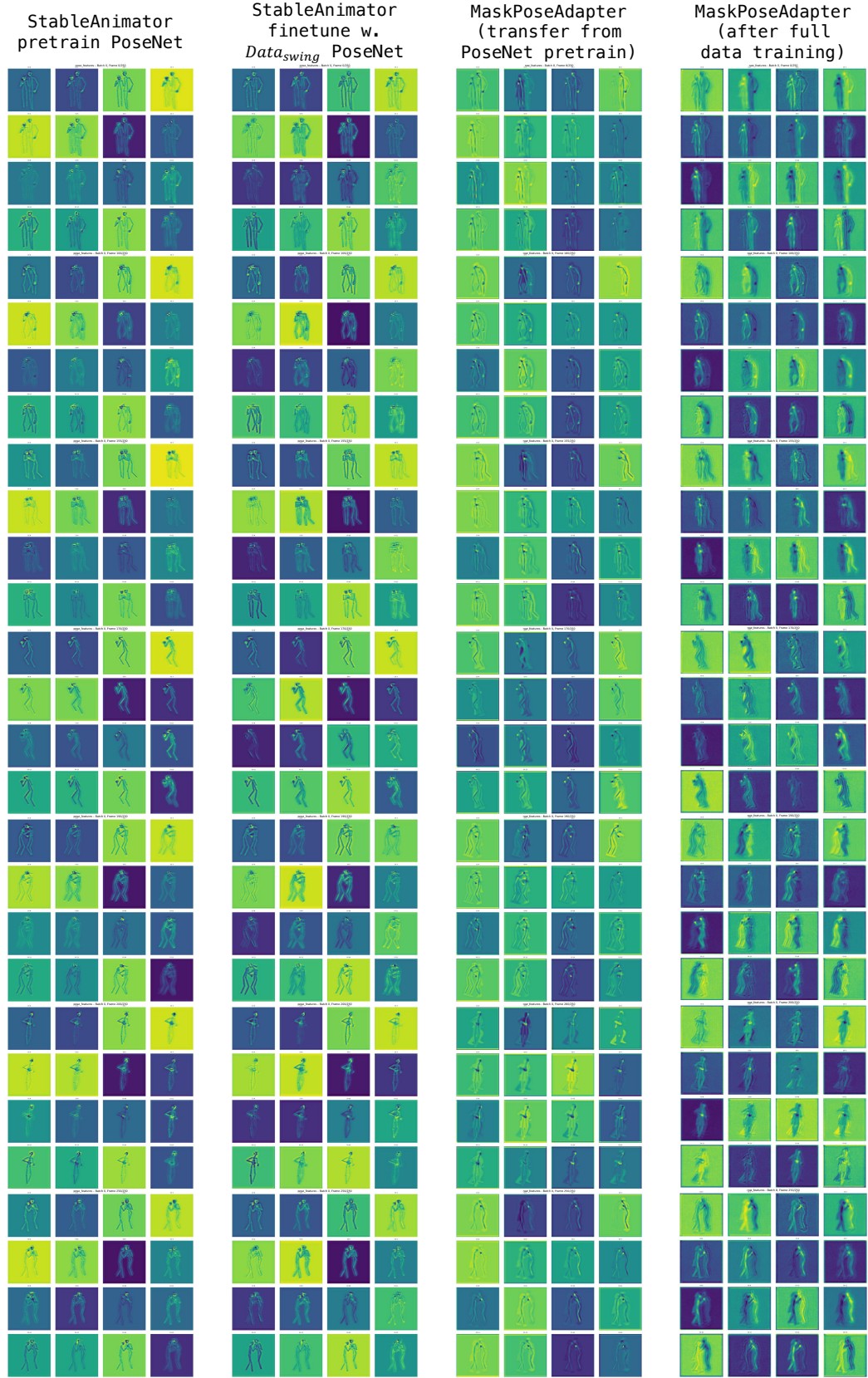

Figure 9: Comparison of PoseNet and MaskPoseAdapter outputs under identical frame inputs.

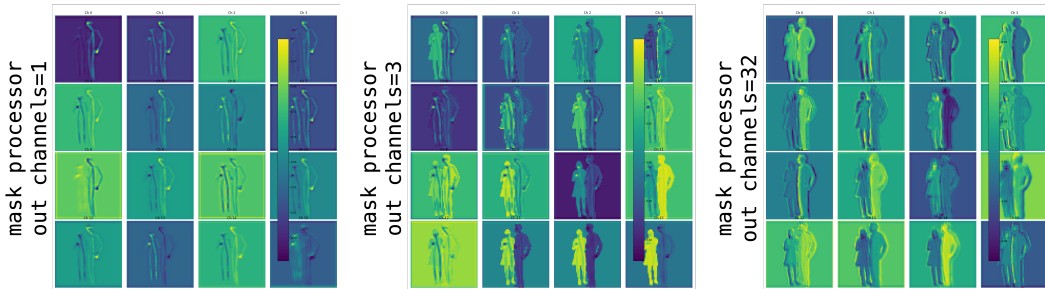

Figure 10: The effect of residual alpha on MaskPoseAdapter output.

Figure 11: The effect of different output channel numbers in the Light mask processor on MaskPoseAdapter output.

### B.4 Experiments on residual alpha and mask processor

Fig. 10 and Fig. 11 illustrate the influence of various Light mask processors and the parameter $\alpha_{\text{res}}$ on the feature maps generated by MaskPoseAdapter. Through extensive experimentation, we determined the optimal number of output channels for the Light mask processor and the value of $\alpha$ that effectively balances the pose and mask features in the output feature maps of MaskPoseAdapter. In practice, when training on the full dataset, we set $\alpha_{\text{res}} = 0.5$ and employ a Light mask processor with 3-channel output.

## C Details of Data Curation Pipeline

Due to the limitations of existing two-person interaction datasets (Maluleke et al., 2024; Sun et al., 2025), which fail to simultaneously provide identity diversity, static backgrounds, and fixed camera positions, we propose a novel data processing pipeline that recovers tracked human pose estimations from monocular RGB videos. Our pipeline extracts independent pose sequences, human silhouette masks, and facial masks for distinct individuals. We collected over 170 hours of paired figure skating videos from the internet and curated more than 26 hours of high-quality two-person figure skating segments, providing tracking masks, pose estimations, and facial masks for each individual subject ID. Additionally, we compiled a 1-hour humanoid robot dataset for fine-tuning our model to support controllable video generation tasks involving humanoid robots.

### C.1 Dataset Collection

We collected various single-person motion videos from existing research to enrich identity information, including TikTokDataset (Jafarian & Park, 2021), Champ (Zhu et al., 2024a), DisPose (Li et al., 2025a) and HumanVid (Wang et al., 2024e). Additionally, we gathered two-person interaction videos from existing research, including partner dancing, dual talking heads, and laboratory-recorded interactions from Swing Dance (Maluleke et al., 2024), Harmony4D (Khirodkar et al., 2024), HI4D (Yin et al., 2023), CHI3D (Fieraru et al., 2023), and Beyond Talking (Sun et al., 2025). While synthetic data has been used for video generation training in prior work (Yin et al., 2024; Wang et al., 2024e), our method focuses on controllable human interaction video generation in real-world scenarios, so we did not use any synthetic data during training.

### C.2 Human Tracking and Subject Selection

We first segment raw videos into scenes using TransNetV2 (Soucek & Lokoc, 2024) and detect humans using YOLOv8x (Jocher et al., 2023). For each person crop $\mathbf{p}_i^t$, we extract 512-dimensional identity features $\mathbf{f}_i^t$ using pre-trained OSNet (Zhou & Xiang, 2019; Zhou et al., 2019; 2021). Our enhanced tracking algorithm combines spatial proximity with ReID similarity to maintain consistent identities across frames. From all tracked identities, we select the two main subjects based on coverage (appearance frequency $\geq 40\%$), consistency, and quality score $Q_i = 0.7 \cdot \text{Coverage}_i + 0.3 \cdot \text{Consistency}_i$.

### C.3 Annotation Generation

Starting from the bounding boxes $\mathbf{b}_i^*$ of key frames, SAMURAI (Yang et al., 2024) bidirectionally propagates masks throughout the video sequence. We extract pose information of 133 keypoints using DWPose (Yang et al., 2023) and assign each pose to independent subject IDs via an IOU matching approach utilizing the masks generated by SAMURAI. MatAnyone (Yang et al., 2025a) produces high-quality alpha mattes from SAMURAI masks, providing data for tasks requiring background replacement (Sun et al., 2025). The complete data processing pipeline is illustrated in Fig. 3, where our Data Curation Pipeline generates four outputs from RGB video input: independent mask sequences, pose sequences, and facial mask sequences for each individual, as well as alpha mask sequences for all subjects.

## C.4 DATA FILTERING

Clips are automatically filtered based on: bbox overlap (max IoU $< 0.1$), size validation ($2\% <$ bbox area $< 80\%$ of frame), exact 2 primary subjects with $\geq 40\%$ coverage, and temporal consistency ($> 90\%$ successful tracking). For PairFS-4K, we additionally perform manual curation to ensure high-quality two-person interactions with clear visibility and balanced representation of skating movements.

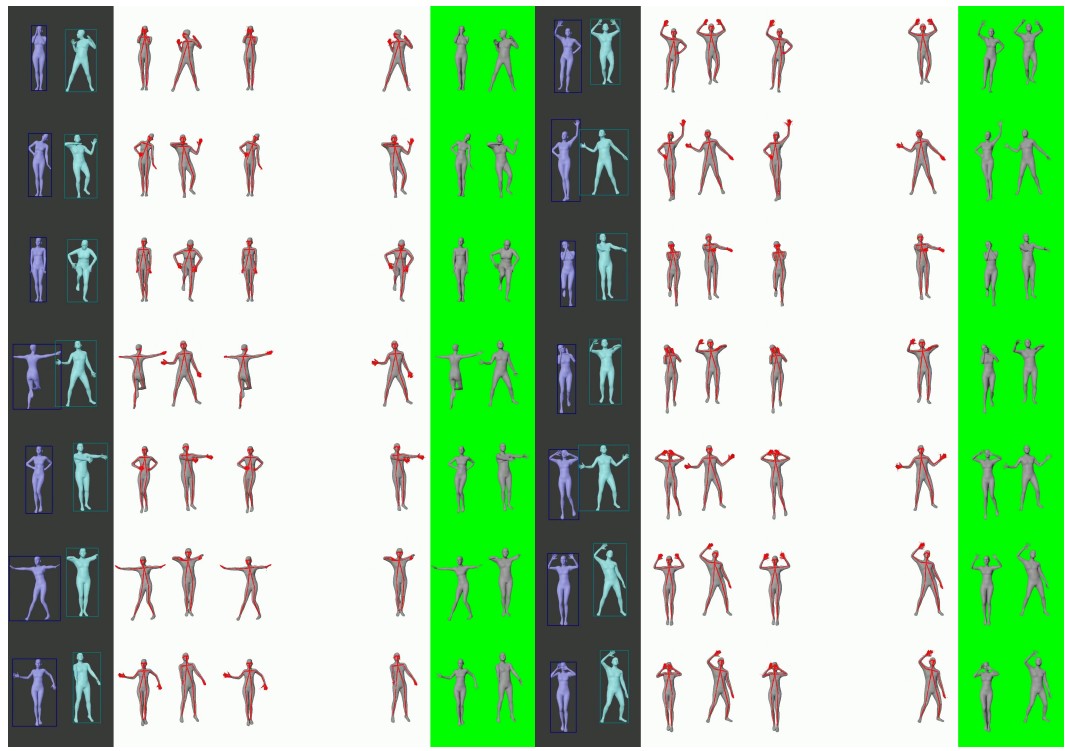

Figure 12: Data curation pipeline results (1/4).

## C.5 PAIRFS-4K DATASET PREPARATION PROCESS

We collected 932 figure skating videos from the internet, including numerous Olympic figure skating compilation videos with multiple shots. Using TransNetV2 (Soucek & Lokoc, 2024), we developed an automatic segmentation script and employed HumanReID and Yolox for identification and tracking of the main subjects. After manually filtering out segments that did not conform to single-person or pair figure skating criteria, we obtained **4.8K figure skating segments with a total duration of approximately 26 hours, and an average segment length of about 20 seconds**. We train our model on TikTokDataset (Jafarian & Park, 2021), Champ (Zhu et al., 2024a), DisPose (Li et al., 2025a), HumanVid (Wang et al., 2024e), Swing Dance (Maluleke et al., 2024), Harmony4D (Khirodkar et al., 2024), CHI3D (Fieraru et al., 2023), Beyond Talking (Sun et al., 2025), and **PairFS-4K**, using resolutions of $512 \times 512$. Due to the limited number of unique identities in HI4D, we exclude it from our training set. A detailed summary of all datasets is provided in Tab. 1. **PairFS-4K is the first two-person figure skating video dataset with over 7,000 unique identities**.

## D TOGETHERVIDEOBENCH BENCHMARK

### D.1 VIDEO GENERATION BENCHMARK OVERVIEW

There have been many benchmarks for evaluating large generative models (Liu et al., 2023b; Zhang et al., 2023a; Sun et al., 2024a; Liu et al., 2024b; Chen et al., 2025b; Collins et al., 2022; He et al.,

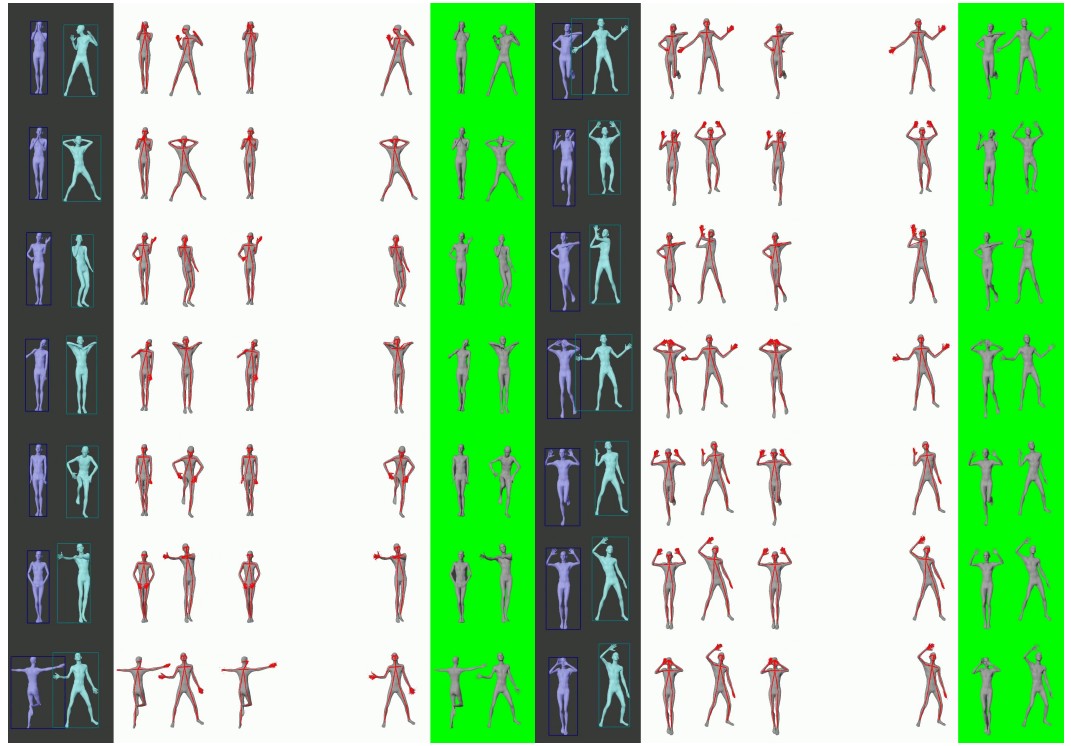

Figure 13: Data curation pipeline results (2/4).

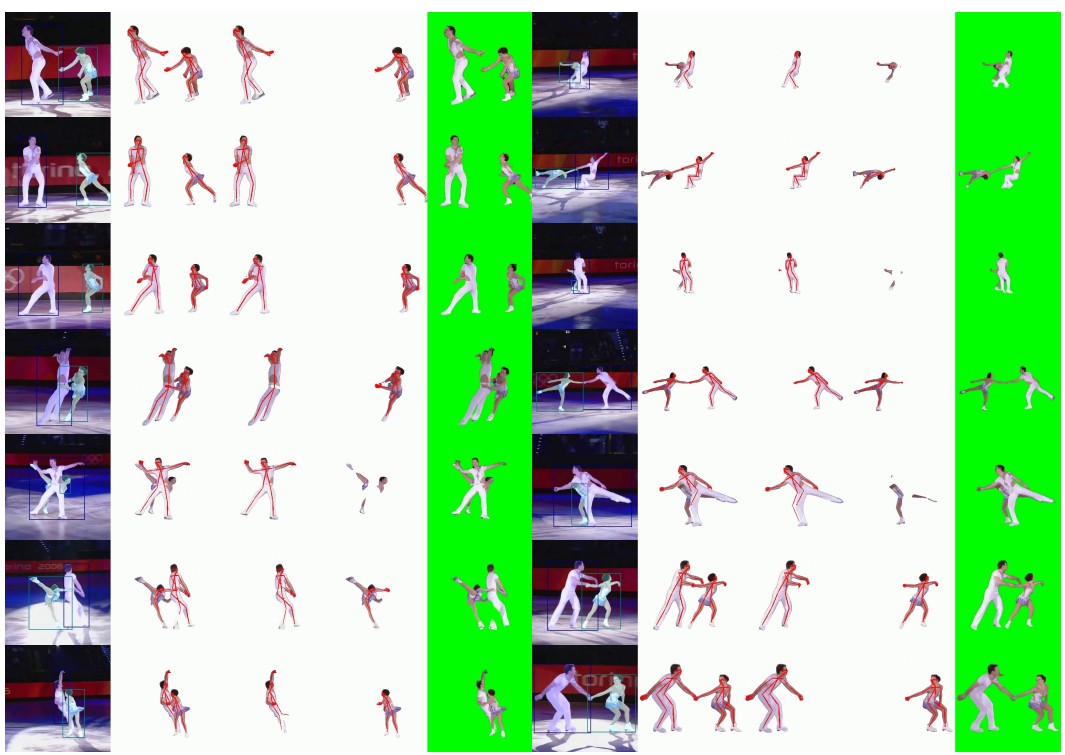

Figure 14: Data curation pipeline results (3/4).

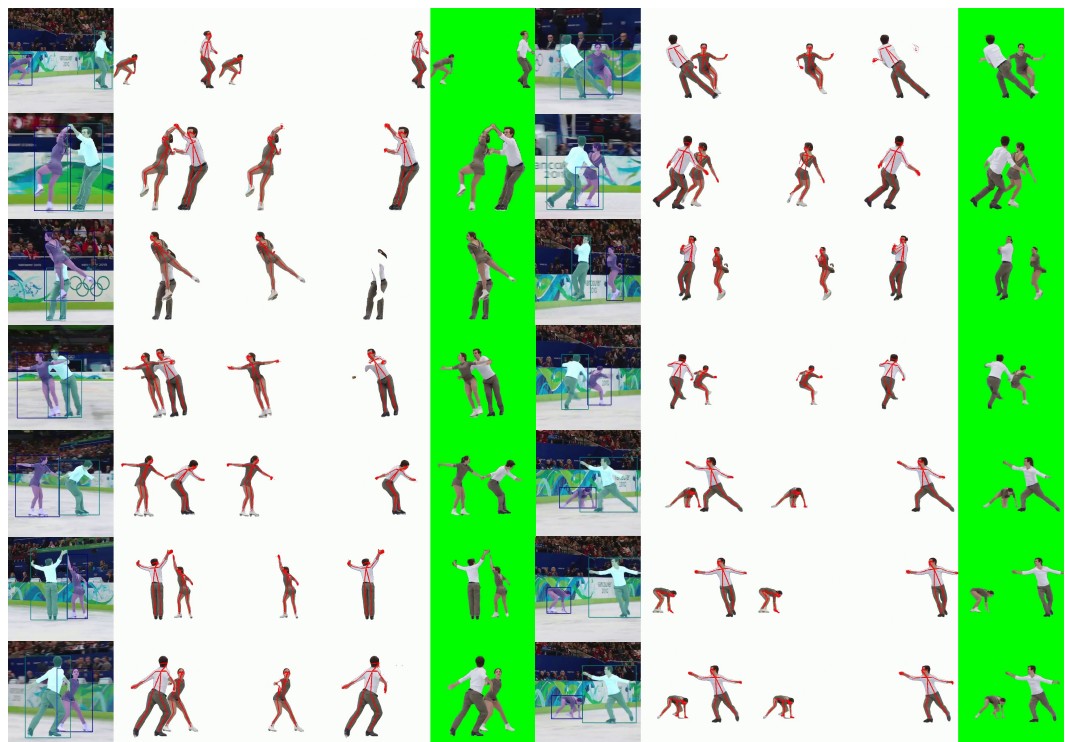

Figure 15: Data curation pipeline results (4/4).

2023; Sun et al., 2023; Zhu et al., 2023; Melistas et al., 2024; Huang et al., 2023; Guo et al., 2024; Chang et al., 2024). Recently, some video understanding methods have also been used to evaluate the quality of generated videos (Tang et al., 2025; Liao et al., 2024; Liu et al., 2024b; Ye et al., 2024; Madan et al., 2024). Despite this, the field of controllable video generation has lacked a reliable evaluation benchmark. Recent controllable video benchmarks (AIST++ (Li et al., 2021), TikTok-Eval (Jafarian & Park, 2021)) have mainly focused on single-person dance or static portrait animations, overlooking the three key challenges faced by realistic multi-person generation: multi-identity consistency (avoiding identity confusion in long sequences), interaction coherence (ensuring physically reasonable and temporally smooth interactions), and strict conditional fidelity (precisely following pose, mask, or text control inputs). To systematically evaluate these dimensions, we propose **TogetherVideoBench**, featuring three orthogonal tracks—*Identity-Consistency*, *Interaction-Coherence*, and *Video Quality*—supported by a unified, automated parsing pipeline that extracts per-person pose, mask, face-crop, and bounding-box representations for fair and reproducible assessment.

**Identity-Consistency**: To evaluate the ability of models to maintain consistent appearance and identity for each individual across long video sequences, we adopt standard multi-object tracking metrics, including HOTA (Luiten et al., 2021), MOTA (Bernardin & Stiefelhagen, 2008), and IDF1 (Ristani et al., 2016). These metrics comprehensively assess detection accuracy, association accuracy, and identity preservation, and are computed using the TrackEval toolkit (Luiten et al., 2021). This track is crucial for ensuring that generated videos do not suffer from identity switches or appearance confusion, especially in multi-person scenarios.

**Interaction-Coherence**: This track focuses on the temporal smoothness and physical plausibility of interactions between multiple humans, as well as the adherence to external control signals. We employ pose adherence (MPJPE-2D) (Cao et al., 2017), object keypoint similarity (OKS) (Lin et al., 2014), and the following metrics: pose structure similarity (PoseSSIM), motion smoothness (SmoothRMS), temporal dynamics error (TimeDynRMSE), and Fréchet Video Motion Distance (FVMD) (Liu et al., 2024a) to comprehensively evaluate the quality of human motion and interaction.

**Video Quality**: To assess the overall visual fidelity and semantic consistency of generated videos, we use a suite of widely adopted metrics, including SSIM (Wang et al., 2004), FVD (Unterthiner

et al., 2019), FID (Heusel et al., 2017), CLIP (Hessel et al., 2021), and the following metrics: LPIPS, L1, PSNR, DISTS (Ding et al., 2020), ST-SSIM (Moorthy & Bovik, 2010), GMSD-T (Yan et al., 2015). These metrics collectively measure both the perceptual quality and the alignment of generated content with the intended conditions. We calculate the metrics for both the overall frame and the human mask region of each frame separately, as shown in Fig. 16. Since the backgrounds of some evaluation data exhibit slight jittering, we believe that the quantitative evaluation of the human mask region is more indicative of human ID consistency and video quality in the generated videos than the quantitative evaluation of the full frame.

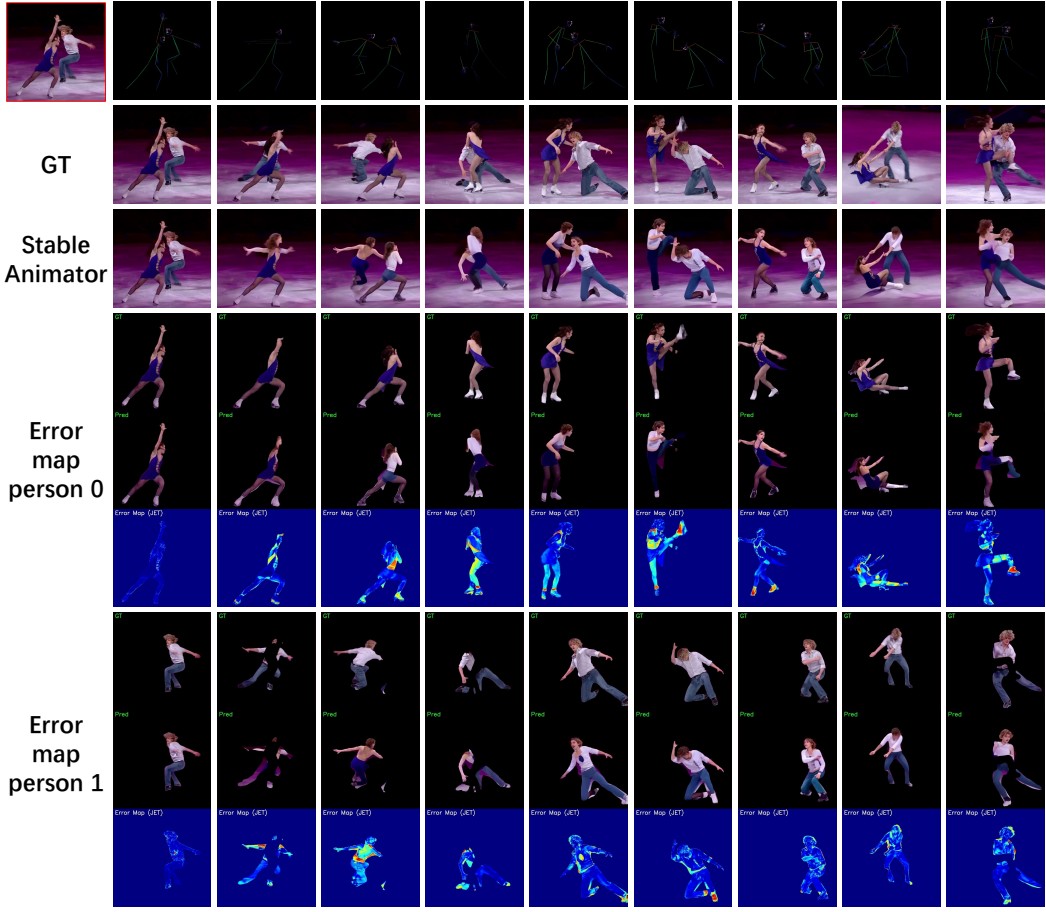

Figure 16: We use individual human masks for each person to conduct quantitative evaluation. The error map shown in the figure is the L1 Loss error map, which calculates the pixel-level absolute difference between the GT and predicted images.

All tracks share a unified Data Curation Pipeline that automatically extracts per-person pose, mask, face-crop, and bounding box for both ground truth and generated videos, ensuring reproducibility and fair comparison. For each video, we compute the relevant metrics for every individual and report the average across all videos in each group.

## D.2 EVALUATION DATASET

While laboratory-recorded datasets such as Harmony4D (Khirodkar et al., 2024), HI4D (Yin et al., 2023), and CHI3D (Fieraru et al., 2023) provide precise annotations, their videos are typically limited to 3–12 seconds, feature single scenes, and involve minimal position exchanges between subjects. As a result, they are insufficient for evaluating long-duration, multi-position, and realistic human interactions. To address this gap, we have manually curated and edited 100 high-quality two-person interaction videos from public competitions, films, documentaries, and social media, forming the core evaluation set of TogetherVideoBench. These videos encompass a wide range of real-world

interaction patterns, including exchange-intensive swing and Lindy-Hop routines, Latin ballroom duets, pair figure skating, boxing, wrestling and combat sequences, partner acrobatics and acro-yoga throws, everyday social gestures (such as handshakes and hugs), and two-person conversations. Each clip features exactly two performers, with nearly static cameras and backgrounds. Frequent occlusions, position exchanges, and physical contact between subjects introduce long-range motion, viewpoint changes, and identity-switching challenges—factors, making it a suitable testbed.

### D.3 METRICS

Below are the evaluation metrics and computation procedures used in the three tracks of TogetherVideoBench. To ensure reproducibility, both ground-truth and generated videos are first processed by our Data Curation Pipeline (Sec. C), which yields for each subject:

- **Pose sequences:** 133 keypoints per frame via DWPose (Yang et al., 2023).
- **Human masks:** per-frame human masks via SAMURAI (Yang et al., 2024).
- **Bounding boxes:** tight boxes around each human mask (for MOT eval (Luiten et al., 2021)).

#### D.3.1 TRACK 1 – IDENTITY-CONSISTENCY

- **IDF1 ↑:**
  After frame–level association with the Hungarian algorithm, let IDTP, IDFP and IDFN be identity–true positives, false positives and false negatives.

$$\text{IDF1} = \frac{2\,|\text{IDTP}|}{2\,|\text{IDTP}| + |\text{IDFP}| + |\text{IDFN}|}. \tag{23}$$

  It is the harmonic mean of identity precision and recall and therefore measures how often the *correct ID label* is maintained.

- **IDP / IDR ↑:**
  Precision and recall components of IDF1.

$$\text{IDP} = \frac{|\text{IDTP}|}{|\text{IDTP}| + |\text{IDFP}|}, \quad \text{IDR} = \frac{|\text{IDTP}|}{|\text{IDTP}| + |\text{IDFN}|}. \tag{24}$$

- **HOTA ↑:**
  Higher-Order Tracking Accuracy (Luiten et al., 2021) decomposes into $\text{DetA}$ (detection accuracy), $\text{AssA}$ (association accuracy) and $\text{LocA}$ (localisation accuracy):

$$\text{HOTA} = \sqrt{\text{DetA} \times \text{AssA}}, \quad \text{LocA} = 1 \; - \; \frac{1}{|\text{TP}|} \sum_{b \in \text{TP}} \big(1 - \text{IoU}(b)\big). \tag{25}$$

- **MOTA / MOTP ↑:**
  CLEAR-MOT summary:

$$\text{MOTA} = 1 \; - \; \frac{\text{FP} + \text{FN} + \text{IDSW}}{\text{GT dets}}, \quad \text{MOTP} = 1 \; - \; \frac{\sum_{\text{TP}}\big(1 - \text{IoU}\big)}{|\text{TP}|}. \tag{26}$$

- **IDSW ↓, FP ↓, FN ↓:**
  Absolute counts of identity switches, false positives and false negatives.

#### D.3.2 TRACK 2 – INTERACTION-COHERENCE

All keypoints are first temporally aligned and isotropically scale–shift aligned via a similarity transform.

- **MPJPE-2D ↓:**
  Let $\hat{\mathbf{x}}_{tpj}$ and $\mathbf{x}_{tpj}$ be the predicted and ground-truth pixel coordinates of joint $j$ of person $p$ at frame $t$, after SIM3 alignment; $T, P, J$ denote total frames, persons, and joints. Then

$$\text{MPJPE-2D} = \frac{1}{T\,P\,J} \sum_{t=1}^{T} \sum_{p=1}^{P} \sum_{j=1}^{J} \big\|\hat{\mathbf{x}}_{tpj} - \mathbf{x}_{tpj}\big\|_2. \tag{27}$$

- **OKS↑**:
  For each frame $t$, flatten over $P \times J$ valid keypoints. Let $d_k$ be the Euclidean error of the $k$th keypoint, $\sigma_k$ its COCO standard deviation, and $\mathcal{A}$ the estimated person area. Then

$$\text{OKS}_t = \frac{1}{K} \sum_{k=1}^{K} \exp\left(-\frac{d_k^2}{2\,\sigma_k^2 (\mathcal{A} + 10^{-6})}\right), \qquad \text{OKS} = \frac{1}{T} \sum_{t=1}^{T} \text{OKS}_t. \tag{28}$$

- **Pose-Heat SSIM↑**:
  Rasterise the set of keypoints at each frame into a Gaussian heatmap $H(\cdot)$ of size $H \times W$ with $\sigma = 4\,\text{px}$, then

$$\text{PoseHeatSSIM} = \frac{1}{T} \sum_{t=1}^{T} \text{SSIM}\big(H(\hat{\mathbf{X}}_t),\, H(\mathbf{X}_t)\big), \tag{29}$$

where $\hat{\mathbf{X}}_t, \mathbf{X}_t \in \mathbb{R}^{P \times J \times 2}$ are the keypoint arrays.

- **SmoothRMS↓**:
  Compute the third-order temporal derivative (jerk) of each trajectory, scaled by frame rate $f$:

$$\dddot{\mathbf{x}}_{tpj} = \frac{d^3}{dt^3} \mathbf{x}_{tpj} \times f^3. \tag{30}$$

Then

$$\text{SmoothRMS} = \sqrt{\frac{1}{T\,P\,J} \sum_{t=1}^{T} \sum_{p=1}^{P} \sum_{j=1}^{J} \big\| \dddot{\mathbf{x}}_{tpj} \big\|_2^2}. \tag{31}$$

- **Time-Dyn RMSE↓**:
  With the second-order derivative (acceleration)

$$\ddot{\mathbf{x}}_{tpj} = \frac{d^2}{dt^2} \mathbf{x}_{tpj} \times f^2, \tag{32}$$

define

$$\text{TimeDynRMSE} = \sqrt{\frac{1}{T\,P\,J} \sum_{t=1}^{T} \sum_{p=1}^{P} \sum_{j=1}^{J} \big\| \ddot{\mathbf{x}}_{tpj} \big\|_2^2}. \tag{33}$$

- **FVMD↓**:
  Model the velocity vectors of all keypoints as 2D Gaussians $\mathcal{N}(\mu_p, \Sigma_p)$ for prediction and $\mathcal{N}(\mu_g, \Sigma_g)$ for ground truth, where $\mu = \mathbb{E}[\dot{\mathbf{x}}]$ and $\Sigma = \text{Cov}[\dot{\mathbf{x}}]$. Then

$$\text{FVMD} = \big\| \mu_p - \mu_g \big\|_2^2 \;+\; \text{Tr}\big(\Sigma_p + \Sigma_g - 2\,(\Sigma_p\,\Sigma_g)^{\frac{1}{2}}\big). \tag{34}$$

### D.3.3  TRACK 3 – VIDEO QUALITY

- **L1↓**:
  Let $I_t(x,y,c)$ and $\hat{I}_t(x,y,c)$ be the ground-truth and predicted RGB pixel values at frame $t$, spatial location $(x,y)$ and channel $c$, over $T$ frames of size $H \times W$ and $C = 3$ channels. Then

$$\text{L1} = \frac{1}{T\,H\,W\,C} \sum_{t=1}^{T} \sum_{x=1}^{W} \sum_{y=1}^{H} \sum_{c=1}^{C} \big|I_t(x,y,c) - \hat{I}_t(x,y,c)\big|. \tag{35}$$

- **PSNR↑**:
  Compute the per-frame mean squared error

$$\text{MSE} = \frac{1}{H\,W\,C} \sum_{x=1}^{W} \sum_{y=1}^{H} \sum_{c=1}^{C} \big(I_t(x,y,c) - \hat{I}_t(x,y,c)\big)^2, \tag{36}$$

then

$$\text{PSNR} = 20 \log_{10}\Big(\frac{255}{\sqrt{\text{MSE}}}\Big). \tag{37}$$

- **SSIM ↑:**
  For each frame $t$ and each channel $c$, compute

$$\text{SSIM}_t^c = \text{SSIM}\big(I_t(\cdot,\cdot,c),\ \hat{I}_t(\cdot,\cdot,c)\big), \tag{38}$$

  then average:

$$\text{SSIM} = \frac{1}{T\,C}\sum_{t=1}^{T}\sum_{c=1}^{C}\text{SSIM}_t^c. \tag{39}$$

- **LPIPS ↓:**
  On a $256 \times 256$ crop, let $\phi_\ell(\cdot)$ be the $\ell$-th layer feature map and $w_\ell$ learned weights. Then

$$\text{LPIPS} = \frac{1}{L}\sum_{\ell=1}^{L}\frac{1}{H_\ell W_\ell}\big\| w_\ell \odot \big(\phi_\ell(I) - \phi_\ell(\hat{I})\big)\big\|_1. \tag{40}$$

- **DISTS ↓:**
  Let $f_\ell(\cdot)$ be VGG16 feature maps, $\tilde{f}_\ell$ their normalized versions, and $G(\cdot)$ the Gram matrix. Define

$$\text{structure}_\ell = \frac{\langle \tilde{f}_\ell(I),\ \tilde{f}_\ell(\hat{I})\rangle}{\|\tilde{f}_\ell(I)\|\,\|\tilde{f}_\ell(\hat{I})\|}, \quad \text{texture}_\ell = \text{MSE}\big(G(f_\ell(I)),\ G(f_\ell(\hat{I}))\big). \tag{41}$$

  Then

$$\text{DISTS} = \frac{1}{L}\sum_{\ell=1}^{L}\Big(0.5\,\text{structure}_\ell + 0.5\,\big(1 - \text{texture}_\ell\big)\Big). \tag{42}$$

- **CLIPScore ↑:**
  We encode each frame from the ground-truth and generated videos into CLIP image embeddings $v_t, \hat{v}_t \in \mathbb{R}^d$, normalize them to unit vectors, and compute the frame-wise cosine similarity:

$$s_t = \frac{v_t^\top \hat{v}_t}{\|v_t\| \cdot \|\hat{v}_t\|}. \tag{43}$$

  The final CLIPScore is obtained by averaging over all $T$ frames:

$$\text{CLIPScore} = \frac{1}{T}\sum_{t=1}^{T} s_t. \tag{44}$$

- **ST-SSIM ↑:**
  With window length $w = 3$, define for each spatio-temporal block

$$\text{SSIM}_{\text{3D}} = \text{SSIM}\big(I_{t:t+w-1},\ \hat{I}_{t:t+w-1}\big), \tag{45}$$

  then

$$\text{ST-SSIM} = \frac{1}{T-w+1}\sum_{t=1}^{T-w+1}\text{SSIM}_{\text{3D}}. \tag{46}$$

- **GMSD-Temporal ↓:**
  For each $t = 2,\ldots,T$, let

$$g_t(x,y) = \big\|\nabla I_t(x,y)\big\|_2, \quad \hat{g}_t(x,y) = \big\|\nabla \hat{I}_t(x,y)\big\|_2, \tag{47}$$

  and

$$\text{GMS}_t(x,y) = \frac{2\,g_t\,\hat{g}_t + \varepsilon}{g_t^2 + \hat{g}_t^2 + \varepsilon}. \tag{48}$$

  Then

$$\text{GMSD-Temporal} = \sqrt{\frac{1}{(T-1)\,H\,W}\sum_{t=2}^{T}\text{Var}_{x,y}\big(\text{GMS}_t(x,y)\big)}. \tag{49}$$

- **FVD** ↓:
  Extract I3D features for each non-overlapping 16-frame clip, compute means $\mu_r, \mu_f$ and covariances $\Sigma_r, \Sigma_f$, then

$$\text{FVD} = \left\| \mu_r - \mu_f \right\|_2^2 + \text{Tr}\left( \Sigma_r + \Sigma_f - 2(\Sigma_r \Sigma_f)^{1/2} \right). \tag{50}$$

- **FID** ↓:
  On all frames, extract Inception-V3 features, form $(\mu_r, \Sigma_r)$ and $(\mu_f, \Sigma_f)$, and use

$$\text{FID} = \left\| \mu_r - \mu_f \right\|_2^2 + \text{Tr}\left( \Sigma_r + \Sigma_f - 2(\Sigma_r \Sigma_f)^{1/2} \right). \tag{51}$$

- **CLIP-FID** ↓:
  Identical to FID but using CLIP embeddings instead of Inception features:

$$\text{CLIP} - \text{FID} = \left\| \mu_r - \mu_f \right\|_2^2 + \text{Tr}\left( \Sigma_r + \Sigma_f - 2(\Sigma_r \Sigma_f)^{1/2} \right). \tag{52}$$

All Track-3 metrics are reported both on the *full frame* and on each human mask (per-person); the final masked score is the arithmetic mean over the two performers.

## E  USER STUDY

To assess perceptual quality, we conducted a user study involving 50 participants (25 general users and 25 domain experts). Each participant viewed 20 video pairs in DanceTogEval-100 dataset (ground-truth vs. generated) and rated the generated video using a 5-point Likert scale across three dimensions. The evaluation criteria were defined as follows:

**Track 1: Identity-Consistency**  *"To what extent are the identity characteristics (face, clothing, body shape) of the individuals consistently preserved throughout the generated video?"*

- **5**: Identity is perfectly consistent at all times; individuals remain clearly distinguishable even after position exchanges.
- **4**: Identity is mostly consistent, with only minor and infrequent confusion.
- **3**: Moderate consistency; noticeable identity confusion or drift occurs.
- **2**: Frequent confusion; the two individuals are often hard to distinguish.
- **1**: Identity consistency completely fails; individuals are indistinguishable.

**Track 2: Interaction-Coherence**  *"How natural and temporally coherent are the motions and interactions (e.g., holding hands, lifts, synchronized movements) between the individuals in the generated video?"*

- **5**: Motions are highly natural and fluid; interactions are physically plausible and temporally smooth.
- **4**: Motions are generally natural; interactions are mostly reasonable with occasional minor glitches.
- **3**: Motions are acceptable but somewhat stiff; interactions appear unnatural at times.
- **2**: Motions are clearly unnatural; interactions are implausible and exhibit abrupt changes.
- **1**: Motions are extremely rigid; interactions are completely unrealistic.

**Track 3: Video Quality**  *"What is the overall visual quality of the generated video? (e.g., sharpness, detail preservation, presence of artifacts)"*

- **5**: Visually sharp with rich details and almost no artifacts; quality approaches real footage.
- **4**: Good quality with minor artifacts or slight blur in some frames.
- **3**: Acceptable quality, but noticeable artifacts or distortions are present.
- **2**: Poor quality with frequent artifacts, blurriness, or unnatural textures.
- **1**: Very poor quality with severe artifacts and obvious AI-generated appearance.

E.1 RESULTS AND ANALYSIS

The average scores (out of 5) across 1,000 ratings per track are summarized in Tab. 7. Our full model achieves the highest scores across all tracks, with an average of **4.61/5.0**, significantly outperforming all baselines.

Table 7: User study results (mean scores out of 5). Higher is better.

| Method | Identity Consistency | Interaction Coherence | Video Quality | Average |
|---|---|---|---|---|
| Animate Anyone | 2.31 | 2.47 | 2.68 | 2.49 |
| Champ | 1.89 | 2.21 | 2.34 | 2.15 |
| MimicMotion | 1.92 | 2.18 | 2.29 | 2.13 |
| HumanVid | 3.14 | 3.29 | 3.42 | 3.28 |
| UniAnimate | 2.78 | 2.91 | 3.05 | 2.91 |
| UniAnimate-DiT | 2.42 | 2.56 | 2.71 | 2.56 |
| DisPose | 1.95 | 2.23 | 2.37 | 2.18 |
| StableAnimator | 3.38 | 3.51 | 3.45 | 3.45 |
| StableAnimator w. $Data_{swing}$ | 3.89 | 4.02 | 4.11 | 4.01 |
| **DanceTog w.** $Data_{swing}$ | 4.28 | 4.35 | 4.19 | 4.27 |
| **DanceTog w.** $Data_{full}$ | 4.47 | 4.52 | 4.38 | 4.46 |
| **DanceTog w.** $Data_{full} + Data_{PairFS}$ | **4.61** | **4.68** | **4.53** | **4.61** |

**Analysis of Human Preference vs. Artifacts.** Although quantitative metrics are strong, we acknowledge the visual challenges inherent in this task. The qualitative superiority of **DanceTog** is further evidenced by the user preference margin: specifically in *Identity-Consistency*, users preferred our method over the second-best baseline (StableAnimator) by a margin of approximately **36.4%** (4.61 vs 3.38).

While some generated samples may exhibit minor boundary artifacts or a synthetic appearance, we attribute this primarily to two factors:

(i) **Extreme Task Difficulty:** The task involves complex multi-person interactions (e.g., occlusion, contact, position swapping) where baselines often suffer catastrophic failures (as shown in Fig. 18-Fig. 29). In contrast, our model maintains structural integrity during these critical moments.

(ii) **Data Pre-processing Artifacts:** To train on diverse motion backgrounds, we utilized a background replacement strategy based on extracted human masks. Occasional failures in the segmentation model (e.g., MatAnyOne) can introduce boundary noise in the training data, which propagates to the generated results.

Despite these isolated artifacts, the user study conclusively demonstrates that human evaluators prioritize the **semantic correctness** (identity preservation and interaction logic) of our method over the pixel-perfect but identity-drifting results of baseline methods.

## F ADDITIONAL QUANTITATIVE RESULTS

All Track3 metrics are shown in Tab.8 and Tab.9. The experimental results prove that DanceTog achieves better performance than StableAnimator when trained on the identical Swing Dance dataset.

## G INFERENCE AND TRAINING EFFICIENCY

As shown in Tab. 10, DanceTog achieves competitive inference speed (0.88 fps) while maintaining superior generation quality. Our method is significantly faster than most recent baselines such as UniAnimate (0.25 fps) and MimicMotion (0.06 fps), and comparable to lightweight methods like Animate Anyone (0.97 fps) and StableAnimator (0.89 fps). Note that, like all human animation

Table 8: Comparison of models using **Full Frame** evaluation metrics.

| Method | L1↓ | PSNR↑ | SSIM↑ | LPIPS↓ | DISTS↓ | CLIP↑ | ST-SSIM↑ | GMSD-T↓ | FVD↓ | FID↓ | C-FID↓ |
|---|---|---|---|---|---|---|---|---|---|---|---|
| AnimateAnyone | 37.32 | 13.23 | 0.49 | 0.56 | 0.27 | 0.91 | 0.54 | 0.42 | 108.2 | 118.1 | 27.7 |
| Champ | 43.70 | 11.93 | 0.49 | 0.56 | 0.29 | 0.91 | 0.39 | 0.36 | 125.7 | 114.6 | 25.6 |
| MimicMotion | 52.08 | 11.04 | 0.47 | 0.58 | 0.32 | 0.91 | 0.37 | 0.39 | 121.0 | 116.6 | 26.2 |
| HumanVid | 38.93 | 13.67 | 0.52 | 0.50 | 0.26 | 0.93 | 0.53 | 0.35 | 97.2 | 90.2 | 18.6 |
| UniAnimate | 37.95 | 13.62 | 0.55 | 0.53 | 0.29 | 0.89 | 0.61 | 0.42 | 132.0 | 151.2 | 42.8 |
| UniAnimate-DiT | 43.11 | 12.34 | 0.50 | 0.53 | 0.28 | 0.92 | 0.45 | 0.42 | 111.9 | 100.3 | 20.8 |
| DisPose | 42.52 | 12.28 | 0.54 | 0.54 | 0.31 | 0.91 | 0.41 | 0.39 | 127.4 | 127.9 | 31.0 |
| StableAnimator | 33.44 | 14.60 | 0.57 | 0.44 | 0.24 | 0.94 | 0.66 | 0.40 | 85.7 | 84.1 | 18.1 |
| StableAnimator w. $Data_{swing}$ | 30.31 | 15.27 | 0.60 | 0.42 | 0.22 | 0.94 | 0.69 | 0.42 | 78.8 | 79.3 | 16.1 |
| **DanceTog w.** $Data_{swing}$ | 32.62 | 15.12 | 0.59 | 0.44 | 0.23 | 0.94 | 0.68 | 0.38 | 79.3 | 82.1 | 14.7 |
| **DanceTog w.** $Data_{full}$ | 29.94 | 15.81 | 0.61 | 0.42 | 0.22 | 0.95 | 0.70 | 0.39 | 76.9 | 77.6 | 13.1 |
| **DanceTog w.** $Data_{full} + Data_{PairFS}$ | 29.52 | 15.85 | 0.61 | 0.42 | 0.22 | 0.95 | 0.70 | 0.39 | 76.3 | 75.1 | 12.6 |

Table 9: Comparison of models using **Human Masked Region** evaluation metrics.

| Method | L1↓ | PSNR↑ | SSIM↑ | LPIPS↓ | DISTS↓ | CLIP↑ | ST-SSIM↑ | GMSD-T↓ | FVD↓ | FID↓ | C-FID↓ |
|---|---|---|---|---|---|---|---|---|---|---|---|
| AnimateAnyone | 59.92 | 10.45 | 0.92 | 0.06 | 0.13 | 0.92 | 0.70 | 0.18 | 44.8 | 101.4 | 19.2 |
| Champ | 83.35 | 8.36 | 0.92 | 0.07 | 0.17 | 0.90 | 0.58 | 0.17 | 69.2 | 178.7 | 34.2 |
| MimicMotion | 77.02 | 8.75 | 0.92 | 0.07 | 0.16 | 0.90 | 0.56 | 0.17 | 65.5 | 180.9 | 33.9 |
| HumanVid | 47.97 | 12.13 | 0.93 | 0.05 | 0.12 | 0.93 | 0.76 | 0.15 | 34.9 | 72.4 | 14.2 |
| UniAnimate | 56.34 | 11.05 | 0.92 | 0.06 | 0.13 | 0.92 | 0.70 | 0.17 | 45.0 | 109.8 | 21.4 |
| UniAnimate-DiT | 64.48 | 9.89 | 0.91 | 0.06 | 0.14 | 0.90 | 0.68 | 0.18 | 51.4 | 119.6 | 21.5 |
| DisPose | 76.75 | 8.93 | 0.92 | 0.07 | 0.16 | 0.90 | 0.60 | 0.17 | 64.7 | 196.0 | 36.4 |
| StableAnimator | 48.51 | 12.00 | 0.93 | 0.05 | 0.12 | 0.93 | 0.75 | 0.15 | 38.4 | 71.8 | 15.7 |
| StableAnimator w. $Data_{swing}$ | 41.41 | 13.06 | 0.93 | 0.04 | 0.11 | 0.94 | 0.80 | 0.14 | 29.0 | 66.7 | 12.5 |
| **DanceTog w.** $Data_{swing}$ | 34.49 | 14.76 | 0.94 | 0.03 | 0.09 | 0.94 | 0.85 | 0.14 | 21.5 | 57.5 | 9.5 |
| **DanceTog w.** $Data_{full}$ | 32.80 | 15.15 | 0.94 | 0.03 | 0.09 | 0.94 | 0.85 | 0.14 | 20.6 | 56.1 | 8.9 |
| **DanceTog w.** $Data_{full} + Data_{PairFS}$ | 30.14 | 15.82 | 0.94 | 0.03 | 0.08 | 0.95 | 0.87 | 0.14 | 17.1 | 48.0 | 7.9 |

methods, DanceTog is not a real-time approach, but provides a practical balance between quality and efficiency for controllable multi-person video generation.

## H APPLICATIONS: HUMAN–ROBOT INTERACTION VIDEO GENERATION

After fine-tuning on our HumanRob-300 humanoid-robot video dataset, *DanceTogether* can generate realistic interaction videos between a humanoid robot and a human (see Fig. 17). This demonstrates the effectiveness and generalization ability of *DanceTogether*, offering new insights for embodied-AI and human-robot interaction research. After the robot and the human exchange positions, both agents retain their original identities. The method also handles fine-grained interactions, such as handshakes and sparring, remarkably well. This part of the video results can be found on the supplementary webpage.

## I ADDITIONAL QUALITATIVE RESULTS

Fig. 1, Fig. 18 and Fig. 19 showcase the generalization capability of DanceTog. With the same driving video and different reference images, DanceTog generates diverse outputs. Fig. 20, Fig. 21, Fig. 22, and Fig. 23 present qualitative comparisons across consecutive frames for different cases. The top row in each figure shows the input reference image and the corresponding pose sequence. The pose sequence is estimated from a ground truth video, and the first frame is used as the reference image input for each baseline. Our proposed DanceTog method consistently outperforms all baselines in generating video frames with rich interaction details. Notably, it preserves individual identity features even when the two subjects exchange positions. For qualitative video comparisons, please refer to the supplementary webpage.

Figs. 24–29 show qualitative comparisons of all baselines. We extracted consecutive frames where position swapping occurs. The leftmost column is the GT video. We used the first frame of the GT video as the reference image (not the first frame shown in the figures), and the dwpose results estimated from the GT video as the pose condition input for each baseline (corresponding to the GT images in the first column). Due to file size limitations, the images below are compressed. Please refer to the webpage in the supplementary materials for the original videos.

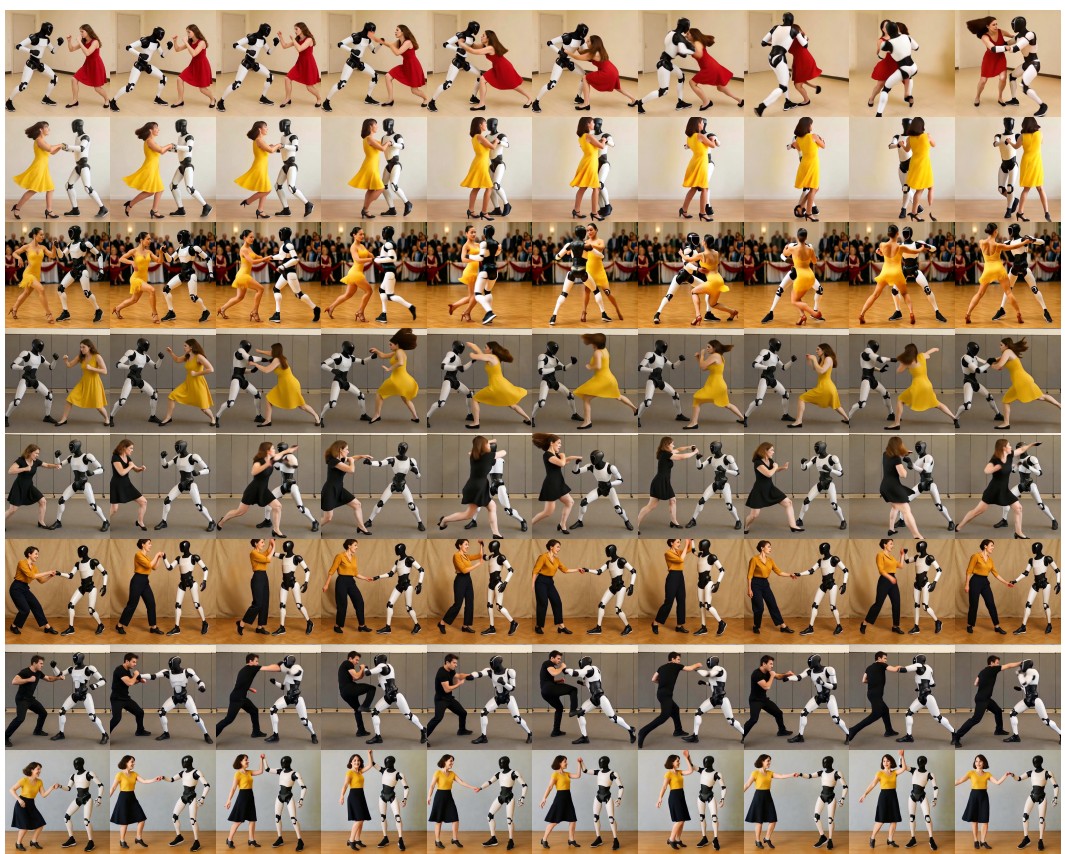

Figure 17: Using the first frame as the reference image, we perform inference on human–robot interaction sequences conditioned on independent pose maps and human masks.

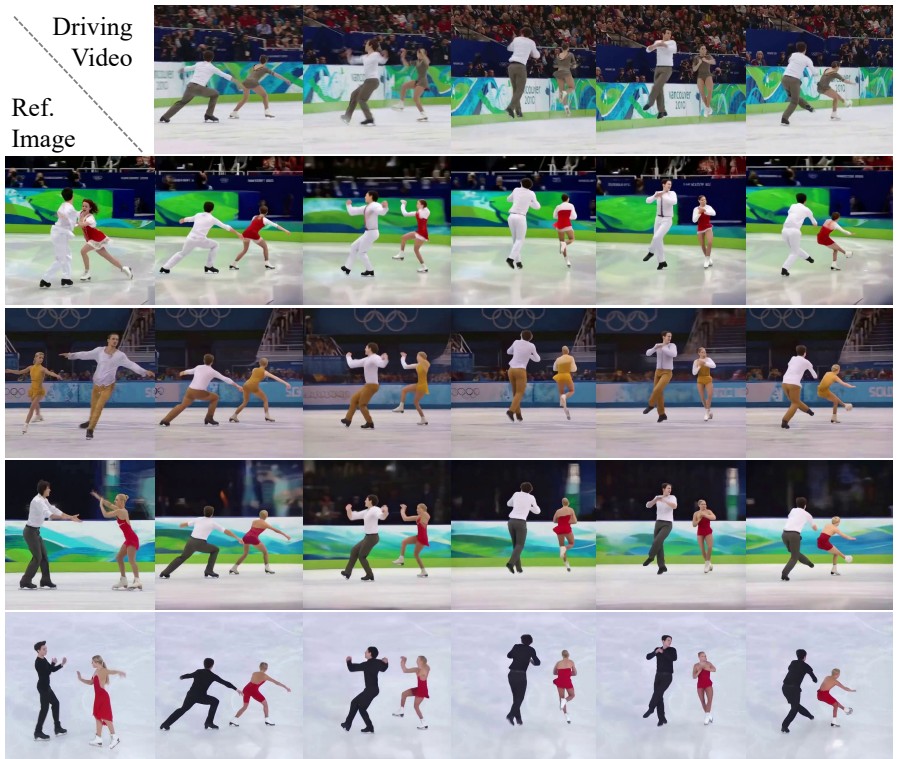

Figure 18: Additional motion transfer results (1/2).

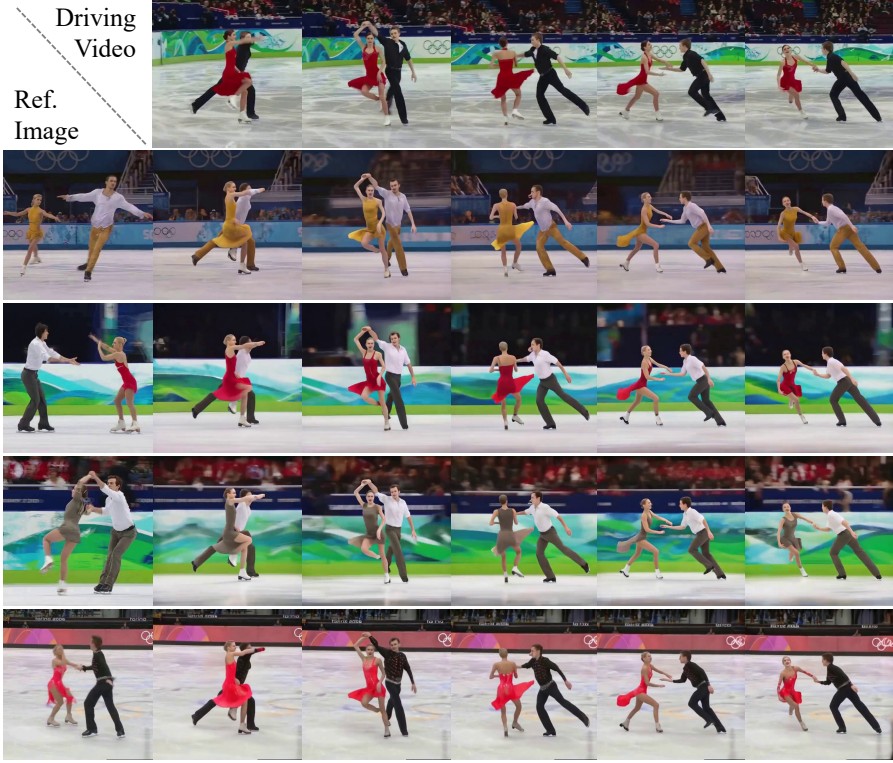

Figure 19: Additional motion transfer results (2/2).

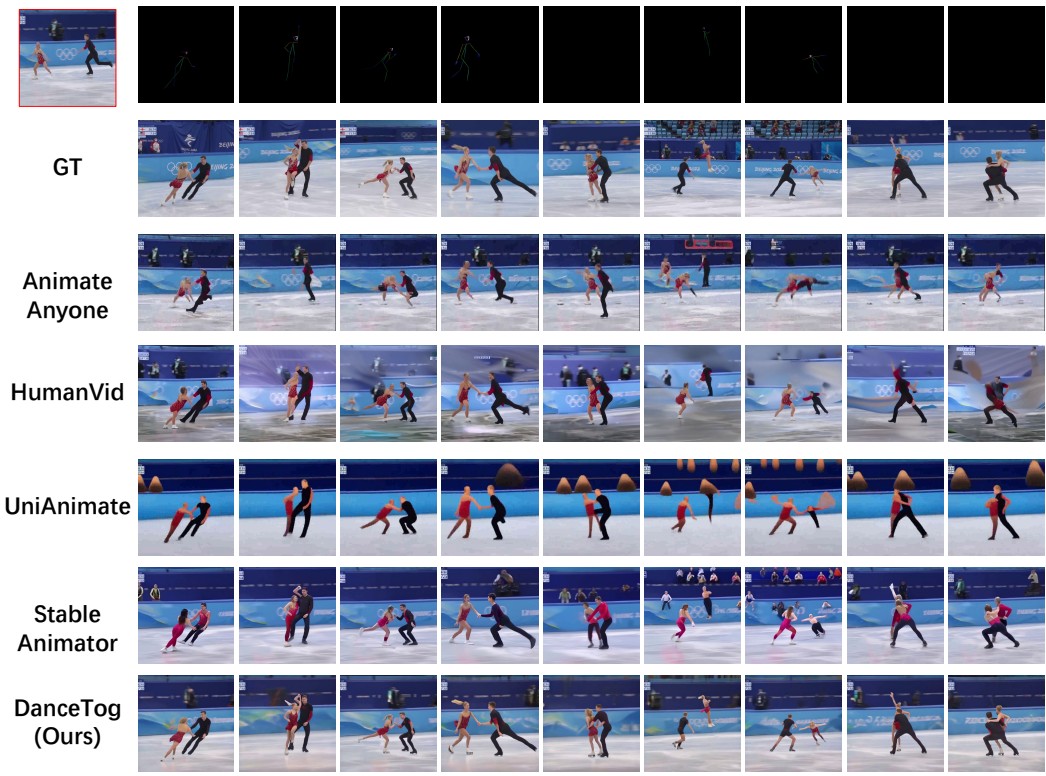

Figure 20: Additional animation results (1/4). The image with red borders is the reference images.

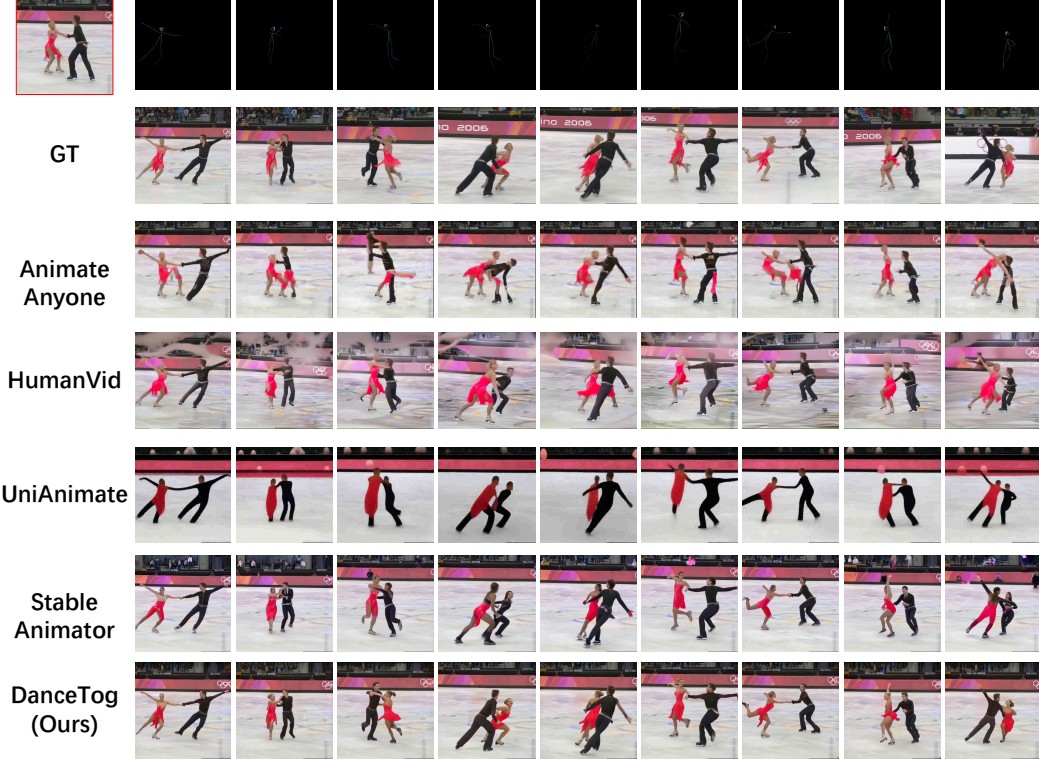

Figure 21: Additional animation results (2/4). The image with red borders is the reference images.

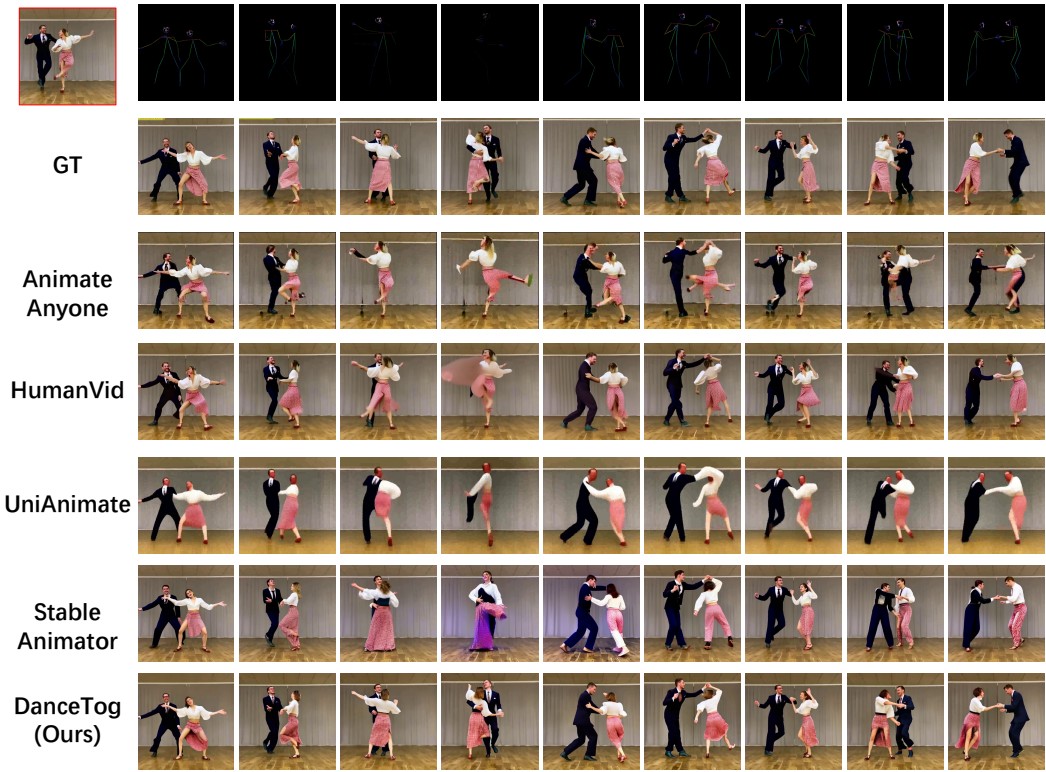

Figure 22: Additional animation results (3/4). The image with red borders is the reference images.

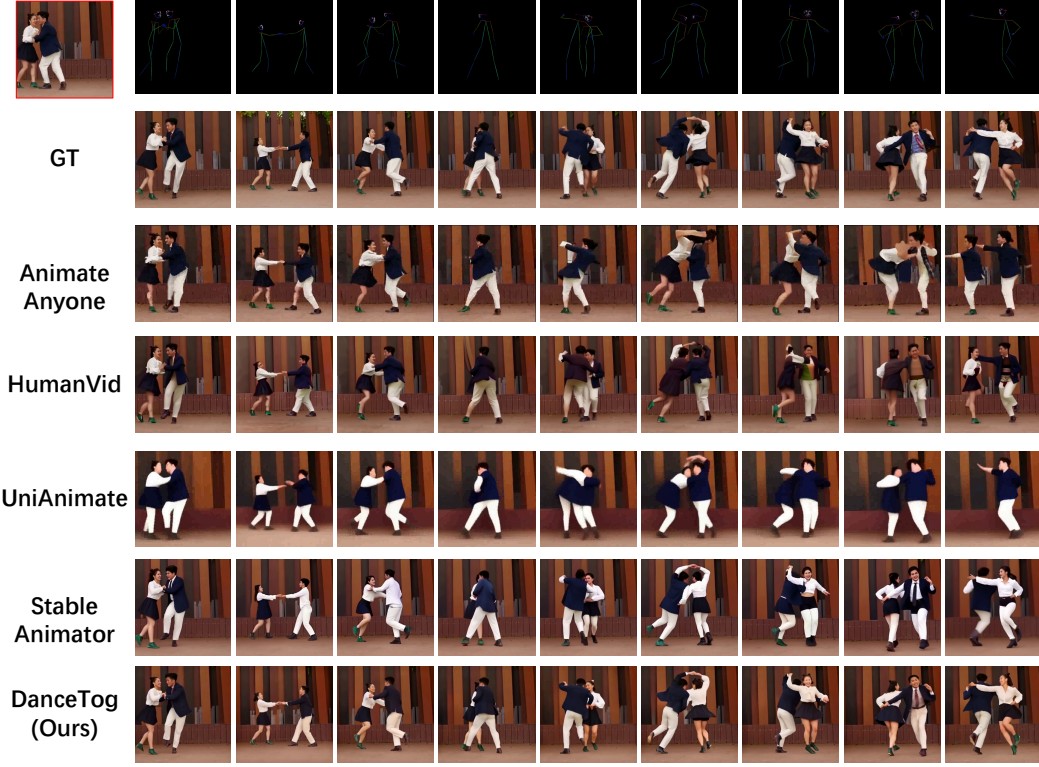

Figure 23: Additional animation results (4/4). The image with red borders is the reference images.

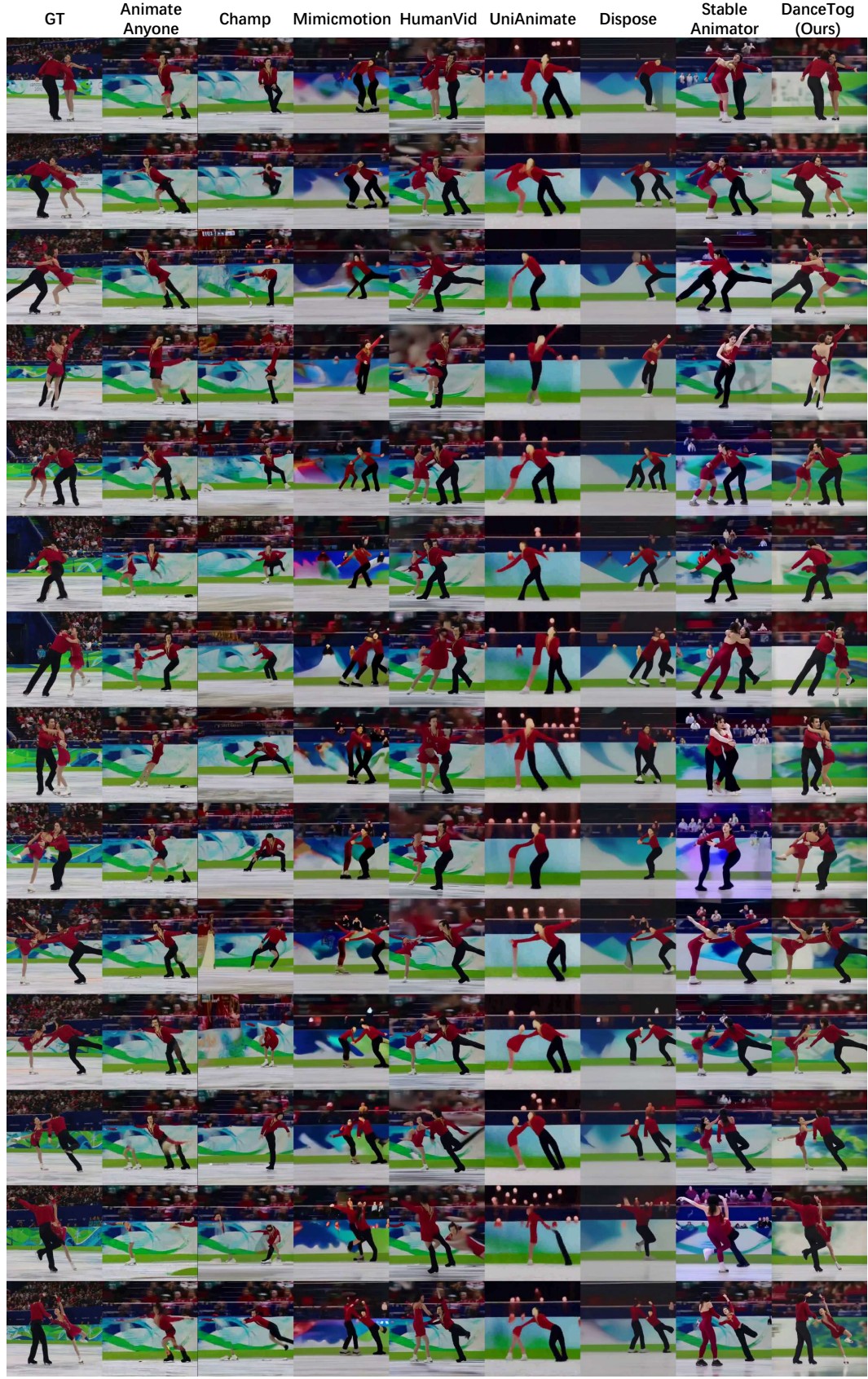

Figure 24: Additional animation results (1/6).

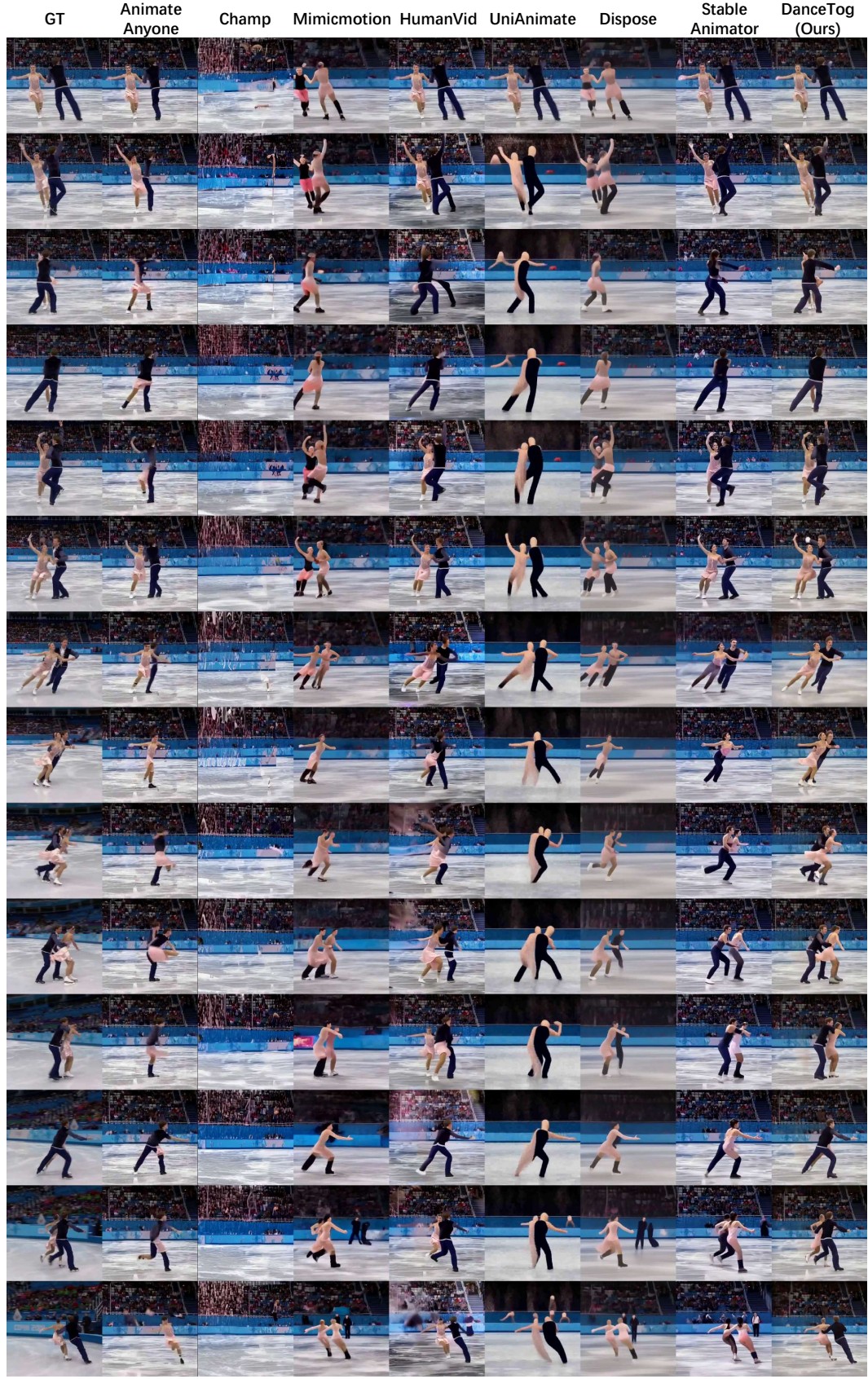

Figure 25: Additional animation results (2/6).

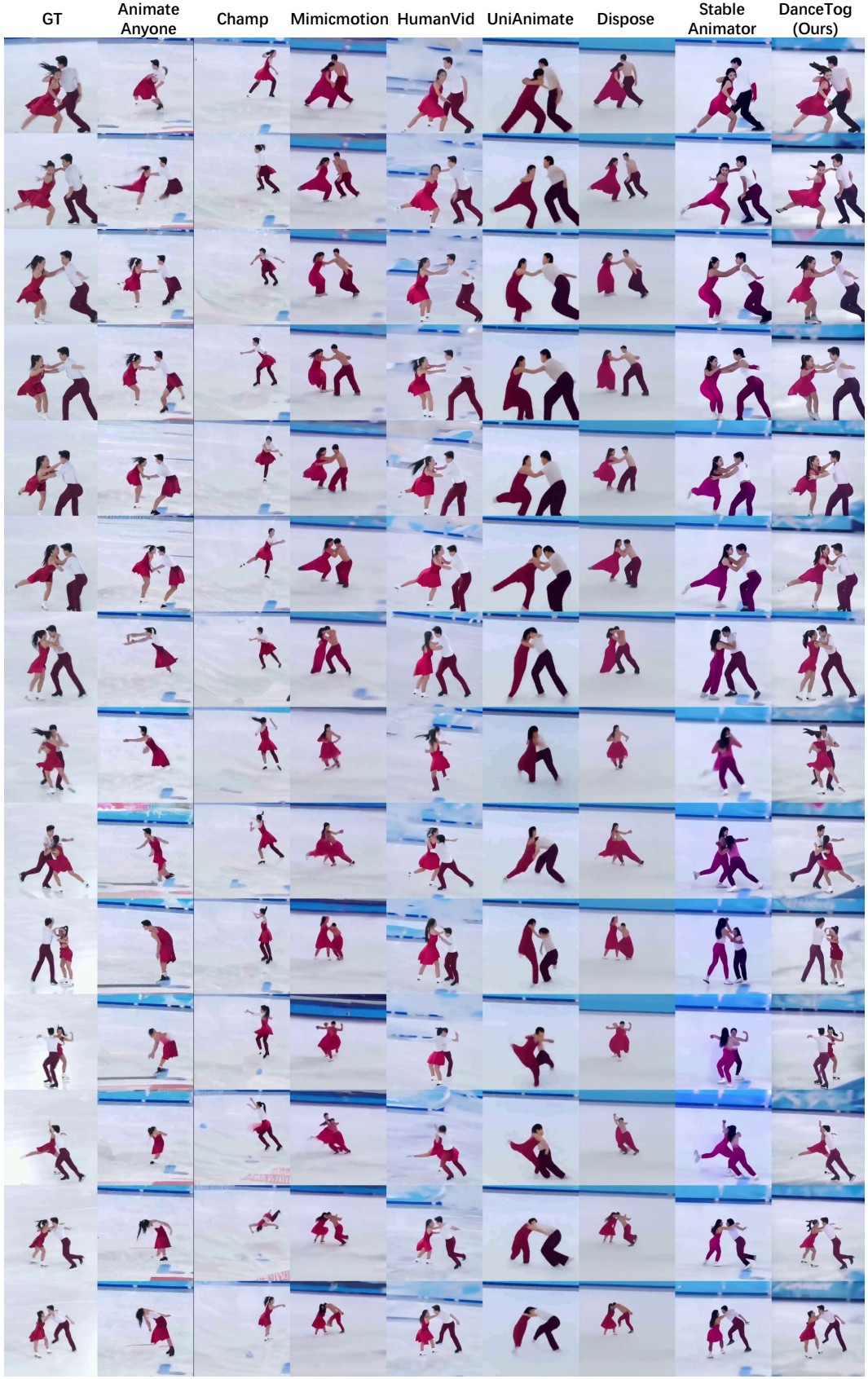

Figure 26: Additional animation results (3/6).

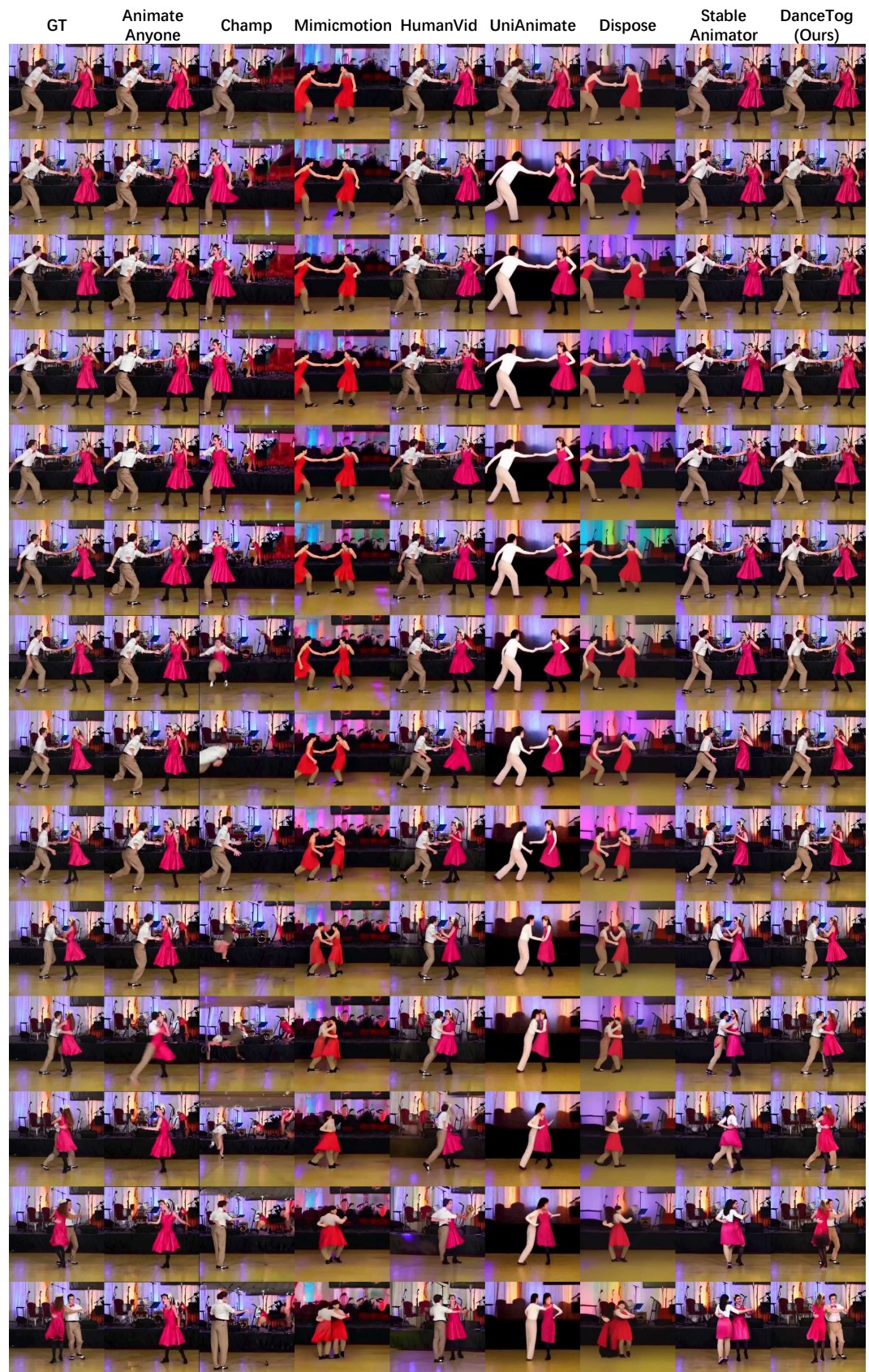

Figure 27: Additional animation results (4/6).

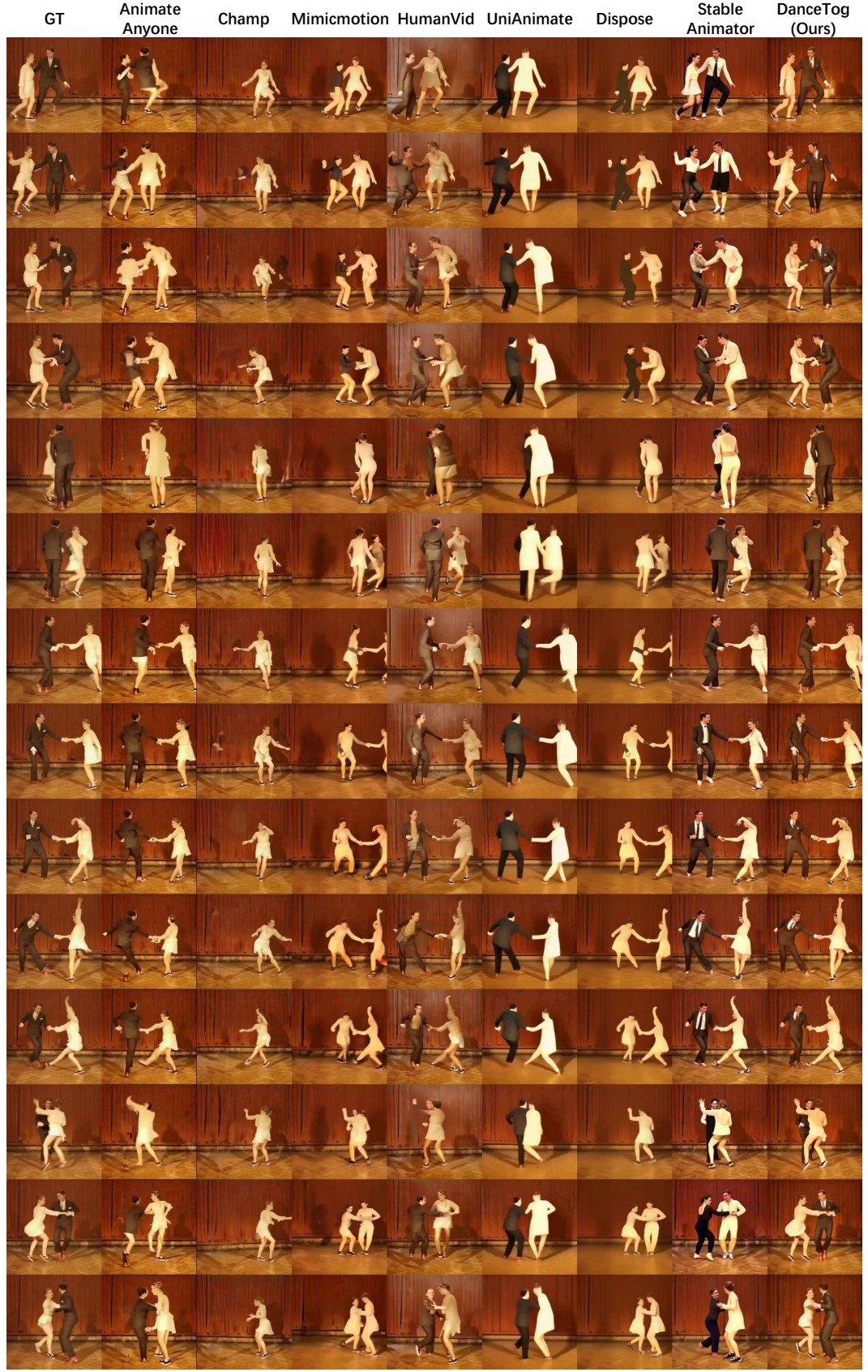

Figure 28: Additional animation results (5/6).

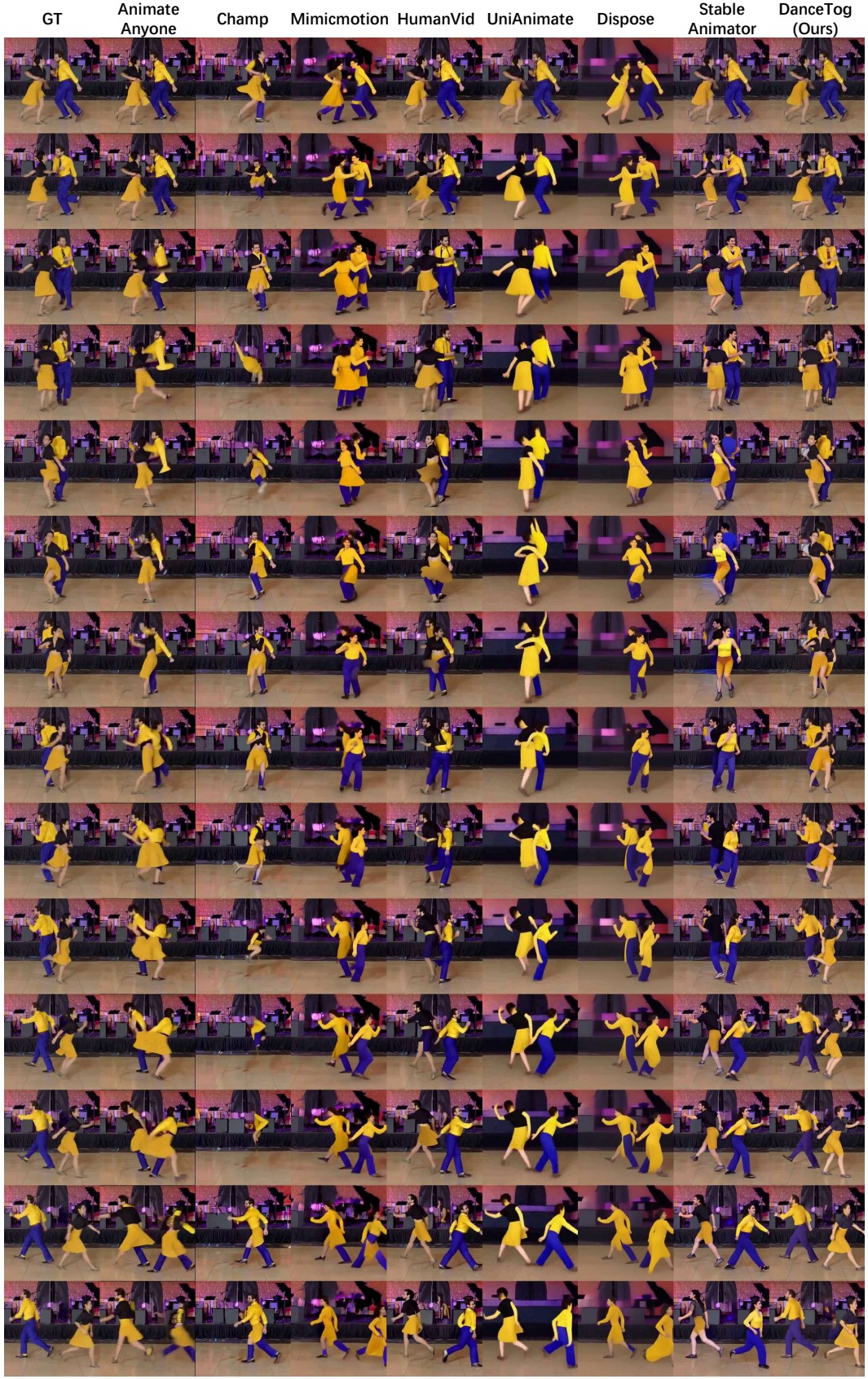

Figure 29: Additional animation results (6/6).

Table 10: Inference speed and training configuration comparison. All inference speeds are measured on a single A800 GPU (excluding model loading time). Training configurations indicate the computational resources and time required. FPS: frames per second.

| Method | Venue | Training Config | Open Source | Inference Speed (fps) |
|---|---|---|---|---|
| *Our Method* | | | | |
| **DanceTog w.** $Data_{swing}$ | – | 8×A100, 4 days (20 ep.) | ✓ | **0.88** |
| **DanceTog w.** $Data_{full}$ | – | 9 nodes×8×A800, 3 days (20 ep.) | ✓ | **0.88** |
| **DanceTog w.** $Data_{full} + Data_{PairFS}$ | – | 9 nodes×8×A800, 4 days (20 ep.) | ✓ | **0.88** |
| *Baseline Methods* | | | | |
| Animate Anyone Hu (2024) | CVPR 2024 | – | ✓* | 0.97 |
| StableAnimator Tu et al. (2024) | CVPR 2025 | 8×A100, 4 days (20 ep.) | ✓ | 0.89 |
| Champ Zhu et al. (2024a) | ECCV 2024 | – | ✓† | 1.42 |
| HumanVid Wang et al. (2024e) | NIPS 2024 | – | ✓† | 0.81 |
| Dispose Li et al. (2025a) | ICLR 2025 | – | × | 0.42 |
| UniAnimate Wang et al. (2025a) | SCIS 2025 | – | × | 0.25 |
| MimicMotion Zhang et al. (2024b) | ICML 2025 | – | × | 0.06 |
| UniAnimate-DiT Wang et al. (2025b) | – | – | × | 0.03 |

✓*: Training code available, supports dual-person data input.
✓†: Open source but data format incompatible or requires special parameters (e.g., camera parameters).
×: Not open source.
ep.: epochs.

## J  LIMITATIONS

While DanceTogether achieves state-of-the-art performance on two-person interaction benchmarks, it has several limitations. First, our framework is optimized for up to two actors; extending it to handle larger groups would incur substantial computational and memory overhead and may require hierarchical or factorized conditioning mechanisms. Second, the quality of generated videos depends heavily on the accuracy of the input pose and mask sequences. Severe occlusions, fast motion blur, or failures in the underlying detectors (e.g., DWPose (Yang et al., 2023), SAMURAI (Yang et al., 2024)) can degrade identity preservation and interaction fidelity. Third, we assume a mostly static camera and relatively simple backgrounds; dynamic camera motion or highly cluttered scenes may introduce artifacts or identity confusion. Fourth, like most diffusion-based methods, DanceTogether is computationally intensive and incurs non-trivial latency, limiting real-time applications.

## K  FUTURE WORKS

DanceTogether enables diverse applications in creative industries (film, gaming, VR/AR) and embodied-AI research. A particularly promising direction is robot imitation learning, where our framework can synthesize large-scale training data for human-robot collaboration (Albaba et al., 2025) without expensive real-world collection, as demonstrated by our HumanRob-300 results (Fig. 17).

## L  THE USE OF LARGE LANGUAGE MODELS (LLMS)

Large language models (LLMs) were used only to assist with language editing and minor text improvements in the preparation of this manuscript. They were not involved in the design of the research, the development of methods, the execution of experiments, or the interpretation of results. All scientific content, including analyses, conclusions, and contributions, remains the work of the authors.

