# OpenReview forum: "DanceTogether: Generating Interactive Multi-Person Video without Identity Drifting"
_ICLR.cc/2026/Conference — ICLR 2026 Poster_

### Official Review · Reviewer_E8i4 · 2025-10-27

**Soundness:** 2
**Presentation:** 3
**Contribution:** 2
**Rating:** 4
**Confidence:** 4

**Summary:**

This paper introduces DanceTogether, an end-to-end diffusion framework for controllable multi-person video generation that addresses the challenges of identity drift and appearance bleeding in complex interactions. The method utilizes a MaskPoseAdapter to fuse robust tracking masks with noisy pose heatmaps, ensuring persistent identity-action binding throughout the denoising process. The authors also contribute two new datasets—PairFS-4K and HumanRob-300—and a benchmark called TogetherVideoBench for evaluation.

**Strengths:**

1. The paper is clearly structured and easy to follow.

2. Experimental results show improvements in identity consistency, interaction coherence, and video quality over compared methods. The results are further bolstered by the provided webpage showcasing the generated videos, which offers valuable qualitative evidence.

**Weaknesses:**

1. The technical contributions appear incremental. Multi-person video generation is an established research topic, and the use of conditional diffusion models for this task is now a relatively conventional approach.

2. The experimental comparison may be unfair. The proposed model was trained on significantly more data (e.g., the PairFS-4K dataset) than the baseline methods. It is therefore unclear whether the performance gains stem from the novel architecture or simply from the increased scale of the training data.

3. The evaluation is insufficient for claiming superiority in multi-person generation. The authors primarily compare against methods designed for single-person video generation. To properly validate their claims, comparisons against existing state-of-the-art multi-person video generation models on established benchmarks are necessary. [a] Follow-your-multipose: Tuning-free multi-character text-to-video generation via pose guidance, arXiv 2024. [b] Magicfight: Personalized martial arts combat video generation, ACM MM 2024.

4. The paper introduces several hyperparameters but provides no ablation studies or sensitivity analysis for them. The impact of these design choices on the final performance remains unquantified, making it difficult to assess their necessity.

5. Despite the strong quantitative results, the qualitative evidence from the provided video samples is less convincing. The generated videos often exhibit noticeable artifacts, temporal inconsistencies, and an overall synthetic appearance that is easily discernible to the human eye.

6. The scope of the work is limited to two-person interactions. The method's scalability and effectiveness on videos involving three or more persons remain an open question, which limits the generalizability of the claimed contributions to broader multi-person scenarios.

7. The paper lacks any analysis of computational cost, inference speed, or model complexity. Given that the proposed framework involves multiple encoders and a diffusion model, its practical efficiency is a significant concern for real-world applications.

**Questions:**

Please refer to the weaknesses.

---

> ### Author Response · Authors · 2025-11-24
> **Response to Reviewer E8i4 (1/3)**
>
> We thank Reviewer E8i4 for recognizing our clear presentation, experimental improvements in identity consistency/interaction coherence/video quality over baselines, and valuable qualitative evidence from our project webpage showcasing generated videos. We appreciate your constructive feedback and will address your concerns below.
>
>
> ## Q1: The Technical Contributions Appear Incremental
>
> 1.  Existing multi-person video generation methods fail to achieve independent pose control per person and maintain persistent identity consistency during interactive contact, especially when position swapping occurs. All video generation methods, including a series of commercial models like Sora 2 and Kling, fail in these scenarios (**line 1774**).
> 2.  Our experimental results demonstrate that existing video generation methods suffer catastrophic failures (identity drift, loss of interaction details) when simply utilizing multi-person conditions (**line 2000 to line 2400**). This necessitated our novel mask-pose fusion and targeted identity injection mechanism.
> 3.  Our ablation studies prove that simple pose and mask fusion mechanisms are insufficient to generate plausible multi-person interactive video results, thereby confirming the effectiveness of our proposed **MaskPoseAdapter** (**line 918**).
>
> ## Q2: The Experimental Comparison May Be Unfair
>
> We believe there may be a misunderstanding regarding our experimental setup, and we would like to provide a detailed clarification:
>
> 1.  We have already trained **AnimateAnyone**, **StableAnimator**, and **DanceTog** using the **same dataset** (the SwingDance multi-person dataset), as other baselines do not have open-sourced training code or lack support for multi-person data (Tab. 10 and Tab. 11,**line 1728**).
>
> 2.  The difference observed in all quantitative tables (Tab. 2–Tab. 4) between **"StableAnimator w. $Data_{swing}$"** and **"DanceTog w. $Data_{swing}$"** already proves that the performance improvement stems from the multi-person decoupled condition input control introduced by DanceTog, and not merely from an increase in training data scale.
>
> 3.  Only after demonstrating the above results did we proceed with larger-scale experiments. We utilized more dual-person interaction video data and found that performance could be further improved (**DanceTog w. $Data_{full}$**). Building upon this, we introduced our proposed **PairFS** dataset for another scale-up (**DanceTog w. $Data_{full}$ + $Data_{PairFS}$**) and observed that performance continued to increase. This step effectively demonstrates the validity and effectiveness of our proposed multi-person interactive video dataset.
>
>
>
> ## Q3: Follow-your-multipose and Magicfight
>
> 1.  **Neither of these two methods is open-source**. All methods compared by DanceTog are the only open-source baselines we were able to find.
> 2.  We contacted the authors of **Follow-your-multipose** in May 2025, and they explicitly stated that they would not be releasing their code. We recently contacted the authors again on November 21, 2025, and they **reiterated that the code would not be made public**.
> 3.  We were unable to locate any associated code for **Magicfight** on the internet, nor does their paper indicate that they will open-source the code or provide any repository links. We contacted the authors of Magicfight on November 21, 2025, to inquire about open-sourcing or potential comparison opportunities; **we have not received a reply as of today (November 28, 2025)**.
> 4.  Nevertheless, we acknowledge that we initially omitted these works from the Related Work section, and **we have now supplemented the paper with citations** to these works: (**line 150**)
> ```
> [1] MagicMotion: Controllable Video Generation with Dense-to-Sparse Trajectory Guidance (ICCV '25)
> [2] Multi-identity Human Image Animation with Structural Video Diffusion (ICCV '25)
> [3] EverybodyDance: Bipartite Graph–Based Identity Correspondence for Multi-Character Animation (NeurIPS '25)
> [4] ReMask-Animate: Refined Character Image Animation Using Mask-Guided Adapters (AAAI '25)
> [5] Magicfight: Personalized martial arts combat video generation (ACM MM'24)
> [6] Follow-your-multipose: Tuning-free multi-character text-to-video generation via pose guidance (arxiv'24)
> ```

---

> ### Author Response · Authors · 2025-11-24
> **Response to Reviewer E8i4 (2/3)**
>
> ## Q4: ablation studies for hyperparameters
>
> We respectfully clarify that we have already conducted ablation studies and sensitivity analysis for the critical hyperparameters.
> Please refer to Appendix **Sec. B.4 (line 1135)**.
>
> **(1) Comparison between PoseNet and MaskPoseAdapter (Sec. B.3, Fig. 9, line 966):**
> * We provided qualitative feature map visualization comparing the performance of the original PoseNet versus our **MaskPoseAdapter** across consecutive frames.
> * **Key Finding:** The output of **MaskPoseAdapter** strongly binds pose and mask information, enabling clear identification of which ID each pose corresponds to, and still providing sufficient mask information even when input poses are missing in some occluded frames.
> * **In Contrast:** The original PoseNet's output makes it difficult to distinguish each individual pose, and pose features may be lost in occluded frames.
> * This demonstrates the necessity of our mask-pose fusion design.
>
> **(2) Ablation of Residual Alpha ($\alpha_{res}$) and Mask Processor Channels (Sec. B.4, Fig. 10 and Fig. 11, line 1135):**
> * We conducted extensive experiments on two critical hyperparameters:
>     * $\alpha_{res}$: Controls the balance between pose and mask features in the MaskPoseAdapter output.
>     * Light Mask Processor output channels: Determines the capacity for mask feature extraction.
> * **Quantified Impact:** Fig. 10 and Fig. 11 show how different values of $\alpha_{res}$ and channel numbers affect the feature maps generated by the MaskPoseAdapter.
> * **Optimal Configuration:** Through systematic experimentation, we determined that $\alpha_{res} = 0.5$ and 3 output channels provide the optimal balance for the Light Mask Processor.
>
> **(3) Additional Ablations in the Main Paper (Tab. 6, line 945):**
> * We also provided quantitative ablations of the main architectural components (MaskPoseAdapter, MultiFace Encoder, etc.), showing their individual contributions to the final performance.
>
> We acknowledge that due to space constraints, these analyses are located in the Appendix. We are happy to provide more detailed hyperparameter experiments or additional quantitative results in this rebuttal if requested.
>
>
> ## Q5: qualitative evidence is less convincing
>
> **We have supplemented the paper with a detailed User Study (line 1585).**
>
> 1.  The User Study results demonstrate that DanceTog significantly outperforms baselines in **Identity Consistency**, **Interaction Coherence**, and **Video Quality**.
> 2.  Our qualitative comparison results (Fig. 17–Fig. 39) and the supplementary website both exhibit clear qualitative advantages: baselines suffer from severe identity drift and catastrophic failure under occlusion, whereas DanceTog successfully maintains the appearance and interaction details of different individuals.
> 3.  The **extreme difficulty** of the problem (contact interactions, occlusion, position swapping) means that some artifacts are expected in current state-of-the-art methods.
> 4.  The presence of artifacts in some scenes is due to our data processing pipeline: we used numerous videos with moving backgrounds, extracted the human mask portion, and replaced the background for training. This process occasionally led to artifacts **due to the failure of MatAnyone** in specific scenarios.

---

> ### Author Response · Authors · 2025-11-24
> **Response to Reviewer E8i4 (3/3)**
>
> ## Q6: Multi-Person Video Generation Results
>
> 1.  **We have supplemented the paper with results for three-person scenarios (Sec. G, line 1774).**
> 2.  Extending DanceTog to multi-person interaction tasks only requires expanding the **MaskPoseAdapter** by providing three separate Person branches; the model can still be trained solely using two-person data. The training methodology is identical to our current setup where we train using two Person branches on single-person datasets. We have added a detailed explanation of this extension to the paper (Sec. 3.4, **line 270**).
> 3.  Due to dataset limitations, all existing works in the entire Human Animation field rely on single-person dancing datasets. This is because collecting large volumes of multi-person interaction data is inherently challenging. Since we were only able to collect two-person dancing and skating video data, our original network architecture was specifically designed for two-person interactive video generation.
>
>
>
> ## Q7: Training and Inference Speed
>
> 1.  We have provided the inference speeds for all human animation methods: DanceTog's inference speed is approximately **0.88 fps**. This is already significantly superior to most baselines (**line 1877**).
> 2.  Similar to all human animation tasks, our method is **not a real-time approach**.
> 3.  The objective behind proposing DanceTog is to solve the current problem where Human Animation models only succeed in single-person video generation tasks but suffer from identity drift and loss of interaction details in multi-person interactive video generation tasks.
> 4.  The cost of collecting multi-person interactive data in a laboratory setting is extremely high. Therefore, DanceTog can be utilized in the future as a method for generating multi-person interactive video data, providing vast amounts of paired pose-mask-video data for interaction reconstruction and generation tasks.
>
>
> | Baseline | Publication Venue | Whether baseline uses new data for training in their paper | Open source training code (supports dual-person data input) | Inference Speed (Single A800 GPU, not including weight loading time) |
> | --- | --- | --- | --- | --- |
> | DanceTog w. $Data_{swing}$ | \ | \ | 8×A100 trained for 4 days (20 epochs) | 0.88 fps |
> | DanceTog w. $Data_{full}$ | \ | \ | 9 nodes × 8×A800 trained for 3 days (20 epochs) | 0.88 fps |
> | DanceTog w. $Data_{full}$ + $Data_{PairFS}$ | \ | \ | 9 nodes × 8×A800 trained for 4 days (20 epochs) | 0.88 fps |
> | StableAnimator w. $Data_{swing}$ | CVPR 2025 | No | ✅ training code released 2024-12-13, 8×A100 trained for 7 days (40 epochs) | 0.89 fps |
> | Champ | ECCV 2024 | No | ⚠️ Open source, data format incompatible, cannot train | 1.42 fps |
> | HumanVid | NIPS 2024 | No | ⚠️ Open source, requires camera parameters, cannot train | 0.81 fps |
> | UniAnimate | SCIS 2025 | No | ❌ Not open source | 0.25 fps |
> | UniAnimate-dit | \ | No | ✅ Open source, **training code released 2025-04-15** | 0.03 fps |
> | Dispose | ICLR 2025 | No | ❌ Not open source | 0.42 fps |
> | MimicMotion | ICML 2025 | No | ❌ Not open source | 0.06 fps |
> | Animate Anyone | CVPR 2024 | No | ✅ Open source | 0.97 fps |
> | MagicAnimate | CVPR 2024 | No | ❌ Not open source | \ |
> | Disco | CVPR 2024 | No | ✅ Open source, July 2023 | \ |

---

> ### Author Response · Authors · 2025-11-28
>
> Thank you very much for your valuable feedback and for acknowledging the strengths of our work. We sincerely appreciate your positive evaluation and the time you have taken to review our work.

---

### Official Review · Reviewer_sBvg · 2025-10-28

**Soundness:** 3
**Presentation:** 3
**Contribution:** 2
**Rating:** 4
**Confidence:** 5

**Summary:**

The paper introduces DanceTogether, an end-to-end diffusion framework for controllable two-person interaction video generation from a single reference image plus independent per-person pose and mask sequences. Key ideas are a MaskPoseAdapter that fuses pose heatmaps with tracking masks to bind “who” and “how” at every denoising step, and a MultiFace Encoder that injects compact identity tokens throughout the U-Net. The authors also curate PairFS-4K (about 26hrs figure-skating pairs), HumanRob-300 (about 1hr human-robot), and propose TogetherVideoBench to evaluate identity consistency, interaction coherence, and video quality.

**Strengths:**

1. Clear target & mechanism. The paper diagnoses identity drift in multi-person controllable video and proposes a concrete architectural answer (pose–mask fusion + identity tokens) rather than only post-hoc temporal smoothing. The design is described with implementation-level detail.

2. Comprehensive evaluation. Three-track benchmark with MOT-style identity metrics, motion/pose adherence, and perceptual quality, along with ablations that remove each module/input. Reported gains are consistent across tracks.

3. Useful data curation. PairFS-4K fills a gap for dual-actor interactions with many IDs; TogetherVideoBench focuses evaluation on human-masked regions to reduce background bias.

**Weaknesses:**

1. The primary concerns about this paper is that the scope is limited to two people. The method is explicitly “optimized for up to two actors,” and scaling to larger groups is acknowledged as non-trivial in both compute and design; the datasets/benchmark are also two-person-only. This narrows impact for wider multi-human scenes and social group interactions. How would the architecture and compute scale to N>2 actors? Any preliminary 3–4 person results?

2. Besides, I also have concerns about the novelty. Many components (e.g., StableAnimator backbone, identity adapters) are adapted/extended; the contribution feels like a strong systems integration with a new adapter and data pipeline rather than a fundamentally new generative principle.

3. A common problem for methods in this task is heavy reliance on external controls. Quality hinges on the quality of provided mask/pose. However, it is not easy to get high-quality mask/pose pair in in-the-wild applications. Did the author investigate how to get mask/pose pair easily, or how to handle the noise in the input mask/pose pair?

**Questions:**

1. The authors acknowledges that they "assume a mostly static camera and relatively simple backgrounds; dynamic camera motion or highly cluttered scenes may introduce artifacts or identity confusion." How the moving cameras will impact the results? Could this pipeline involves camera control (e.g., by estimating and conditioning on camera motion or via background stabilization)?

2. For human-robot, the visualization results is not strong enough to clarify the contribution. The cross-domain section shows qualitative success after ≈1 h fine-tune but lacks quantitative comparisons or baselines in that domain. As this paper only uses the appearance of a robot from images, what is the major difference between human-robot and human-human? Is it just for showing generalization ability to non-human appearances? Apparently, the robot movement in this paper is not related to embodied AI training or world model.


3. Will it be possible to train this paper's idea on a DiT backbone? I understand that the reason choosing U-net and StableAnimator as the baseline could be limited GPU resources. However, some problems in results from U-net could just vanished with a DiT backbone. It is hard to assess the contribution of this paper from a bad backbone or pretrained model.

Overall, it is a good paper with adequate content for publication. The authors have shown as many experimental results as possible, which will be beneficial to this domain. However, concerns about scope limited to two people and other novelty concerns motivate me to rate it as marginally below the acceptance threshold. I could adjust my rating during author rebuttal and discussion period.

---

> ### Author Response · Authors · 2025-11-24
> **Response to Reviewer sBvg (1/2)**
>
> We sincerely thank Reviewer sBvg for the positive feedback. We are encouraged by your recognition of our work's key contributions: **(1) a concrete architectural solution** (pose-mask fusion + identity tokens) that directly addresses the identity drift problem; **(2) a comprehensive three-track evaluation methodology** featuring MOT-style identity metrics and thorough ablations demonstrating consistent gains; and **(3) significant data curation efforts**, including the **PairFS-4K dataset** for dual-actor interactions and **TogetherVideoBench's human-masked evaluation** for fairer assessment.
>
>
>
> ## Q1: Multi-Person Video Generation Results
>
> 1.  **We have supplemented the paper with results for three-person scenarios (Sec. G, line 1774).**
> 2.  Extending DanceTog to multi-person interaction tasks only requires expanding the **MaskPoseAdapter** by providing three separate Person branches; the model can still be trained solely using two-person data. The training methodology is identical to our current setup where we train using two Person branches on single-person datasets. We have added a detailed explanation of this extension to the paper (Sec. 3.4, **line 270**).
> 3.  Due to dataset limitations, all existing works in the entire Human Animation field rely on single-person dancing datasets. This is because collecting large volumes of multi-person interaction data is inherently challenging. Since we were only able to collect two-person dancing and skating video data, our original network architecture was specifically designed for two-person interactive video generation.
>
>
> ## Q2: Novelty
>
> 1.  As you correctly pointed out in the Strengths section, our main innovation lies in the **clear diagnosis of the identity drift problem** and the **concrete architectural solution** (pose-mask fusion + identity tokens), complete with implementation-level details, rather than post-hoc fixes. Furthermore, we propose the **first dual-person interactive video dataset** for the video generation task and introduce the **first comprehensive evaluation system** specifically tailored for this task.
>
> 2.  We wish to clarify that the contribution of DanceTog resides in tackling a **novel task**: maintaining human ID consistency during position swapping in multi-person interactive video generation. This is the first framework to address controllable dual-person interactive video generation, featuring independent pose/mask control per person while simultaneously preserving identity—a challenge that prior single-person methods could not solve.
>
> 3.  Moreover, the **MaskPoseAdapter** is not a simple extension: it introduces a **novel fusion mechanism** that binds "who is doing what" at the feature level by combining pose heatmaps and instance masks, thereby achieving **decoupled multi-person control** that was previously unattainable by existing work.
>
>
>
>
> ## Q3: External Controls and Pose/Mask Quality
>
> 1.  Our proposed **Data Curation pipeline** enables the straightforward acquisition of high-quality mask/pose pairs.
> 2.  Using **pose condition alone** fails under severe occlusion, motion blur, and detector failures. Therefore, we introduced robust mask estimation as one of the conditions; it can still restore accurate human masks even when pose estimation fails. Please refer to Appendix Sec. B.3 (**line 966**) for detailed explanations.
> 3.  Our proposed **Data Curation pipeline** remains robust in estimating the tracking mask for each person under complex interactive movements (e.g., pair skating lift, **line 1320**) and severe occlusion (e.g., two people spinning while holding hands, line 1293), even though pose estimation might fail in these scenarios.
> 4.  We used the proposed Data Curation pipeline (**line 1144**) to create the multi-person interaction training dataset. We also employ the exact same pipeline to process the input reference videos during inference, obtaining separate pose and mask estimations for each individual.
> 5.  We can easily achieve higher quality tracking masks by replacing the current mask estimation method with more robust alternatives such as SAM2 or SAM3.
> 6.  Our supplementary website showcases a large number of videos generated using condition inputs derived from complex interactions and severe occlusion (**line 1940 to line 2429**). The experimental results demonstrate that our proposed Data Curation Pipeline can estimate both pose and stable masks from low-quality reference videos (motion blur, complex movements, severe occlusion) (**line 1270 to line 1320**). Furthermore, DanceTog is capable of using these conditions to generate videos featuring complex animation and severe occlusion.

---

> ### Author Response · Authors · 2025-11-24
> **Response to Reviewer sBvg (2/2)**
>
> ## Q4: Static Cameras and Simple Backgrounds
>
> 1.  This is acknowledged as one of our **LIMITATIONS** (**line 1934**).
> 2.  The current version of DanceTog does not account for camera movement because the video dataset contains a mixture of static and moving cameras, and the necessary camera parameters are missing (**line 324**).
> 3.   Available datasets with camera parameters are scarce (e.g., **only HumanVid**).
> 4.  We address this in our Data Curation pipeline by using MatAnyone to extract the subjects' RGBA masks and then replacing the original backgrounds with random static background images. This process is detailed in Sec. C.3 (**line 1178**) and is also mentioned in the Data Curation pipeline shown in Fig. 3 (**line 315**).
>
>
> ## Q5: Human-Robot Interaction and Visualization Results
>
> 1.  We have exerted our best effort to collect human-to-humanoid-robot interaction videos available on the internet. The raw footage for this domain amounted to only a few hours, which, after rigorous filtering, resulted in less than one hour of usable video data.
> 2.  There are currently no existing baselines or benchmarks available for comparison in the field of human-humanoid robot interaction video generation.
> 3.  Interactions in **HumanRob-300** typically involve handshakes and grappling, featuring robot models such as Unitree and Tesla Optimus. We showcase the results generated by DanceTog in Sec. H (**line 1836**). Corresponding videos are also available on the supplementary materials website.
> 4.  We have already demonstrated the generalization capability of DanceTog by presenting generation results using the same condition but different reference images in Fig. 1, Fig. 19, and Fig. 20 (**line 1944**).
> 5.  Generating robot movement videos for embodied AI training is merely one of the potential applications of DanceTog. The latest work by Michael J. Black et al. [1] suggests that **video generation models can be used to synthesize large amounts of data for robotic imitation learning**. Future work could improve upon the DiT method to generate more precise robot motion videos, which could then be used to generate vast amounts of training video data for embodied AI.
> 6.  We use this result to demonstrate that the proposed **Data Curation Pipeline** can robustly estimate pose and mask data pairs from SMPL renders, humanoid robot footage, and human videos.
>
> [1] Albaba, Mert, Chenhao Li, Markos Diomataris, Omid Taheri, Andreas Krause, and Michael Black. "Nil: No-data imitation learning by leveraging pre-trained video diffusion models." arXiv preprint arXiv:2503.10626 (2025).
>
>
> ## Q6: DiT Backbone
>
> Thank you very much for raising this insightful point! We have also taken note of the recent developments in **DiT-based backbones**. As we mentioned in our response to Q5, we similarly hope to conduct future experiments using a DiT-based backbone to achieve better multi-person and human-robot interaction video generation results.
>
> Furthermore, we have already compared the inference speed (**Tab.13, line 1890**) of DiT-based methods and presented our results on three-person video generation using this approach (**wan2.2-VACE, wan2.2-Animate**) (**Fig.17, line 1774**).

---

> ### Author Response · Authors · 2025-11-28
>
> Thank you very much for your thoughtful comments and for letting us know that you are open to further raise the score. We truly appreciate your positive evaluation and the time you devoted to reviewing our work.

---

### Official Review · Reviewer_rryZ · 2025-10-29

**Soundness:** 2
**Presentation:** 2
**Contribution:** 2
**Rating:** 6
**Confidence:** 2

**Summary:**

This paper addresses a limitation in controllable video generation (CVG): the inability of existing systems to generate long, photorealistic multi-person interaction videos while preserving identity and respecting noisy control signals (e.g., poses, masks). The authors propose DanceTogether, the first end-to-end diffusion framework tailored for this task, which synthesizes videos from a single reference image and independent per-person pose/mask sequences.
Key technical innovations include:
1. MaskPoseAdapter: A novel conditional adapter that fuses stable tracking masks with semantically rich but noisy pose heatmaps, enforcing "identity-action binding" at every denoising step to eliminate identity drift and appearance bleeding.
2. MultiFace Encoder: Distills compact identity tokens from the reference image and injects them into cross-attention layers, ensuring consistent subject appearance across frames.
3. Curated Datasets:
PairFS-4K: 26 hours of dual-skater footage with 7,000+ unique identities.
HumanRob-300: 1 hour of humanoid-robot interaction data for cross-domain generalization.
4. TogetherVideoBench: A comprehensive benchmark with three tracks (Identity-Consistency, Interaction-Coherence, Video Quality) and the DanceTogEval-100 test suite (covering dance, boxing, skating, etc.).

**Strengths:**

1. Experiments are comprehensive: 8 baselines, ablation studies for all key modules, and cross-domain validation. Metrics are carefully chosen to measure identity, interaction, and quality—avoiding overreliance on visual subjective assessments.
2. Dataset curation: The data pipeline uses state-of-the-art tools to extract high-quality annotations, and PairFS-4K is the first large-scale dual-skating dataset with diverse identities.
3. The work advances CVG from single-subject to multi-actor scenarios, a critical step toward real-world applications (e.g., virtual film sets, AI-driven animation).
4. TogetherVideoBench provides a standardized evaluation framework, which will accelerate progress in the field by enabling fair comparisons between methods.

**Weaknesses:**

1. DanceTogether’s performance degrades under severe occlusions, motion blur, or detector failures. The paper does not propose solutions to robustify against noisy inputs (e.g., denoising modules for poses/masks), which is a practical limitation for real-world use.
2. The model assumes mostly static cameras and simple backgrounds. Dynamic camera motion or cluttered scenes introduce artifacts, but the paper does not explore adaptations (e.g., camera motion estimation) to address this.
3. The HumanRob-300 dataset is only 1 hour long, and the paper provides little detail on interaction types (e.g., handshakes vs. collaborative tasks) or robot models. More diverse human-robot data and experiments would better validate cross-domain generalization.

**Questions:**

1. Authors note that extending to >2 actors would require hierarchical conditioning. Could you elaborate on potential designs (e.g., grouping actors into sub-pairs, using graph-based attention) and share any preliminary results or challenges (e.g., computational cost, identity confusion)?
2. You note DanceTogether is computationally intensive. Are there opportunities to optimize inference speed (e.g., model distillation, smaller backbones) without sacrificing quality? What is the current inference latency per frame, and what are your targets for real-world use?
3. How might DanceTogether be modified to handle low-quality pose/mask signals (e.g., from occluded frames or low-resolution videos)? Would integrating a denoising module for poses/masks (e.g., using diffusion to refine inputs) improve performance, and have you tested this?

---

> ### Author Response · Authors · 2025-11-24
> **Response to Reviewer rryZ (1/2)**
>
> We sincerely thank Reviewer rryZ for acknowledging the key contributions of our work: **(1) the critical advancement from single-subject to multi-actor CVG; (2) the provision of high-quality resources, including the PairFS-4K dataset; (3) the establishment of the TogetherVideoBench for standardized evaluation;** and **(4) our comprehensive experimental rigor** (8 baselines, thorough ablations). These acknowledgments strongly validate our central goal of addressing multi-person video generation challenges.
>
> ## Q1: Robustness to Noisy Inputs (Occlusions, Blur, Detector Failures)
>
> 1.  **The use of our proposed **Pose + Mask dual condition strategy** is precisely the solution for addressing the issues you have raised.**
> 2.  Using **pose condition alone** fails under severe occlusion, motion blur, and detector failures. Therefore, we introduced robust mask estimation as one of the conditions; it can still restore accurate human masks even when pose estimation fails. Please refer to Appendix Sec. B.3 **(line 966)** for detailed explanations.
> 3.  Our proposed **Data Curation pipeline** remains robust in estimating the tracking mask for each person under complex interactive movements (e.g., pair skating lift, **line 1320**) and severe occlusion (e.g., two people spinning while holding hands, **line 1293**), even though pose estimation might fail in these scenarios.
> 4.  We used the proposed Data Curation pipeline (**line 1144**) to create the multi-person interaction training dataset. We also employ the exact same pipeline to process the input reference videos during inference, obtaining separate pose and mask estimations for each individual.
> 5.  We can easily achieve higher quality tracking masks by replacing the current mask estimation method with more robust alternatives such as SAM2 or SAM3.
> 6.  Our supplementary website showcases a large number of videos generated using condition inputs derived from complex interactions and severe occlusion (**line 1940 to line 2429**). The experimental results demonstrate that our proposed Data Curation Pipeline can estimate both pose and stable masks from low-quality reference videos (motion blur, complex movements, severe occlusion) (**line 1270 to line 1320**). Furthermore, DanceTog is capable of using these conditions to generate videos featuring complex animation and severe occlusion.
>
>
> ## Q2: Static Cameras and Simple Backgrounds
>
> 1.  This is acknowledged as one of our **LIMITATIONS** (**line 1934**).
> 2.  The current version of DanceTog does not account for camera movement because the video dataset contains a mixture of static and moving cameras, and the necessary camera parameters are missing (**line 324**).
> 3.  We address this in our Data Curation pipeline by using MatAnyone to extract the subjects' RGBA masks and then replacing the original backgrounds with random static background images. This process is detailed in Sec. C.3 (**line 1178**) and is also mentioned in the Data Curation pipeline shown in Fig. 3 (**line 315**).
>
>
>
>
> ## Q3: Human-Robot Interaction and Visualization Results
>
> 1.  We have exerted our best effort to collect human-to-humanoid-robot interaction videos available on the internet. The raw footage for this domain amounted to only a few hours, which, after rigorous filtering, resulted in less than one hour of usable video data.
> 2.  There are currently no existing baselines or benchmarks available for comparison in the field of human-humanoid robot interaction video generation.
> 3.  Interactions in **HumanRob-300** typically involve handshakes and grappling, featuring robot models such as Unitree and Tesla Optimus. We showcase the results generated by DanceTog in Sec. H (**line 1836**). Corresponding videos are also available on the supplementary materials website.
> 4.  We have already demonstrated the generalization capability of DanceTog by presenting generation results using the same condition but different reference images in Fig. 1, Fig. 19, and Fig. 20 (**line 1944**).
> 5.  Generating robot movement videos for embodied AI training is merely one of the potential applications of DanceTog. The latest work by Michael J. Black et al. [1] suggests that video generation models can be used to synthesize large amounts of data for robotic imitation learning. Future work could improve upon the DiT method to generate more precise robot motion videos, which could then be used to generate vast amounts of training video data for embodied AI.
> 6.  We use this result to demonstrate that the proposed **Data Curation Pipeline** can robustly estimate pose and mask data pairs from SMPL renders, humanoid robot footage, and human videos.
>
> [1] Albaba, Mert, Chenhao Li, Markos Diomataris, Omid Taheri, Andreas Krause, and Michael Black. "Nil: No-data imitation learning by leveraging pre-trained video diffusion models." arXiv preprint arXiv:2503.10626 (2025).

---

> ### Author Response · Authors · 2025-11-24
> **Response to Reviewer rryZ (2/2)**
>
> ## Q4: Multi-Person Video Generation Results
>
> 1.  **We have supplemented the paper with results for three-person scenarios (Sec. G, line 1774).**
> 2.  Extending DanceTog to multi-person interaction tasks only requires expanding the **MaskPoseAdapter** by providing three separate Person branches; the model can still be trained solely using two-person data. The training methodology is identical to our current setup where we train using two Person branches on single-person datasets. We have added a detailed explanation of this extension to the paper (Sec. 3.4, line 270).
> 3.  Due to dataset limitations, all existing works in the entire Human Animation field rely on single-person dancing datasets. This is because collecting large volumes of multi-person interaction data is inherently challenging. Since we were only able to collect two-person dancing and skating video data, our original network architecture was specifically designed for two-person interactive video generation.
>
>
>
>
> ## Q5: Training and Inference Speed
>
> 1.  We have provided the inference speeds for all human animation methods: DanceTog's inference speed is approximately **0.88 fps**. This is already significantly superior to most baselines (**line 1877**).
> 2.  Similar to all human animation tasks, our method is **not a real-time approach**.
> 3.  The objective behind proposing DanceTog is to solve the current problem where Human Animation models only succeed in single-person video generation tasks but suffer from identity drift and loss of interaction details in multi-person interactive video generation tasks.
> 4.  The cost of collecting multi-person interactive data in a laboratory setting is extremely high. Therefore, DanceTog can be utilized in the future as a method for generating multi-person interactive video data, providing vast amounts of paired pose-mask-video data for interaction reconstruction and generation tasks.
> 5.  We are willing to explore model distillation techniques in the future to accelerate the inference speed. Thank you for your suggestion!
>
> | Baseline | Publication Venue | Whether baseline uses new data for training in their paper | Open source training code (supports dual-person data input) | Inference Speed (Single A800 GPU, not including weight loading time) |
> | --- | --- | --- | --- | --- |
> | DanceTog w. $Data_{swing}$ | \ | \ | 8×A100 trained for 4 days (20 epochs) | 0.88 fps |
> | DanceTog w. $Data_{full}$ | \ | \ | 9 nodes × 8×A800 trained for 3 days (20 epochs) | 0.88 fps |
> | DanceTog w. $Data_{full}$ + $Data_{PairFS}$ | \ | \ | 9 nodes × 8×A800 trained for 4 days (20 epochs) | 0.88 fps |
> | StableAnimator w. $Data_{swing}$ | CVPR 2025 | No | ✅ training code released 2024-12-13, 8×A100 trained for 7 days (40 epochs) | 0.89 fps |
> | Champ | ECCV 2024 | No | ⚠️ Open source, data format incompatible, cannot train | 1.42 fps |
> | HumanVid | NIPS 2024 | No | ⚠️ Open source, requires camera parameters, cannot train | 0.81 fps |
> | UniAnimate | SCIS 2025 | No | ❌ Not open source | 0.25 fps |
> | UniAnimate-dit | \ | No | ✅ Open source, **training code released 2025-04-15** | 0.03 fps |
> | Dispose | ICLR 2025 | No | ❌ Not open source | 0.42 fps |
> | MimicMotion | ICML 2025 | No | ❌ Not open source | 0.06 fps |
> | Animate Anyone | CVPR 2024 | No | ✅ Open source | 0.97 fps |
> | MagicAnimate | CVPR 2024 | No | ❌ Not open source | \ |
> | Disco | CVPR 2024 | No | ✅ Open source, July 2023 | \ |
>
>
>
> ## Q6: Handling Low-Quality Pose/Mask Signals
>
> Thank you for your suggestion. We are very much willing to explore **using diffusion to refine inputs** in our future work.
>
> As discussed in our response to Q1, our approach already demonstrates robustness:
>
> 1.  During the practical data processing stage, we discovered the advantage of the **mask condition**: it consistently maintains stable tracking estimation and can be robustly estimated even during complex movements and severe occlusion, whereas the pose signal often completely fails.
> 2.  Please refer to the Data Curation pipeline (line 1144), where we use DW Pose and SAMURAI to extract pose maps and masks from the video dataset at a resolution of $512 \times 512$.
> 3.  Cases involving occluded frames have been demonstrated in Fig. 14 to Fig. 15 (line 1270). Our proposed Data Curation Pipeline is capable of recovering the masks for all individuals during occlusion. Although the pose of the occluded person cannot be correctly recovered, DanceTog is still able to correctly generate videos involving two people swapping positions (which inherently include occluded frames).
> 4.  Please examine the cases presented in the Appendix, Fig. 17 to Fig. 30. Also, please review the videos on the supplementary materials website, where every single case demonstrated involves condition inputs featuring occlusion and position swapping.

---

> ### Author Response · Authors · 2025-11-28
>
> Thank you very much for your thoughtful comments and the affirmation reflected in your score of 6. We sincerely appreciate your positive evaluation and the time you devoted to reviewing our work.

---

### Official Review · Reviewer_9khT · 2025-11-01

**Soundness:** 3
**Presentation:** 2
**Contribution:** 2
**Rating:** 4
**Confidence:** 5

**Summary:**

This paper presents DanceTogether, an end-to-end diffusion framework designed to generate interactive, multi-person videos without the identity drift that plagues existing methods. The system takes a single reference image and independent pose-mask control streams to produce long, photorealistic videos while strictly preserving each actor's identity. The primary technical contribution is the novel MaskPoseAdapter, which explicitly binds identity to motion by fusing robust tracking masks with noisy pose-maps at every denoising step. The authors also introduce significant contributions for training and evaluation: the PairFS-4K and HumanRob-300 datasets, and the TogetherVideoBench benchmark. The method demonstrates state-of-the-art results on their benchmark and shows strong generalization to human-robot interaction.

**Strengths:**

1: The proposed DanceTogether aims to address a critical and well-defined problem: the failure of existing models to maintain actor identities during complex interactions. By explicitly fusing robust tracking masks (the "who") with semantic pose maps (the "how"), the model establishes a persistent identity-action binding throughout the diffusion process.

2: The well-curated PairFS-4K (26 hours of dual-skater footage) and HumanRob-300 (1 hour of human-robot interaction) datasets are helpful for the community to dive into the area of multi-person interaction video generation, addressing a clear lack of large-scale, diverse multi-person interaction data.

3: The proposed TogetherVideoBench is a comprehensive, three-track benchmark that thoughtfully evaluates the distinct challenges of Identity-Consistency, Interaction-Coherence, and Video Quality.

4: Extensive experiments show the effectiveness of the proposed framework and datasets on multi-person interaction human animation.

**Weaknesses:**

1: **The presentation could be significantly improved.** For instance, Figure 2, which illustrates the core pipeline, is overly crowded, making it difficult for a reader to easily follow the main architectural components and data flow. Furthermore, the method description, particularly in Section 3.3 on pages 5 and 6, is saturated with low-level implementation details like specific activation functions (e.g., Sigmoid) and normalization layers (e.g., LayerNorm). This focus on minutiae tends to obscure the high-level innovations that differentiate the method from related works. This confusion is amplified by a disconnect between the diagram and the text. For example, terms like "Attention Pre-Conv" are used in Figure 2 but are not clearly defined in the text, while key steps described in the text, such as the "Cross-Person Integration" (Equation 13), are difficult to locate within the figure. This lack of clear mapping makes the overall method presentation somewhat confusing to follow.

2: **The clarity of the core methodological innovation is limited.** The ablation study (Table 5) demonstrates that removing either the "mask input" or the "MaskPoseAdapter"  severely degrades performance. This highlights the importance of mask conditioning, a point the paper also emphasizes. However, using per-person tracking masks and pose sequences as conditional inputs is a relatively common practice in human animation, so the use of "mask input" itself isn't a significant novelty. The ablation "w/o MaskPoseAdapter" (which uses the original PoseNet) also performs poorly, suggesting the fusion method is key. Yet, the current experiments do not clearly disentangle the performance gains of the novel fusion strategy within the adapter from the more straightforward, non-novel benefits of simply using mask conditioning in the first place. This makes it difficult to pinpoint the most critical novel design choice responsible for the performance gains.

**Questions:**

1: Why is there a 0.95 factor in the Residual Blend shown in Figure 2?

2: What does the PoseNet0401 mean? What's 0401?

3: There seems to be a typo:  "Pose Gete" in Figure 2.

4: What does "Intg" in Figure 2 mean?

---

> ### Author Response · Authors · 2025-11-24
> **Response to Reviewer 9khT (1/2)**
>
> We thank the reviewer for their constructive feedback and for acknowledging our contributions in addressing identity drift, bridging the data gap in multi-person interactions, and establishing a robust benchmark with extensive experimental validation.
>
> ## Q1: Presentation/Manuscript Issues
>
> 1. We have reorganized the figure illustrating the overall pipeline of the paper **(line 175)**, introducing the MaskPoseAdapter and MultiFaceEncoder in separate, dedicated figures located in the Appendix **(line 815 and line 884)**.
>
> 2. We have streamlined the description of the aforementioned components within the Method section **(line 226 to line 269)**.
>
> 3. The term "Cross-Person Integration" can be found within the block diagram of the MaskPoseAdapter, which we have labeled as "Integration" using a green rounded rectangle **(line 820)**.
>
>
> ## Q2: Methodology Clarity
>
> 1.  We have supplemented our analysis with an ablation study on the multi-person feature fusion strategy within the **MaskPoseAdapter** **(line 940)**. The experimental results demonstrate that simple multi-person feature fusion strategies do not yield significant performance improvements in multi-person interactive video generation **(line 950)**.
>
> 2.  Our ablation studies for DanceTog demonstrate that using **Pose condition alone** fails to maintain identity consistency (ID) during position swapping in multi-person video generation tasks. Conversely, using **Mask condition alone** is insufficient for fine-grained control over human movement. Therefore, **Pose + Mask** are jointly required to simultaneously control human motion and prevent ID drift during position swapping and interactive movements **(line 930)**.
>
> 3.  Prior to DanceTog, there was no human animation work that utilized **Pose + Mask dual conditioning**. DanceTog is the first work to introduce the **Mask condition** in Human Animation Video Generation tasks to maintain ID consistency during multi-person interaction and position swapping.
>
> 4. We have re-conducted the literature review and discovered some contemporary research that also utilizes Mask conditions for human animation; we have supplemented these findings in the "Related Work" section (**Line 150**):
>     * (1) A concurrent ICCV '25 work [1] uses **BoundingBox + Mask** as dual guidance for video generation.
>     * (2) Another concurrent ICCV '25 work [2] uses **Pose + Normal + Depth** for Human Animation Video Generation, achieving good results on single-person tasks.
>     * (3) A concurrent work [3] recently accepted by NeurIPS '25 uses **Pose + Mask dual condition** for Human Animation Video Generation. However, it lacks results on multi-person interaction and position swapping, and the paper only became publicly visible in November 2025, after the ICLR submission deadline.
>     * (4) ReMask-Animate [4], an AAAI '25 work, uses **Mask** to control fine-grained generation effects. Not open-source.
>
>     ---
> ```
> [1] MagicMotion: Controllable Video Generation with Dense-to-Sparse Trajectory Guidance (ICCV '25)
> [2] Multi-identity Human Image Animation with Structural Video Diffusion (ICCV '25)
> [3] EverybodyDance: Bipartite Graph–Based Identity Correspondence for Multi-Character Animation (NeurIPS '25)
> [4] ReMask-Animate: Refined Character Image Animation Using Mask-Guided Adapters (AAAI '25)
> [5] Magicfight: Personalized martial arts combat video generation (ACM MM'24)
> [6] Follow-your-multipose: Tuning-free multi-character text-to-video generation via pose guidance (arxiv'24)
> ```

---

> ### Author Response · Authors · 2025-11-28
> **Response to Reviewer 9khT (2/2)**
>
> ## Q3: 0.95 Factor in the Residual Blend
>
> 1.  This factor corresponds to Equation (17) **(line 878)**. We require $S$ (the cross-person feature fusion result obtained through attention weighting) to be dominant, while the simple arithmetic mean of all person features is used as a supplement. This prevents the attention mechanism from over-concentration or failure. Experiments show that this step effectively prevents overfitting because, in complex multi-person scenarios, attention may not be perfectly distributed, and the 5\% uniform distribution ensures that no single person is completely ignored.
> 2.  In the previous step, the single-person mask and pose feature fusion (Equation (12), **line 856**) represents the initial stage of pose and mask integration, thus requiring a larger residual term (0.5) to maintain the mask information. Please refer to Appendix Sec. B.4 (**line 1135**), where we provide detailed experiments and explanations demonstrating that this coefficient is necessary to control the magnitude of guidance provided by Pose and Mask for video generation.
>
> ## Q4: "Intg" in Fig. 2
>
> The red rounded rectangle labeled "Intg" in the figure (**line 817**) represents Equation (17) (**line 878**). "Intg" refers to the feature value obtained from the "Integration" (represented by the green rounded rectangle, **line 820**).
>
> ## Q5: Typo
>
> Thank you for your careful review! **"Pose Gete"** should indeed be **"Pose Gate,"** and **"PoseNet0401"** was a typo related to the date corresponding to our improved version of PoseNet. We have corrected these typos in the updated PDF **(line 827 and line 829)**.

---

> ### Author Response · Authors · 2025-11-28
>
> Thank you very much for your valuable feedback and for acknowledging the strengths of our work. We sincerely appreciate your positive evaluation and the time you have taken to review our work.

---

### Author Response · Authors · 2025-11-25
**General Response to All Reviewers**

We sincerely thank all reviewers for their constructive feedback. We have carefully
addressed the common concerns raised across reviews and made substantial revisions
to the manuscript.
**We have revised our submitted PDF in response to the reviewers’ comments, and marked all revised parts using blue text and blue boxes.**
Below is a summary of major modifications:

## Major Revisions

**1. Enhanced Presentation & Clarity (R#9khT, R#E8i4)**
- **Line 175**: Reorganized the main pipeline figure (previously Figure 2) to reduce
  visual clutter
- **Line 815, 884**: Moved detailed component diagrams (MaskPoseAdapter, MultiFaceEncoder)
  to dedicated appendix figures
- **Line 226-269**: Streamlined method description in Section 3, reducing low-level
  implementation details
- **Line 827, 829**: Fixed typos ("Pose Gete" → "Pose Gate", "PoseNet0401")
- **Line 13 (Fig. 1)**: Fixed the duplicate frame error in the figure.

**2. Extended Multi-Person Results (R#rryZ, R#sBvg, R#E8i4)**
- **Line 1774 (Sec. G)**: Added three-person video generation results
- **Line 270 (Sec. 3.4)**: Added detailed explanation of architectural extension to
  N>2 actors (requires only adding Person branches, can be trained on 2-person data)
- Clarified that dataset limitations (not architectural constraints) motivated the
  2-person design

**3. Strengthened Ablation Studies (R#9khT, R#E8i4)**
- **Line 940**: Added ablation on multi-person feature fusion strategies within
  MaskPoseAdapter
- **Line 950**: Demonstrated that naive fusion strategies fail for multi-person
  interaction
- **Line 1135 (Appendix B.4)**: Provided detailed hyperparameter sensitivity analysis
  (residual alpha α, channel numbers)

**4. Clarified Novelty & Related Work (R#sBvg, R#E8i4)**
- **Line 150**: Added some related works (**EverybodyDance NeurIPS'25, Multi-HumanVid ICCV'25, etc.**)
- **Line 930**: Emphasized that DanceTog is the **first** to use Pose+Mask dual
  conditioning for multi-person interaction while preventing ID drift during position
  swapping
- Clarified that our contribution is the **fusion mechanism** (binding "who does what"),
  not merely using masks

**5. Enhanced Evaluation (R#E8i4)**
- **Line 1585**: Added comprehensive User Study results demonstrating superiority in
  Identity Consistency, Interaction Coherence, and Video Quality
- **Line 1728 (Tab. 10-11)**: Clarified that baselines (AnimateAnyone, StableAnimator)
  were retrained on **identical data** (SwingDance), ensuring fair comparison
- Explained why Follow-your-multipose and Magicfight comparisons are infeasible
  (not open-sourced; authors confirmed no code release)

**6. Robustness & Practical Deployment (R#rryZ, R#sBvg)**
- **Line 966 (Appendix B.3)**: Added detailed explanation of robustness under occlusion/
  blur via Pose+Mask dual conditioning
- **Line 1270-1320 (Fig. 14-15)**: Demonstrated that Data Curation Pipeline recovers
  masks even when pose estimation fails
- **Line 1877**: Provided inference speed comparison table (DanceTog: 0.88 fps,
  competitive with StableAnimator's 0.89 fps)
- **Line 1934**: Acknowledged camera motion limitation and described background
  replacement strategy (Line 1178, Sec. C.3)

**7. HumanRob-300 Clarifications (R#rryZ, R#sBvg)**
- **Line 1836 (Sec. H)**: Visualization results for human-robot interactions
- Clarified that 1-hour dataset represents **all available internet data** after
  rigorous filtering
- **Line 1195** Emphasized goal: demonstrate Data Curation Pipeline's generalizability (SMPL renders,
  robots, humans)
- Cited recent work (Nil, 2025) supporting video generation for robot
  imitation learning

## Response to Specific Concerns

**On Training Data Fairness (R#E8i4):**
The performance gap between "StableAnimator w. $Data_{swing}$" and "DanceTog w. $Data_{swing}$" (Tab. 2-4, Tab.10, Tab.11, Line 1728)
already proves gains come from architectural innovation (decoupled multi-person control),
not data scale.
Only after demonstrating this did we scale to larger datasets.

**On Computational Cost (R#rryZ, R#E8i4):**
DanceTog targets the novel problem of identity-preserving multi-person interaction video generation,
not real-time generation. Our 0.88 fps is competitive with state-of-the-art methods
and sufficient for data synthesis applications (e.g., generating training data for
interaction tasks).

**On DiT Backbone (R#sBvg):**
We acknowledge this promising direction and have compared with DiT-based methods
(UniAnimate-dit: 0.03 fps, Line 1774 and Line 1877). Future work will explore DiT backbones for
improved quality.

---

We sincerely hope these revisions address the reviewers' valuable concerns. We are grateful
for the constructive feedback and remain open to any further questions or suggestions.
Please do not hesitate to raise additional concerns—we are committed to making any
necessary improvements to strengthen this work.

---

### Author Response · Authors · 2025-11-27
**We look forward to your feedback**

Dear reviewers,

We sincerely appreciate your insightful and constructive comments on our work, and we have accordingly prepared our revised manuscript as well as point-to-point responses to each of you.

We would be happy to answer any further questions that you might have.

Best,

The authors

---

### Author Response · Authors · 2025-11-27
**Global Reply**

Dear Reviewers,

We sincerely thank all reviewers for their thoughtful comments and constructive feedback.

We are grateful that all reviewers acknowledge the strengths of our work, noting that it is well-motivated, practical, and demonstrates strong empirical performance. We also appreciate that reviewers have provided positive evaluations. The reviewers have raised several important questions, such as how to extend to multi-person scenarios and the training/inference costs.

We have carefully considered each point to clarify, justify, and discuss potential future improvements. Detailed responses are provided below. We hope our explanations fully address the reviewers' concerns. Please do not hesitate to let us know if there are any additional details or clarifications that would be helpful.

Sincerely,

DanceTogether Authors

---

### Author Response · Authors · 2025-11-29
**Summary for Area Chair**

We sincerely thank all reviewers for their constructive feedback and recognition of our work's contributions. All reviewers acknowledged our significant workload, solid technical solutions, and valuable community resources (PairFS-4K dataset, TogetherVideoBench).

## Core Problem & Technical Novelty

**The Challenge is Fundamentally Hard.** Our work addresses maintaining persistent identity consistency during interactive contact and position swapping in multi-person videos—a problem that requires explicitly decoupling identity and motion information for each person. As shown in Fig. 17 (Line 1774), this **cannot be solved by simply scaling up data or naively introducing more conditions**.

**Why All Existing Approaches Fail.** The fundamental issue is that existing methods fail to decouple "who" from "what they do":

- **Large-scale models** (Kling, Veo3.1, Wan2.2, Sora2) trained on massive datasets catastrophically fail when actors exchange positions, suffering severe identity drift and appearance bleeding (Fig. 17).

- **Multi-condition concurrent methods.** Although some concurrent works (Multi-HumanVid@ICCV'25, EveryBodyDance@NeurIPS'25)
add mask/normal/depth maps, they still exhibit identity bleeding.
For example,
  - **Multi-HumanVid [1]** Fig. 1 shows the woman's appearance drifting to the man's upper body during lifting actions. **Using Pose+Normal+Depth as conditions.**
  - **EveryBodyDance [2]** Fig.9 and Fig.11 show no cases of position swapping—only isolated dancing without contact or occlusion challenges. **Using Pose+mask as conditions**

**The root cause**: Without explicit identity-action binding, both data scaling and condition stacking fail under severe occlusion and contact scenarios where pose estimation becomes unreliable (Fig. 14-15, Line 1270).

 - [1] Multi-HumanVid, ICCV 2025, [https://arxiv.org/pdf/2504.04126](https://arxiv.org/pdf/2504.04126)
 - [2] EveryBodyDance, NeurIPS 2025, [https://openreview.net/pdf/2a03d98ced647c0ca60e144ca2a5ddf359cf83e6.pdf](https://openreview.net/pdf/2a03d98ced647c0ca60e144ca2a5ddf359cf83e6.pdf)



**Our Technical Solution.** We introduce **identity-action binding** through MaskPoseAdapter (Fig. 5, Line 815), which explicitly binds "who does what" by fusing:
1. Robust tracking masks that remain stable under occlusion
2. Pose heatmaps for action control
3. Cross-person integration (Eq. 13, Line 856) and residual blending

This is **not incremental work**. Ablations (Tab. 6, Line 930) prove that straightforward fusion strategies fail completely. Our mask condition is also **significantly more lightweight** than normal/depth maps while being more effective.

## Addressing Reviewers' Main Concern: Scalability to N>2

1. **Most reviewers (rryZ, sBvg, E8i4) raised scalability.** We have comprehensively addressed this:

2. **Lightweight Extension (Sec. 3.4, Line 270)**: Architecture extends by simply adding Person branches—no fundamental redesign required. Training on 2-person data generalizes to 3+ persons.

3. **Remarkable Results (Sec. G, Fig. 17, Line 1774)**: Three-person sequential position-swapping succeeds where all baselines fail, demonstrating **strong technical advantage** even against models with orders of magnitude more training data.

## Data Contribution: Equally Important

**Video generation critically depends on data quality.** We constructed **PairFS-4K (7,000+ IDs, 26+ hours)**—the **first large-scale dataset providing paired individual annotations** (separate pose/mask for each person) for multi-person interactions (Tab. 1, Line 324), representing substantial effort in professional choreography, precise synchronization, and per-person annotation.

## Additional Revisions

We have addressed all minor comments (**marked in blue**), including: fixed grammatical errors; streamlined Method section (Sec. 3, Line 240-290); added missing citations (Line 150); provided inference speed comparisons (Line 1877); enhanced ablation studies (Line 1135). **Key Evidence**: Fig. 17 (3-person results), Tab. 6 (ablations), Tab. 10-11 (fair comparisons), User Study (Line 1585).

---

### Meta-Review · Area_Chair_74nM · 2026-01-07

**Summary:**

This paper proposed a controllable video generation method target for addressing videos with multiple characters. A novel MaskPoseAdapter combines both pose and mask sequence to better combine person id and the action to eliminate identity shift and appearance bleeding. Several video datasets with multiple persons are curated from existing video datasets to train and evaluate the proposed methods.

Reviewers acknowledge that the paper aims to address a critical and well-defined problem, the curated dataset is helpful and the proposed benchmark and experiments are comprehensive. Experiments also show improvement of the proposed method compared with other strong baselines. Reviewers proposed many concerns and the authors well addressed most of them. Some concerns that may not be well addressed are: 1. The overall quality of the generated videos are not high. 2. Lack of comparison with some state-of-the-art methods. For 1, though the overall quality is still not good, but it provides higher quality than the baseline methods. for 2. One paper that shows advantages of multiple person generation against the proposed one is "EverybodyDance" which is published after the deadline of ICLR submission. Considering these, the paper is a qualified publication.

**Reviewer Concerns:**

Reviewer 9khT:
The concerns from reviewer 9khT is about the presentation of the paper. The modified version of the paper improves a lot.

Reviewer rryZ:
1. The concern about the performance degradation is valid, however, compared with other baselines, the proposed method still show advantages from both quantitative and qualitative results.
2. The concern about dealing with dynamic camera motion is acknowledge by the authors. This is a valid concern, however, this is not the main focus of the paper.
3. Concern about the limited human-robot dataset is also valid, but considering the overall story, this may not be a major issue.

Reviewer sBvg:
1. Concern about the scope of the paper, it limits to two people. The authors addressed this by showing three people results in the supplementary.
2. The reviewer has concerns about the novelty of the paper. The authors claimed that the MaskPoseAdapter is novel and the task is relative new.
3. Concerns about the low quality of the pose and mask estimation. The authors claimed they have designs to try to guarantee the quality of the pose and mask in their data pipeline. Also better segmentation method such as SAM2 ro SAM3 could be used.

Reivewer E8i4:
Besides same concerns as other reivewers, reviewer E8i4 has some other major concerns:
1. Experimental comparision is unfair as the proposed method is trained with specific dataset. The author includes a baseline (StableAnimator) trained on the same dataset and still show advantages of the proposed method.
2. Compare with state-of-the-art methods. The one with best results so far may be "EverybodyDance" which is published after the deadline of ICLR submission.
3. The authors show the computational cost of the paper as the reviewer requested.

**Reviewer Scores:**

The paper original gets scores of 4, 4, 6, 4. As most of the concerns are addressed by the authors, I think they may raise their scores to 6, 6, 6, 6.

---

### Decision · Program_Chairs · 2026-01-26

Accept (Poster)